# Newborn and child-like molecular signatures in older adults stem from TCR shifts across human lifespan

Carolien E. van de Sandt [1,2], Thi H. O. Nguyen [1], Nicholas A. Gherardin [1], Jeremy Chase Crawford [3], Jerome Samir[4], Anastasia A. Minervina[3], Mikhail V. Pogorelyy[3], Simone Rizzetto [4], Christopher Szeto [5,6], Jasveen Kaur[7], Nicole Ranson [7], Sabrina Sonda[7], Alice Harper[7], Samuel J. Redmond[1], Hayley A. McQuilten [1], Tejas Menon[1], Sneha Sant[1], Xiaoxiao Jia[1], Kate Pedrina[8], Theo Karapanagiotidis[8], Natalie Cain[8], Suellen Nicholson[8], Zhenjun Chen [1], Ratana Lim[9], E. Bridie Clemens[1], Auda Eltahla[4], Nicole L. La Gruta [6], Jane Crowe[10], Martha Lappas[9], Jamie Rossjohn [6,11], Dale I. Godfrey[1], Paul G. Thomas [3], Stephanie Gras [5,6], Katie L. Flanagan [7,12,13], Fabio Luciani [4,14] & Katherine Kedzierska [1,14] ✉

CD8⁺ T cells provide robust antiviral immunity, but how epitope-specific T cells evolve across the human lifespan is unclear. Here we defined CD8⁺ T cell immunity directed at the prominent influenza epitope HLA-A*02:01-M1$_{58-66}$ (A2/M1$_{58}$) across four age groups at phenotypic, transcriptomic, clonal and functional levels. We identify a linear differentiation trajectory from newborns to children then adults, followed by divergence and a clonal reset in older adults. Gene profiles in older adults closely resemble those of newborns and children, despite being clonally distinct. Only child-derived and adult-derived A2/M1$_{58}$⁺CD8⁺ T cells had the potential to differentiate into highly cytotoxic epitope-specific CD8⁺ T cells, which was linked to highly functional public T cell receptor (TCR)αβ signatures. Suboptimal TCRαβ signatures in older adults led to less pro-liferation, polyfunctionality, avidity and recognition of peptide mutants, although displayed no signs of exhaustion. These data suggest that priming T cells at different stages of life might greatly affect CD8⁺ T cell responses toward viral infections.

CD8⁺ T cells provide antiviral immunity by eliminating virus-infected cells and establishing long-term immunological memory[1–4]. CD8⁺ T cells recognize epitopes, peptides (p) bound to HLA class I (pHLA-I) on the surfaces of infected cells via TCRαβ. Memory CD8⁺ T cells recognize conserved viral peptides, providing broad-protection against mutat-ing viruses and ameliorate disease severity[1,2]. Epitope-specific CD8⁺ T cells in adults are well studied, but less so in newborns, children

and older adults. To rationally design effective CD8⁺ T cell-targeted immunotherapies and vaccines for all ages, we need to understand how epitope-specific CD8⁺ T cells evolve across human life.

Diversity and clonal composition of TCR repertoires affect T cell functionality and antiviral protection[5–7]. Limited data exist on how age-related TCR changes affect magnitude, functionality and gene profiles of epitope-specific T cells. Previous age-specific TCR studies

were mostly based on bulk, rather than epitope-specific T cells and/or in vitro-expanded T cell lines, and encompassed bulk sequenced TCRα or TCRβ chains, instead of paired TCRαβ. Bulk naive TCR repertoires at birth are highly diverse, lack clonal expansions and nucleotide insertions in complementarity-determining region 3 (CDR3) regions[8,9]. Similar trends were observed for Epstein–Barr virus (EBV), cytomegalovirus (CMV) and Melan-A/MART-1-specific CD8+ T cells[8,10]. Newborn influenza A2/M1_58-specific TCRβ repertoires were biased toward TRBV19, which persisted into adulthood[10,11]. Childhood infections clonally expand virus-specific T cells, further maintained in adults[4,8,10,12,13]. With age, bulk and epitope-specific TCR repertoires showed declining TCR diversity, fewer non-N-inserted clonotypes and longer CDR3α sequences[8,9,14–16]. Aging compromised T cell immunity, including increased terminally differentiated T ($T_{EMRA}$) cells, decreased naive T cells and T cell functionality[8,12,17–21], resulting in diminished recall capacity[22,23]. These changes are defined as immunosenescence[24,25].

Ex vivo epitope-specific, single-cell paired TCRαβ analyses across age groups remain rare and were performed by us for influenza-specific[16] and SARS-CoV-2-specific CD8+ T cells[13]; however, not across the human lifespan. Highly functional public (shared) clonotypes dominated adult A2/M1_58-specific TCRαβ repertoires[16,26,27], whereas older adults had prominent private clonotypes (not shared)[16]. Children had fewer expanded SARS-CoV-2-specific clonotypes compared to adults, despite common TCRαβ motifs[13]. Underlying mechanisms and functional consequences of age-related changes within epitope-specific TCR repertoire across the human lifespan remain unresolved. Limited epitope-specific T cell single-cell RNA-sequence (scRNA-seq) studies focus on adult versus older bulk T lymphocytes in healthy individuals[28,29] or those with COVID-19 (ref. 30), but not across the human lifespan.

We defined epitope-specific CD8+ T cell immunity across the human lifespan ex vivo in newborns, children, adults and older adults. We incorporated single-cell transcriptome and paired TCRαβ analyses to define epitope-specific T cells directed at the prominent and conserved HLA-A*02:01-restricted M1_58–66 peptide derived from influenza A viruses (A2/M1_58) (refs. 11,16,31–33). Public clonotypes present across different individuals[16,26,27,34] allowed us to track numerical, phenotypic, functional and molecular changes within public A2/M1_58+CD8+ TCRαβ clonotypes across human lifespan. We identified age-related TCR repertoire shifts within older epitope-specific CD8+ T cells, stemming from newborn/child-like molecular signatures detected in older adults. Our findings have implications for rationally designed T cell-targeted vaccines and immunotherapies across age groups.

## Results

### Lifespan HLA-A*02:01+ cohort

For our 'lifespan' cohort, we recruited healthy HLA-A*02:01-expressing individuals across four immunologically distinct age groups: newborns ($n = 11$, 0 years), children ($n = 12$; median 9 years, range 3–16), adults ($n = 20$; median 37 years, range 18–58) and older adults ($n = 18$; median 72 years, range 63–88) (Fig. 1a,b and Supplementary Table 1).

We performed ex vivo phenotypic, transcriptome, functional and TCRαβ repertoire analyses of A2/M1_58-specific CD8+ T cells.

### A2/M1_58+CD8+ T cells frequencies peak in adults

Magnitude and phenotype of CD8+ T cells and A2/M1_58+CD8+ T cells were assessed ex vivo using tetramer-associated-magnetic enrichment (TAME) (Fig. 1c and Extended Data Fig. 1a). Frequency of total CD8+ T cells was lowest in newborns (median 15.7%), increased in children (21.5%), peaked in adults (21.9%) and bimodal frequencies were found in older adults (19.1%) (Fig. 1d,e). A2/M1_58+CD8+ T cell frequencies were lowest in newborns (median $5.34 × 10^{-5}$) and children ($7.43 × 10^{-5}$), peaked in adults ($2.33 × 10^{-4}$) and decreased in older adults ($7.36 × 10^{-5}$) (Fig. 1f,g), aligning with reports of declining total CD8+ T cells[17] and A2/M1_58+CD8+ T cells[16] in immunosenescent older adults.

### Age-related phenotypic changes in A2/M1_58+CD8+ T cells

Ex vivo epitope-specific A2/M1_58+CD8+ T cells and total CD8+ T cells displayed different phenotype profiles (Fig. 1c,h–m and Extended Data Fig. 1a–d). Naive A2/M1_58+CD8+ T ($T_{naive}$) cells (CD27+CD45RA+CD95−) decreased with age, newborns (median 98.5%), children (20.4%), adults (6.5%) and older adults (3.2%). Notably, 37.5% of older adults maintained substantial A2/M1_58+CD8+ $T_{naive}$ cells (>10%). A2/M1_58+CD8+ central memory T ($T_{CM}$) cells (CD27+CD45RA−) peaked in adults (80.7%). Children and older adults displayed bimodal A2/M1_58+CD8+ $T_{CM}$ cell profiles, potentially reflecting influenza exposures in children. A2/M1_58+CD8+ effector memory T ($T_{EM}$) cell (CD27−CD45RA−) and stem cell memory ($T_{SCM}$) cell (CD27+CD45RA+CD95+) populations were low across age groups (<10%). Notably, A2/M1+CD8+ $T_{EMRA}$ (CD27−CD45RA+) cells remained low across age groups (<10%), conversely to increasing $T_{EMRA}$ levels within total CD8+ T cells in older adults. Ample $T_{EMRA}$ populations are characteristic for immunosenescence in older adults experiencing chronic CMV infections[12,17]. Similar trends were observed for absolute numbers of A2/M1_58+phenotype+cells/$10^6$ CD8+ T cells (Extended Data Fig. 1b).

CD57 expression, associated with immunosenescence and terminal differentiation[18,35,36], on total CD8+ T cells peaked in older adults (60.8%) (Fig. 1j), but remained low on A2/M1_58+CD8+ T cells across age groups (6.7%; 7 CD57+A2/M1_58+CD8+ T cells/$10^6$CD8+ T cells), including older adults (8.3%; 14 CD57+A2/M1_58+CD8+ T cells/$10^6$CD8+ T cells) (Fig. 1k and Extended Data Fig. 1d). PD-1 expression, an immune checkpoint marker that can be associated with TCR activation, immunosuppression and/or exhaustion[37], was low on total CD8+ T cells across ages (14.5%) (Fig. 1l), whereas PD-1+A2/M1_58+CD8+ T cell frequencies were high in children (54.4%; 37 PD-1+A2/M1_58+CD8+ T cells/$10^6$CD8+ T cells) and adults (57.5%; 138 PD-1+A2/M1_58+CD8+ T cells/$10^6$CD8+ T cells), and decreased below 50% in 68.8% of older adults (41 PD-1+A2/M1_58+CD8+ T cells/$10^6$CD8+ T cells) (Fig. 1m and Extended Data Fig. 1d). CD57/PD-1 and CD38/HLA-DR coexpression was minimal (Extended Data Fig. 1c–f).

CMV testing would reveal whether CMV status affects total CD8+ and/or A2/M1_58-specific CD8+ T cell phenotypes (Supplementary Table 1). Although our dataset was underpowered, A2/M1_58-specific CD8+ T cell

**Fig. 1 | Age-related changes in A2/M1_58+CD8+ T cell frequencies and phenotypes. a**, 'Lifespan' HLA-A*02:01-positive cohort, median age and number of donors per age category. **b**, Age distribution within the HLA-A*02:01-expressing lifespan cohort. **c**, Representative FACS panels and gating strategy for A2/M1_58+CD8+ T cells in the enriched fraction and phenotypic populations $T_{CM}$ (CD27+CD45RA−) cells, $T_{EM}$ (CD27−CD45RA−), $T_{EMRA}$ (CD27−CD45RA+), $T_{naive}$ (CD27+CD45RA+CD95−) and $T_{SCM}$ (CD27+CD45RA+CD95+) cells. Gray dots represent total CD8+ T cells in the unenriched sample, red dots are A2/M1_58+CD8+ T cells in the enriched sample. **d**–**g**, Proportion of total CD8+ T cells (**d,e**) and frequency of A2/M1_58+CD8+ T cells (**f,g**) across different age groups. Open symbols indicate <10 A2/M1_58+CD8+ T cells counted, which were not used for phenotypic analyses. **h,i**, Frequency of naive and memory subsets within the total CD8+ T cell (**h**) or A2/M1_58+CD8+ T cell populations (**i**) across all age groups. **j**–**m**, Frequencies of CD57 and PD-1 expression on CD8+ T cells (**j** and **l**, respectively) and A2/M1_58+CD8+ T cells (**k** and **m**, respectively) per age group. Horizontal bars indicate the median, dots represent individual donors, with $n = 11$ newborns, $n = 12$ children, $n = 20$ adults and $n = 18$ older adults (**b–h,j,l**) and $n = 10$ newborns, $n = 12$ children, $n = 20$ adults and $n = 16$ older adults (**i,k,m**). Black line is a locally estimated scatter-plot smoothing) Loess trend line with error bands shaded in gray representing 95% confidence interval (CI) (**e,g**). Technical replicates were not performed due to limited samples. Statistical analysis was performed using a two-sided Kruskal–Wallis with Dunn's correction for multiple tests. P values are indicated above the graphs. N, newborn; C, children; A, adult; OA, older adult.

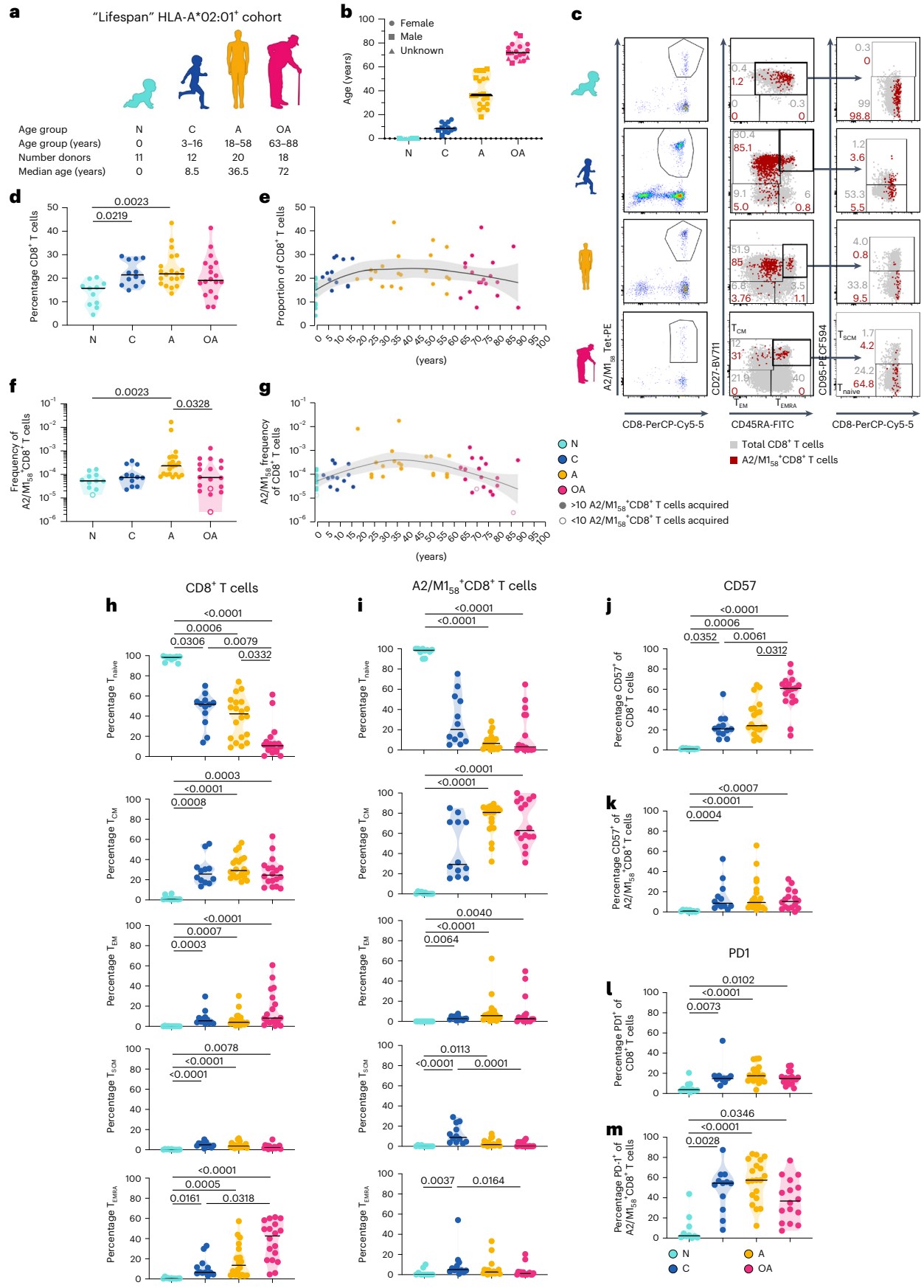

responses were not modulated by concurrent CMV infection, conversely to total CD8[+] T cells (Extended Data Fig. 2).

Overall, A2/M1$_{58}$-specific CD8[+] T cell phenotypes change across the human lifespan, but these changes are distinct from total CD8[+] T cells, with absence of terminally differentiated A2/M1$_{58}$[+]CD8[+] T cells in older adults.

## Four A2/M1$_{58}$[+]CD8[+] T cell clusters across human lifespan

We defined molecular gene signatures in healthy HLA-A*02:01-expressing newborns, children, adults and older adults (n = 3 per group), selected to reflect phenotypic heterogeneity, including dispersion of heterogenicity within T$_{CM}$ and T$_{naive}$ populations in children and older adults, and lack of T$_{EMRA}$ in older adults (Extended Data Fig. 3a). Ex vivo-isolated A2/M1$_{58}$[+]CD8[+] T cells were index-sorted for single-cell transcriptome analysis. A total of 793 cells across all age groups passed quality control for analyses.

Unsupervised dimensionality reduction analysis using Uniform Manifold Approximation and Projection (UMAP) maps revealed four distinct A2/M1$_{58}$[+]CD8[+] T cell clusters (Fig. 2a). Newborns were predominantly detected in clusters 1 and 2, mainly containing naive cells. Memory A2/M1$_{58}$[+]CD8[+] T cells dominated cluster 0 encompassed children, adults and older adults, whereas adults dominated cluster 3 (Fig. 2a,b).

Gene expression profiles in each cluster were consistent with protein-expressed phenotypic profiles obtained from index-sort analysis (Fig. 2c and Extended Data Fig. 3b). Clusters 1 and 2, containing mostly naive cells from newborns, expressed *SELL*, *CCR7* and transcription factors *TCF7* and *SOX4*, hallmark genes for naive T cells. Cluster 2 expressed higher *CD69*, *FOS* and *STAT3* levels, indicating increased activation and TCR signaling compared to cluster 1. Cluster 0 consisted of A2/M1$_{58}$[+]CD8[+] T cells expressing effector and memory markers, including *GZMK*, *CCL5*, *CXCR4*, *CD28* (Fig. 2c) and transcription factors *NR4A2*, *KLF2*, *FOS* and *JUN*. A2/M1$_{58}$[+]CD8[+] T cells in cluster 3 expressed cytotoxicity-associated genes, *GZMA*, *PRF1*, *NKG7* and *KLRD1*, with high expression of *TRAV27*/*TRBV19* gene segments. Cluster properties were confirmed by gene set enrichment analysis (GSEA; Extended Data Fig. 3c).

TRBV19 and TRAV27 represent key features of optimal A2/M1$_{58}$[+]CD8[+] T cells, especially with arginine (R) and serine (S) in CDR3β (refs. 26,27,38). Full-length paired TRAV27-TRBV19-CDR3-RS clonotypes and (TRBV19-)CDR3-RS clonotypes paired to different/unknown TCRα chains were identified mostly within clusters 0 and 3, confirming segregation of clonally expanded A2/M1$_{58}$[+]CD8[+] T cells with varying functional gene features (Fig. 2a,c). TRBV27, uncommon in memory A2/M1$_{58}$[+]CD8[+] T cells, was enriched in cluster 2 (Fig. 2c). UMAP analysis excluding TCR genes revealed a similar four-cluster composition, indicating that clonally expanded A2/M1$_{58}$[+]CD8[+] T cells did not alter cluster compositions (Extended Data Fig. 3d).

Overall, scRNA-seq combined with protein expression phenotypes revealed differential molecular signatures of A2/M1$_{58}$[+]CD8[+] T cells across four clusters, reflecting different age groups, separating naive A2/M1$_{58}$[+]CD8[+] T cells associated with newborns and effector/memory A2/M1$_{58}$[+]CD8[+] T cells mostly found in other age groups.

## Young gene profiles resemble gene profiles for older adults

To identify age group-distinctive gene signatures, we performed differential gene expression analysis between cells stratified by age and

investigated heterogenicity within A2/M1$_{58}$[+]CD8[+] T cells in children and older adults, and lack of exhaustion and terminal differentiation in older A2/M1$_{58}$[+]CD8[+] T cells.

Newborn A2/M1$_{58}$[+]CD8[+] T cells expressed naive gene signatures, with high expression of *CCR7*, *SELL* and *TNFAIP3*, which became less prominent in children and adults, but increased in older adults. Adult A2/M1$_{58}$[+]CD8[+] T cells expressed effector-memory phenotype profiles, including *LITAF*, *KLRG1*, *CXCR4* (contributing to homeostasis self-renewal and homing)[39] and NK-like signatures, *KLRK1* encoding NKG2D (involved in stress-induced cytotoxic response)[40], less prominent in children and older adults (Fig. 2d). Mixed naive/memory phenotypes in children and older adults (Fig. 1i) were verified by shared mixed gene expression profiles (Fig. 2d).

Cytotoxic gene signatures, *ITGB1*, *GNLY*, *CST7*, *GZMA*, *PRF1* and *GZMK*, detected in children, became more pronounced in adults, reflecting their T$_{CM}$/T$_{EM}$ phenotype. Conversely, older adults had lower expression of cytotoxicity genes, despite substantial A2/M1$_{58}$[+]CD8[+] T$_{CM}$ populations, suggesting that older A2/M1$_{58}$[+]CD8[+] T cells are less cytotoxic. Newborns uniquely expressed *LTB* (encoding TNF-C), but no other cytotoxic genes (Fig. 2d). GSEA between age groups confirmed predominant cytotoxic-effector/memory signatures in children and adults, and revealed enriched naive signatures in newborn and older adults, largely driven by reduced effector and differentiated states of A2/M1$_{58}$[+]CD8[+] T cells (Extended Data Fig. 3e).

Newborn A2/M1$_{58}$[+]CD8[+] T cells expressed high levels of *ZFP36L1* and *PIK3IP*, reflective of quiescent naive T cells. High *PIK3IP1* expression in older adults corresponded to heterogeneity within older A2/M1$_{58}$[+]CD8[+] T cells (Fig. 2d). Markers associated with maintaining immune control (*FYN* and *LYAR*), anti-inflammatory cytokines (*IL-10RA*) and controlling T cell differentiation (*ZFP36* and *TXNIP*) were highly expressed in adult A2/M1$_{58}$[+]CD8[+] T cells, less pronounced in children and older adults and absent in newborns. CD37, inhibiting TCR signaling[41], was uniquely expressed in older A2/M1$_{58}$[+]CD8[+] T cells (Fig. 2d).

Transcription factors associated with naive or resting T memory phenotypes (*FOXP1*, *TCF7*, *SOX4* and *LEF1*) were highly expressed in newborns, whereas adults expressed *ID2*, associated with terminal differentiation[42] (Extended Data Fig. 3f). Older adults expressed AP-1 transcription factors *FOS*, *JUNB* and *KLF2*, associated with protection from exhaustion[43], (Extended Data Fig. 3f). Exhaustion-associated transcription factors (*TOX*), exhaustion markers (*TIGIT*, *CTLA4*, *LAG3* and *PDCD1*) and senescence marker CD57 (*B3GAT1*) were minimally expressed across age groups, supporting phenotypic data (Fig. 2d,e and Extended Data Fig. 3f). Pairwise comparison of children and older adult gene profiles revealed that older A2/M1$_{58}$[+]CD8[+] T cells had more heterogenous distribution, with less-differentiated and cytotoxic subsets (Extended Data Fig. 3g). Gene profiles of older adults were more like newborns, whereas children and adult A2/M1$_{58}$[+]CD8[+] T cells had increased effector and cytotoxicity signatures.

As TCR signatures are important in antiviral immune responses, we investigated TCR-associated genes across age groups. *TRBV19* and *TRAV27* gene expression, key features of highly functional A2/M1$_{58}$[+]CD8[+] T cells[26,27,38], increased from children to adults, whereas *TRBV27* expression was higher in newborns and older adults (Fig. 2d). Age-specific TCR changes suggest potential shifts in dominance of key TCR clonotypes across the human lifespan.

**Fig. 2 | Molecular differentiation of A2/M1$_{58}$[+]CD8[+] T cells across human lifespan. a**, Dimensionality reduction (UMAP) and clustering of scRNA-seq data colored by clusters (top left), age groups (top middle), phenotype (top right), TCR features (bottom left) and clone size (bottom right). **b**, Distribution of age (left) and phenotype (right) in each UMAP cluster. Single-cell phenotypes were obtained via index-sorting protein expression data. **c,d**, Selected genes identified from differential expression analysis between A2/M1$_{58}$[+]CD8[+] T cells in each UMAP cluster (**c**) or between A2/M1$_{58}$[+]CD8[+T cells] grouped by age (**d**) using pairwise comparison with a two-side hurdle model (MAST) without correction

for multiple comparison (P < 0.05). Dot size represents the proportion of cells with non-zero expression. **e**, Comparison of exhaustion gene expression levels between A2/M1$_{58}$[+]CD8[+] T cells grouped by their age, three donors per age group with n = 174 single A2/M1$_{58}$[+]CD8[+] T cells from newborns, n = 219 from children, n = 261 from adults and n = 139 from older adults. Pairwise group comparisons were performed with two-sided Wilcoxon rank-sum test and Bonferroni-adjusted P values are reported (*P < 0.05, **P < 0.01, ***P < 0.001, ****P < 0.0001). N, newborn; C, children; A, adult; OA, older adult.

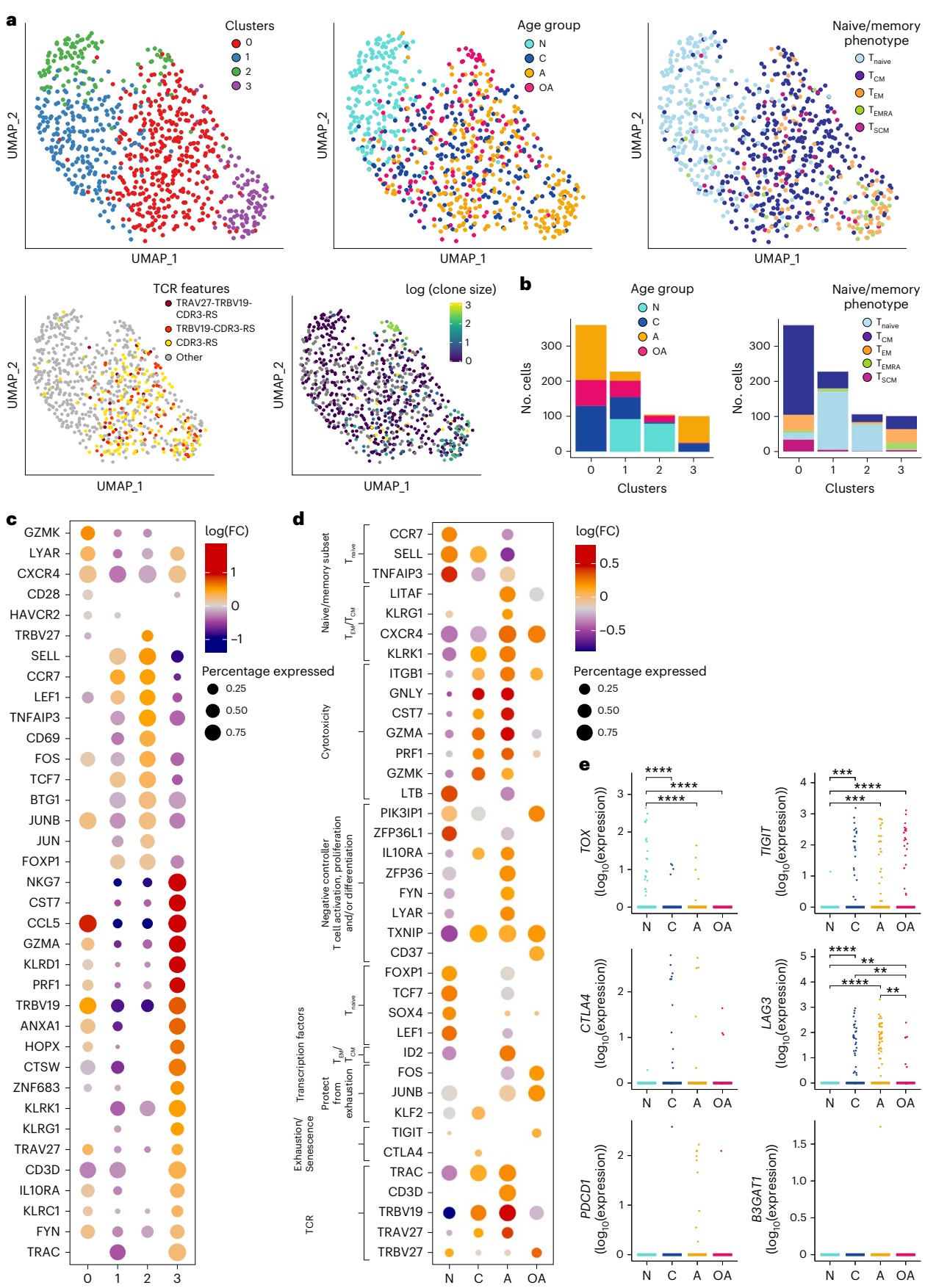

Overall, differentially expressed gene profiles across the human lifespan revealed clear distinctions between naive $A2/M1_{58}{}^+CD8^+$ T cells in newborns, mixed naive/memory profiles in children, cytotoxic-effector-memory profiles in adults and inversion toward naive profiles in older adults, without evidence for exhaustion and terminal differentiation, agreeing with phenotypic data (Fig. 1i–m).

**Age-specific molecular changes stem from distinct clonotypes**
UMAP analysis demonstrated heterogeneity among $A2/M1_{58}{}^+CD8^+$ T cells, with clusters segregating cells based on age group and phenotype, with newborn and older adults sharing common gene and phenotypic signatures (Fig. 2). We thus asked whether distinct lineages can explain molecular and phenotypic differences observed across the human lifespan. We hypothesized two scenarios (1) $A2/M1_{58}$-specific TCR clonotypes are shared between children, adults and older adults, with older TCR clonotypes reverting to a quiescent gene profile similar to newborns and children; and (2) older adults have distinct $A2/M1_{58}$-specific TCR clonotypes compared to adults, with gene expression profiles similar to those detected in newborn and children. To evaluate the relationship between differentiation and fate of the clonal lineage of $A2/M1_{58}{}^+CD8^+$ T cells across age groups, we performed pseudotime trajectory analysis using partition-based graph abstraction (PAGA) based on gene expression data, combined with clonotype information to infer the connections and order of differentiation throughout T cell states (Fig. 3).

Similar to UMAP, PAGA analysis confirmed that $A2/M1_{58}{}^+CD8^+$ T cells separate into clusters based on age group and phenotype with additional segregation of newborns and older adults into two distinct clusters, clusters 3 and 4 (Fig. 3a,b). TCR mapping identified highly functional TRAV27-TRBV19-CDR3-RS TCR features dominated cluster 0 and 2, whereas TRBV27-expressing clonotypes dominated the older adult cluster 4 (Fig. 3c).

We established the $A2/M1_{58}{}^+CD8^+$ T cell pseudotime trajectory across five PAGA clusters. Each cell was assigned a state based on dimensionality reduction or clustering in PAGA. Pseudotime values were inferred for each cell, allowing cells to be ordered along a trajectory across which they can be considered as proxy of cell lineage differentiation (Fig. 3d). Cluster 3, consisting of predominantly naive T cells from newborns, was the logical choice for the differentiation root (Fig. 3b,d). A trajectory was identified by connecting root cluster 3 with clusters 1 and 0, and terminating in cluster 2 (trajectory 1) (Fig. 3a). This trajectory largely correlated with clonal expansion and increased usage of TRAV27-TRBV19-CDR3-RS features (Fig. 3c). Older TRBV27-expressing $A2/M1_{58}{}^+CD8^+$ T cells dominating cluster 4 (Fig. 3c), suggested deviation, branching off from cluster 0 and terminating in cluster 4 (trajectory 2), indicating distinct lineage differentiation in older adults.

To quantify variation of gene and phenotypic signatures along the pseudotime (Fig. 3d), we performed Loess smoothing fit. T cell phenotype analysis showed rapid declining growth rates of naive $A2/M1_{58}{}^+CD8^+$ T cells along the pseudotime, whereas growth rates of $T_{CM}$ and $T_{EM}$ $A2/M1_{58}{}^+CD8^+$ T cells increased, the latter largely consisting of cytotoxic effectors from adults displaying TRBV19-CDR3-RS features (Fig. 3a–c). Indeed, gene expression analysis demonstrated a rapid increase of *TRAV27* and *TRBV19*, along with *NKG7*, *GNLY* and *GZMA* expression, representing increased cytotoxicity profiles along trajectory 1. Conversely, naive-associated genes, including *CD27* and

transcription factors *FOS* and *JUNB* declined over the pseudotime, whereas TCR signaling genes (*LCK*) remained highly expressed.

Overall, our trajectory analysis demonstrated nonlinear differentiation of $A2/M1_{58}{}^+CD8^+$ T cells across the human lifespan. Trajectory 1 consisted of newborns, children and adults along a differentiation branch toward the effector cytotoxic $A2/M1_{58}{}^+CD8^+$ T cells associated with optimal TCR features dominating adult $A2/M1_{58}{}^+CD8^+$ repertoires. Trajectory 2 was dominated by older less-differentiated T cells, encompassing a distinct clonal lineage.

**Diverse TCRαβ repertoires in newborns and older adults**
TCRαβ repertoire diversity and clonal composition affect functionality of $CD8^+$ T cells[5–7]. As transcriptomic analyses revealed prominent *TRAV27/TRBV19* expression in children and adults, and *TRBV27* in newborns and older adults (Fig. 3c), we asked whether TCRαβ signatures underly age-specific gene profiles. We dissected TCRαβ clonal diversity and composition ex vivo within HLA$^-$A*02:01-expressing newborns ($n = 6$), children ($n = 12$), adults ($n = 8$) and older adults ($n = 10$) (Figs. 4–6 and Extended Data Figs. 4 and 5). We analyzed 1,110 paired $A2/M1_{58}{}^+CD8^+$ TCRαβ clonotypes, 66 single TCR α-chains, 447 single TCR β-chains, with unidentifiable matching TCR β-chains or TCR α-chains, respectively (Supplementary Table 2).

Gene segment usage and gene–gene pairing landscapes were generated for pooled TCRαβ repertoires within each group (Fig. 4a–c). Two-dimensional (2D)-kernel principal-component analysis (kPCA) projection of TCR Vα/Vβ/Jα/Jβ gene segments demonstrated diverse TCR repertoires in newborns, clustering in children and adults, before diversifying in older adults (Fig. 4a). TCR diversity scores (TCRdiv) for $A2/M1_{58}{}^+CD8^+$ demonstrated that both TCR α-chains and β-chains contributed to initial clustering from newborns to adults, and subsequent diversification in older adults (Fig. 4b). Non-newborn TCR repertoires showed more heterogeneity across individuals than within individuals, but not in newborns despite higher repertoire diversity (Extended Data Fig. 4a). To measure density within age-specific TCR repertoires and quantify relative contributions of clustered and diverged TCRs, neighbor distance distributions were calculated, where lower average values of distance distribution peaks represented similar clonotype clustering (Fig. 4c). Bimodal distribution of single and paired TCRαβ sequences was observed across age groups, except for newborn single-TCRα sequences, characterized by a single high distribution peak (average newborns αβ, 192.3; β, 70.9; α, 91.4; children αβ, 149.0; β, 49.6; α, 77.4; adults αβ, 103.1; β, 28.4; α, 59.0; older adults αβ, 161.3; β, 57.9; α, 80.8) (Fig. 4c).

Overall, $A2/M1_{58}{}^+CD8^+$ TCRαβ repertoires are highly diverse in newborns, greatly cluster in children and adults, before diversifying in older adults. These changes are attributed to both TCR α-chains and β-chains.

**Young public TCRs replaced by private TCRs in older adults**
Circos analysis of pooled TRAV and TRBV sequences was performed to understand how changes in TCRαβ diversity related to gene segment usage (Fig. 4d and Extended Data Fig. 4b). Newborn $A2/M1_{58}{}^+CD8^+$ TCRαβ repertoires were highly diverse, although newborns expressed TRBV19 bias[10,11]. Clonally expanded TRBV19-expressing TCRs became more prevalent in children and adults. Strong TRAV27–TRBV19 associations were observed in children (10 out of 12 children, 82 out of 714

**Fig. 3 | Single-cell transcriptomics shows evolutionary trajectory across human life. a**, PAGA analysis of single cells ($n = 793$) identified five clusters (left) and their relative connectivity (middle) and colored based on the four UMAP clusters (right). **b**, PAGA analysis colored by age (left), phenotype (right). Single-cell phenotypes were obtained via index sorting. **c**, PAGA analysis colored by public TCR features (left), TRBV27 expression (middle) and clone size (right). Bar charts represent respective age, phenotype and TCR features distributions in each PAGA cluster. **d**, Trajectory derived from scRNA-seq data with colored pseudotime values (large panel). Loess curves represent the changes along the trajectories of age groups (top left), phenotype based on protein expression (top right), TCR features (middle) and the dot-plot shows expression values of selected genes measured via scaled mean expression along the pseudotime size of the dots corresponds to the percentage of cells expressing the gene in the given pseudotime state (bottom). Colored lines are a Loess fit, with error bands shaded in matching colors representing the 95% CI (**d**, bottom left). N, newborn; C, children; A, adult; OA, older adult.

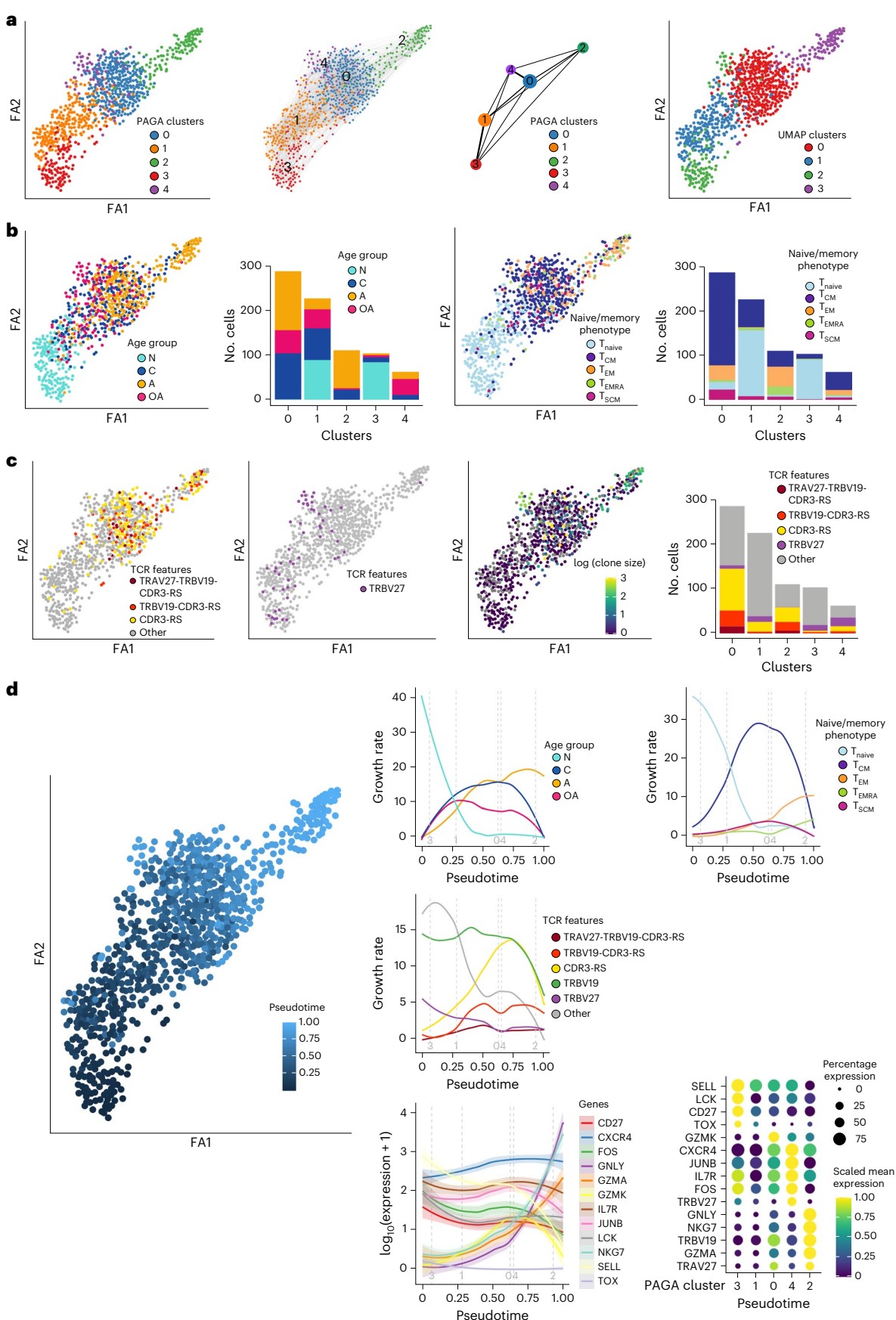

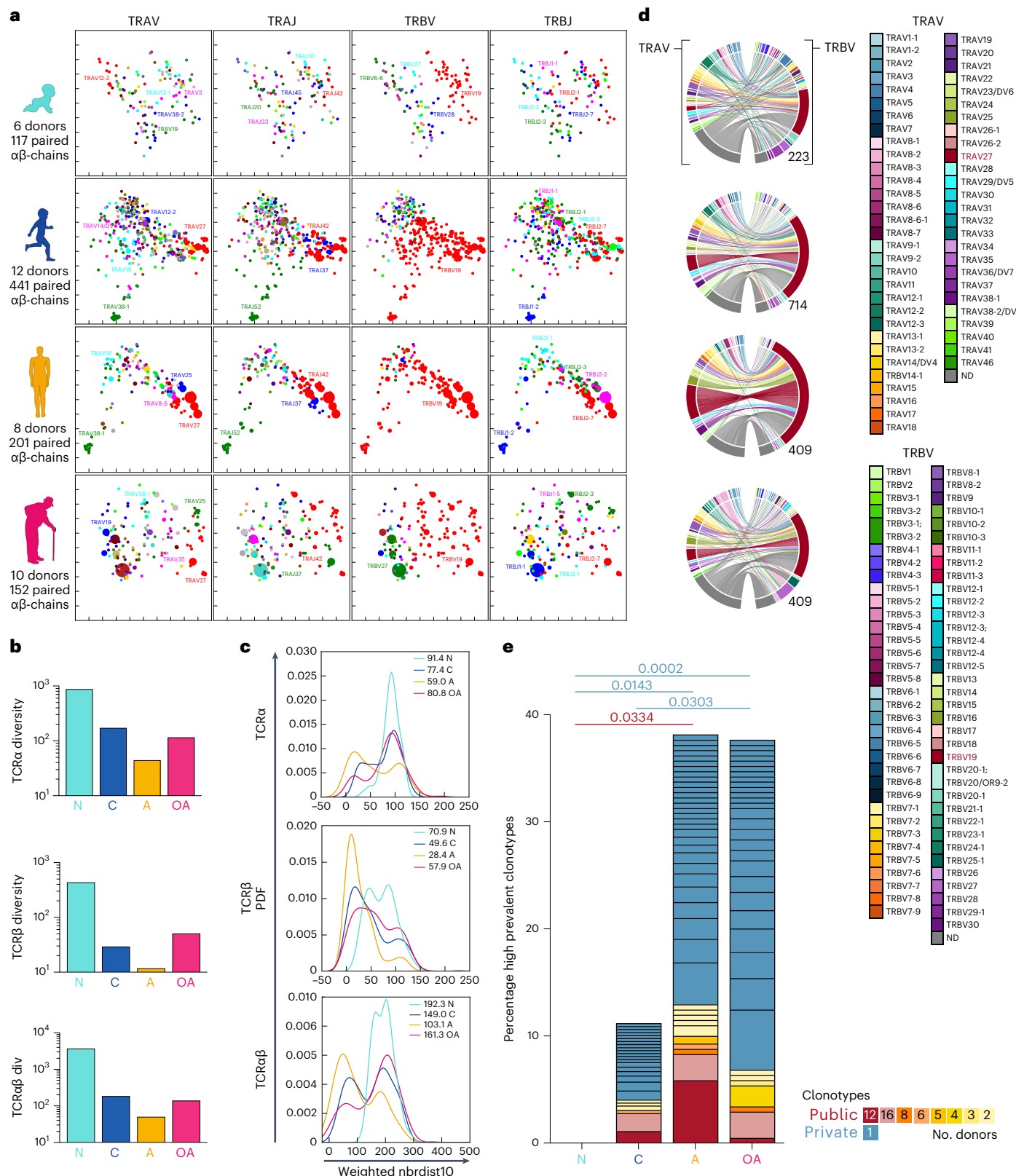

TCRαβ clonotypes) and became more pronounced in adults (8 out of 8 adults, 96 out of 409 TCRαβ clonotypes). Consistent with kPCA and neighbor distance distribution analyses, higher TRAV and TRBV diversity was observed among older adults. TRBV19 became less prevalent in six out of ten older adults and the TRBV19–TRAV27 association was observed in four out of ten older donors (33 out of 409 TCRαβ

clonotypes). Instead, older adults displayed large clonal expansions expressing other TRAV and TRBV gene segments.

We defined distribution of high-prevalent public (shared clonotypes, detected at least twice within each individual), low-prevalent public (shared clonotypes, detected only once within each individual) and high-prevalent private clonotypes (not shared, detected at least

**Fig. 4 | Age-related changes in A2/M1$_{58}$$^+$CD8$^+$ TCRαβ repertoire. a–e**, A2/M1$_{58}$$^+$CD8$^+$ T cells were enriched by TAME followed by single-cell sorting for TCRαβ analysis. **a**, A 2D kernel principal-component analysis (PCA) projection of the A2/M1$_{58}$$^+$CD8$^+$ TCR landscape colored by Vα, Jα, Vβ and Jβ gene usage (left to right) for all four age groups generated by TCRdist. Encoding clone size indicated by symbol size. **b**, TCRdiv diversity measures of the TCRα, TCRβ or paired TCR αβ-chains. **c**, smoothed density profiles of neighbor distance distribution are shown for each age group. A lower distribution peak indicates more clustered A2/M1$_{58}$$^+$CD8$^+$ single TCRα, TCRβ or paired TCRαβ repertoire, average distance values for each age group are depicted within the plot. PDF, probability density function. **d**, TRAV and TRBV clonotype pairing per age group illustrated by circos plots. Left arch segment colors indicate TRAV usage, right outer arch colors depict TRBV usage. Connecting lines indicated *TRAV–TRBV* gene pairing and are colored based on their TRAV usage and segmented based on their CRD3α and CDR3β sequence, the thickness is proportional to the number of TCR clones with the respective pair. The number of sequences considered for each circos plot is shown at the right bottom. **e**, Frequency of high-prevalent (>2 similar TCRs within a single individual) public (shared) and private (not shared) clonotypes across different age groups. Dark red represents high-prevalent public TCR (TRAV27, TRAJ42, CDR3α GAGGGSQGNLIF, TRBV19, TRBV2–7 and CDR3β CASSIRSSYEQYF), whereas the light red are clonotypes expressing the full public TCRβ chain (TRBV19, TRBV2–7 and CDR3β CASSIRSSYEQYF) but the TCR α-chain could not be identified. Numbers in the graph represent the number of donors in which this specific high-prevalent clonotype was identified. Statistical analysis was performed using a two-sided Kruskal–Wallis test with Dunn's correction for multiple tests. *P* values are indicated above the graphs. N, newborn; C, children; A, adult; OA, older adult.

twice in a single individual). High-prevalent public TCRαβ clonotypes were observed during childhood, peaked in adults, before decreasing in older adults, particularly the previously identified full public TCRαβ clonotype (TRAV27/TRAJ42, CDR3α-GAGGGSQGNLIF, TRBV19/TRBV2–7 and CDR3β-CASSIRSSYEQYF) associated with optimal immunity in adults[16,26,27,34] (Fig. 4e and Extended Data Fig. 4c). Loss of public TCRαβ clonotypes in older adults was associated with increased prevalence of private TCR clonotypes (Fig. 4e and Extended Data Fig. 4d).

Our data suggest that repeated influenza virus exposures expand public TCRαβ clonotypes in children and adults, which are replaced by private TCRαβ clonal expansions in older adults.

**Young public CDR3αβ-motifs are less frequent in older adults**

As hypervariable CDR3α and CDR3β regions predominantly mediate fine pHLA-I specificity, we dissected CDR3αβ regions by analyzing length and amino acid sequence to identify TCR motifs. Conversely to diverse lengths observed for A2/M1$_{58}$$^+$CD8$^+$ CDR3α regions across age groups, newborn CDR3β sequences were predominantly 8–10 amino acids in length, whereas an eight-amino acid length dominated in children, adults and older adults (Extended Data Fig. 5a,b and Supplementary Table 2).

We identified CDR3 motif similarities to highlight key conserved residues driving A2/M1$_{58}$-specific TCR recognition either within or between age groups. No CDR3α motif was identified in newborns, attributed to high TCR α-chain diversity. A single top-scoring, TRAV27-TRAJ42-associated, glycine-rich, CDR3α-(CA)GGGSQG(NLI) motif was identified in children, adults and older adults (Fig. 5a). A single glycine was enriched above background levels, suggesting limited involvement in specific pHLA interaction[44]. Clonotypes expressing the full public TRAV27-TRAJ42-associated CDR3α-GAGGGGSQGNLIF or shorter variants, including GGGSQG, GGG or GG were shared between age groups with frequencies increasing in children, peaking in adults and decreasing in older adults (Fig. 5b,c), confirming our finding that TCRα chains contributed to diversifying the older TCR repertoire (Fig. 4b).

Several top-scoring (chi-squared ≥ 100) CDR3β-motifs were identified within age groups (Fig. 5a). TRBV19-associated CDR3β-'IV'-motifs were uniquely identified in three out of five newborns.

Generally, 'IV'-expressing CDR3β clonotypes were detected at a low frequency in children, adults and older adults, except in child TN022. No 'IV'-expressing clonotypes were shared between age groups (Fig. 5d,e and Supplementary Table 2).

A2/M1$_{58}$-specific TRBV19-TRBJ2-7-associated CDR3β-'RS' motifs were prominent in children, adults and older adults (Fig. 5a). CDR3β-'RS'-expressing TCRαβs were frequently shared between individuals across age groups (Fig. 5d). Enriched 'RS' residues are essential for the peg-notch mode of recognition, interacting extensively with the M1$_{58–66}$ peptide and HLA-A*02:01 molecule[26,27,38,44–46]. High-frequency CDR3-'RS'-expressing clonal expansions were observed in children, peaked in adults and decreased in 80% of older adults (Fig. 5d,e). Changes in 'RS'-expressing clonotypes were partially attributed to the public TRAV27/TRAJ42-CDR3α-GAGGGSQGNLIF; TRBV19/TRBV2-7-CDR3β-CASSIRSSYEQYF clonotype (Figs. 4e and 5d).

TRBV19-TRBJ1-2-associated (C)ASSIGxxYGYT(F) CDR3β-motifs harboring 'IG' and 'YGY' residues was among the top-scoring CDR3β motifs in children and adults (Fig. 5a). Although TCRs expressing CDR3β-'IG' and/or 'YGY' were detected across age groups, only those coexpressing 'IG' and 'YGY' were shared between children, adults and older adults, but were absent in newborns (Fig. 5d,e).

Both children and older adults expressed less-prominent CDR3β-'IY'-motifs with highest frequencies in children (Fig. 5a,e). 'IY'-expressing CDR3β motifs were associated with TRBV19 and variable TRBJ gene usage, resulting in limited sharing among age groups (Fig. 5a,d). CDR3β-'(I)F'-motif expressing clonotypes were identified in newborns, children and older adults, and were associated with TRBV19, TRBV27 and variable TRBJ segments (Fig. 5a,e). Despite strong enrichment of the 'F' residue, CDR3β-(I)F-expressing clonotypes were detected at low frequencies and were not shared between age groups, with the exception of DMC19 dominated by the highly frequent CDR3β-'F'-expressing clonotype (71.9%) (Fig. 5d,e and Supplementary Table 2).

Overall, we identified public-associated CDR3α and CDR3β motifs, with CDR3α 'GGGSQG' and CDR3β-'RS' motifs being most prominent. Public-associated CDR3α and CDR3β motifs became less frequent in older adults and were replaced with high-frequency private CDR3 motifs uniquely identified in a single individual.

**Fig. 5 | Age-related changes within the A2/M1$_{58}$$^+$CD8$^+$ CDR3αβ-motifs. a**, The top-scoring A2/M1$_{58}$$^+$CD8$^+$ CDR3α (left TCR logo) and CDR3β (right TCR logo) sequence motifs for each age group. Each logo depicts the V (left side) and J (right side) gene frequencies with the CDR3 amino acid sequence in the middle with the full height (top) and scaled (bottom) by per-residue reparative entropy to background frequencies derived from TCRs with matching gene-segment composition to highlight motif positions under selection. The middle section indicates the inferred rearrangement structure by source region (light gray for V-region, dark gray for J, black for D and red for N insertions) of the grouped receptors. **b**, Persistence of TCRα clonotypes expressing selected prominent CDR3α motifs across different age groups. Colors identify the most prominent CDR3α motifs. Shared clonotypes are connected by colored lines. **c**, Frequency of the most prominent CDR3α motifs GGGSQG, GGG and GG across the different age groups. **d**, Persistence of TCRβ clonotypes expressing selected prominent CDR3β motifs across different age groups. Colors identify the most prominent CDR3β motifs. Shared clonotypes are connected by colored lines. **e**, Frequency of the most prominent CDR3β motifs IV, RS, IG, YGY, IY and IF across the different age groups. Bars indicate the median, dots represent individual donors, with *n* = 6 newborns, *n* = 12 children, *n* = 8 adults and *n* = 10 older adults (**c,f**). Statistical analysis was performed using a two-sided Kruskal–Wallis test with Dunn's correction for multiple tests. *P* values are indicated above the graphs. N, newborn; C, children; A, adult; OA, older adult.

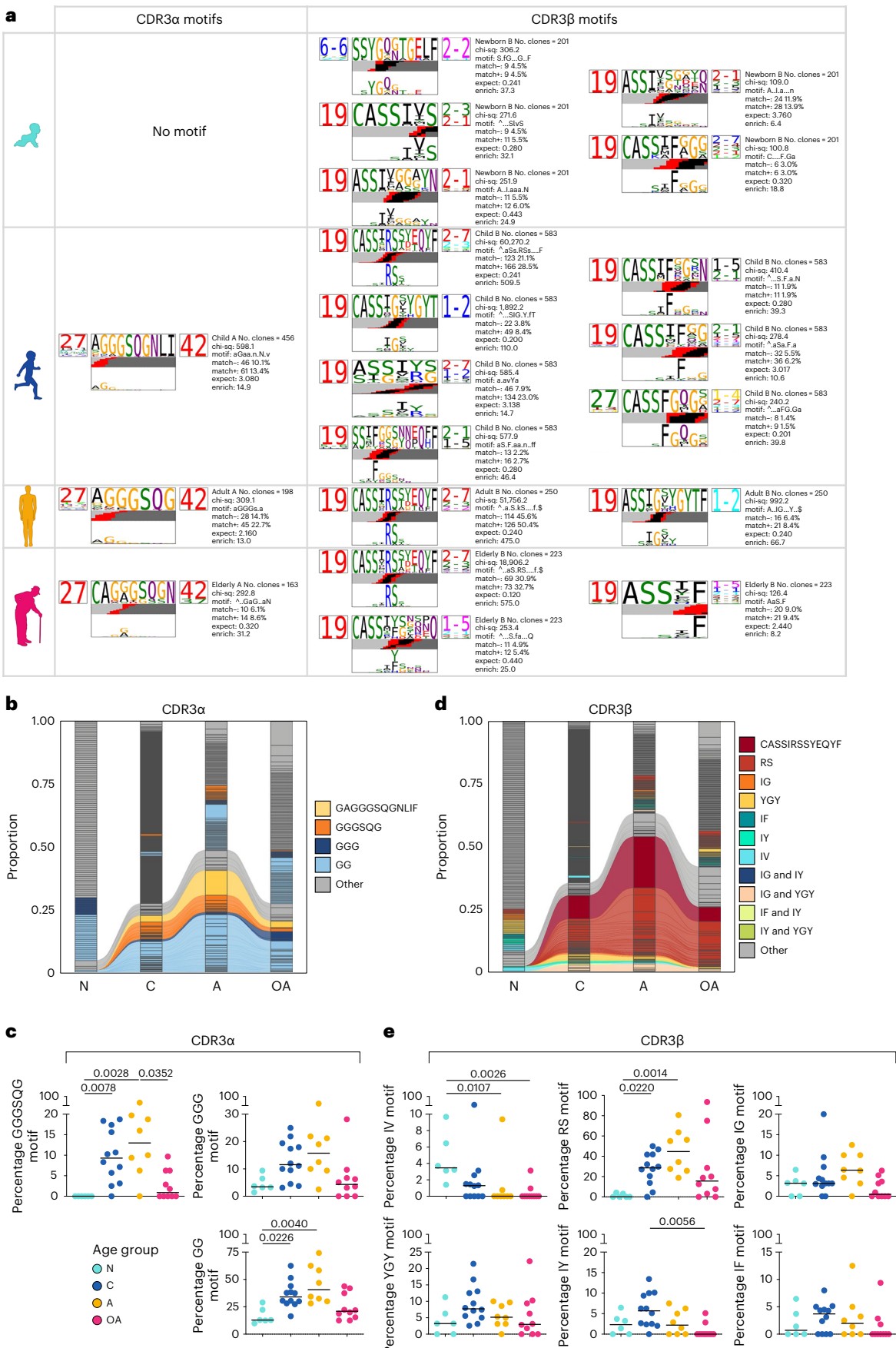

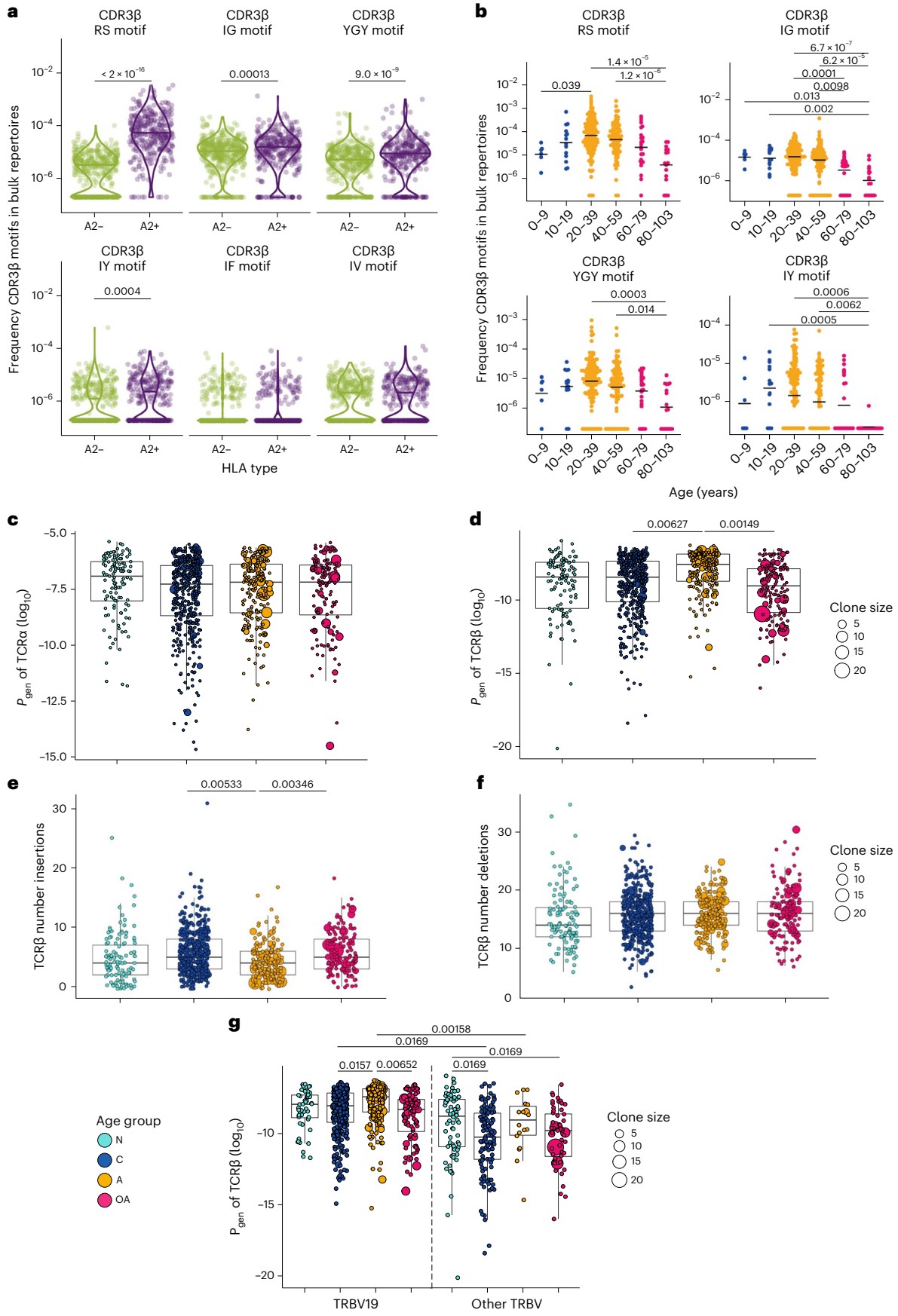

**Fig. 6 | Age-related changes in probability of generation of A2/M1$_{58}$$^+$CD8$^+$ TCRs. a**, Frequency of A2/M1$_{58}$$^+$CD8$^+$ TCRβ motifs in bulk repertoires of HLA-A*2-expressing and negative donors and **b**, across decades of human life for HLA-A*2$^+$ donors. Each dot is the cumulative frequency of TCRβ chains from A2/M1$_{58}$$^+$CD8$^+$ T cells in bulk TCRβ repertoires. Dots represent individual donors, HLA-A*2$^+$ age group 0–9, $n = 6$; 10–19, $n = 14$; 20–39, $n = 157$; 40–59, $n = 111$; 60–79, $n = 24$; 80–103, $n = 17$ and HLA-A*2$^-$ age group age group 0–9, $n = 22$; 10–19, $n = 24$; 20–39, $n = 189$; 40–59, $n = 150$; 60–79, $n = 29$; 80–103, $n = 9$. **c,d**, Probabilities of generation ($P_{gen}$; log$_{10}$ transformed) for all single TCRα (**c**) and TCRβ (**d**) chains proteins from newborns, children, adults and older adults estimated with TCRdist. **e,f**, Number of nucleotide insertions (**e**) and deletions (**f**) for all single TCRβ chains. **g**, $P_{gen}$ (log$_{10}$ transformed) for TCR β-chain proteins that include

TRBV19 (left) or other V (right) gene segments in newborns, children, adults and older adults generated with TCRdist. Box plots represent the median (middle bar), 75% quartile (upper hinge) and 25% quartile (lower hinge) with whiskers extending 1.5 × interquartile range, dots represent individual clonotypes derived from $n = 6$ newborns, $n = 12$ children, $n = 8$ adults and $n = 10$ older adults, with clone size indicated by symbol size. Statistical analysis of $P_{gen}$ and for the number of insertions and deletions between age groups utilized a two-sided mixed-effects model with donor encoded as a random effect, as described in Methods. $P$ values were adjusted ($P_{adj}$) for multiple testing with the Benjamini–Hochberg false discovery rate (FDR) method. N, newborn; C, children; A, adult; OA, older adult.

## Public datasets verify age-related changes public TCR motifs

To validate that A2/M1$_{58}$$^+$CD8$^+$ public-associated TCR CDR3β motifs decline in older adults, we utilized two independent large publicly available bulk TCRβ repertoire datasets from donors aged 0–103 years[8,47] ($n = 79$ and $n = 673$; donors with known ages were analyzed). RS, IG, YGY and IY CDR3β-motifs were enriched in HLA-A*2$^+$ donors compared to HLA-A*2-negative donors, but not IF and IV CDR3β motifs (Fig. 6a). Similar to our 'lifespan' cohort, we observed that A2/M1$_{58}$$^+$CD8$^+$ public-associated CDR3β motifs were differentially expressed across different HLA-A*2$^+$ age groups (Fig. 6b). RS, IG, YGY and IY CDR3β motifs were depleted in older adults, especially in individuals >80 years old. 'RS' motifs were less prevalent in young children (0–9 years) compared to adults (20–39 years) (Fig. 6b).

Thus, our observations of age-dependent prevalence of public-associated CDR3β motifs were confirmed in independent large cohorts of aging donors.

## Higher probability of generation underpins public TCRs

To understand why A2/M1$_{58}$$^+$CD8$^+$ public clonotypes were prominent in children and adults, and declined in older adults, we estimated the probability of generation ($P_{gen}$) of A2/M1$_{58}$$^+$CD8$^+$ TCR α-chains and β-chains using TCRdist[44]. Larger $P_{gen}$ values are associated with easier-to-generate clonotypes, whereas smaller $P_{gen}$ values indicate harder-to-generate (rarer) clonotypes.

TCR α-chain $P_{gen}$ values were similar for all ages (Fig. 6c). Significantly greater $P_{gen}$ values were observed within adult TCR β-chains compared to children and older adults (Fig. 6d), supported by fewer inferred TCRβ N insertions in adult A2/M1$_{58}$-specific TCR β-chains (Fig. 6e). The number of inferred exonuclease deletions were similar across age groups (Fig. 6f), suggesting that relatively easy-to-generate TCR β-chain formation in adults may be driven by reduced terminal deoxynucleotidyl transferase (TdT) activity during V(D)J recombination following early childhood. Indeed, TdT expression decreases between birth and adulthood[48,49], coinciding with increased public A2/M1$_{58}$$^+$CD8$^+$ cells after childhood.

Overall, adult TRBV19-expressing clonotypes had a larger $P_{gen}$ compared to children and older adults, likely driven by expanded full public clonotypes. Child and adult public-associated TRBV19-expressing clonotypes had larger $P_{gen}$ values compared

to other TRBV-expressing clonotypes (Fig. 6g). Conversely, highly prevalent non-TRBV19-expressing TCR clonotypes in children and older adults had significantly lower $P_{gen}$ values compared to newborns (Fig. 6g). Patterns for total TCR clonotypes were complemented by TCR logo analyses depicting V and J gene frequencies, CDR3 amino acid sequences and inferred rearrangement structures of grouped TCRs (Extended Data Fig. 5c,d). TCR clusters expressing public-associated CDR3α-'(G)GGSQG' or CDR3β-'RS' and -'IG' had higher probabilities of generation in children, adults and older adults (Extended Data Fig. 5c,d). TCR clusters expressing private-associated CDR3β-'(I)F' had lower probability of TCR recombination (Extended Data Fig. 5d).

Overall, child and adult A2/M1$_{58}$$^+$CD8$^+$ TCRs, often expressing TRBV19 public-associated CDR3α-'GGGSQG' and/or CDR3β-'RS' motifs, are easier to generate, explaining why they are shared between HLA-A*02:01-expressing individuals. Conversely, large clonal expansions in older adults, often associated with non-TRBV19 non-public CDR3α and/or CDR3β sequences, with increased numbers of insertions and deletions, are harder to generate and less likely to be shared.

## Public clonotypes underpin A2/M1$_{58}$$^+$CD8$^+$ T cell proliferation

To understand whether age-related changes in TCRαβs have functional consequences, we assessed proliferation linked to clonal composition of A2/M1$_{58}$$^+$CD8$^+$ T cells across age groups ($n = 3–4$ per group), following in vitro stimulation with the M1$_{58–88}$ peptide (Fig. 7, Extended Data Figs. 6 and 7, Supplementary Movie and Supplementary Table 3). Proliferation was observed by day 3–4, particularly in children and adults and associated with stronger proliferation capacity, resulting in higher A2/M1$_{58}$$^+$CD8$^+$ T cell numbers compared to older adults and newborns (Fig. 7a–c). Except older adult BP114, who had the largest fold change on day 10 despite low cell numbers, reflecting their T$_{CM}$ cell phenotype at day 0 (Fig. 7a,b and Extended Data Fig. 7a). Newborn A2/M1$_{58}$$^+$CD8$^+$ T cells had low proliferation capacity, reflecting their naive phenotype (Fig. 7b and Extended Data Fig. 7a). Fluctuation in fold change originated from high diversity in newborn TCRαβ repertoires, resulting in diverse TRBV and TRAV signatures in proliferating A2/M1$_{58}$$^+$CD8$^+$ TCR clonotypes (Extended Data Fig. 6a and Supplementary Table 3). Proliferating TCRs of other groups were dominated by TRBV19-expressing clonotypes paired with variable TRBJ, TRAV and TRAJ gene segments, resulting in TCRαβ repertoire diversity, except for DMC3 (Supplementary Table 3).

**Fig. 7 | Proliferation and polyfunctionality of A2/M1$_{58}$$^+$CD8$^+$ T cells across human life. a,b**, Total numbers (**a**) and fold increase (**b**) of A2/M1$_{58}$-tetramer$^+$CD8$^+$ T cells from day 0 (ex vivo tetramer enrichment, due to low frequency), day 3, 4, 5, 6, 7 and 10 following in vitro M1$_{58–66}$ peptide stimulation of newborn, child, adult and older adult peripheral blood mononuclear cells (PBMCs) (no previous enrichment). **c**, Representative FACS plots indicating the gating strategy used to characterize dividing (red) and undivided (blue) A2/M1$_{58}$$^+$CD8$^+$ T cells by the loss of cell trace violet over a 10-d expansion of representative donors. Gray dots represent total CD8$^+$ T cells. **d**, Persistence of TCRα clonotypes expressing selected prominent CDR3α motifs across a 10-d expansion in each age group. Shared TCRα clonotypes are connected by colored lines. **e**, Persistence of TCRβ clonotypes expressing selected prominent CDR3β

motifs across a 10-d expansion in each age group. Colors identify the most prominent CDR3β motifs identified ex vivo (Fig. 5) (**d,e**). Shared TCRβ clonotypes are connected by colored lines. **f**, Representative FACS plots indicating the gating strategy used to characterize proliferating (red) and non-proliferating (blue) A2/M1$_{58}$$^+$CD8$^+$ T cells by the loss of cell trace violet expressing IFN-γ, TNF, GrzB and perforin following M1$_{58–66}$ peptide re-stimulation on day 9. **g**, Pie charts representing average fractions of divided and undivided A2/M1$_{58}$$^+$CD8$^+$ T cells, the number of coexpressed molecules IFN-γ, TNF, GrzB and perforin (slices) and specific combination (arcs). Statistical analysis was performed using a two-sided Tukey's multiple comparisons test. Exact significant $P$ values are indicated in similar colors as the representative slice. N, newborn; C, children; A, adult; OA, older adult.

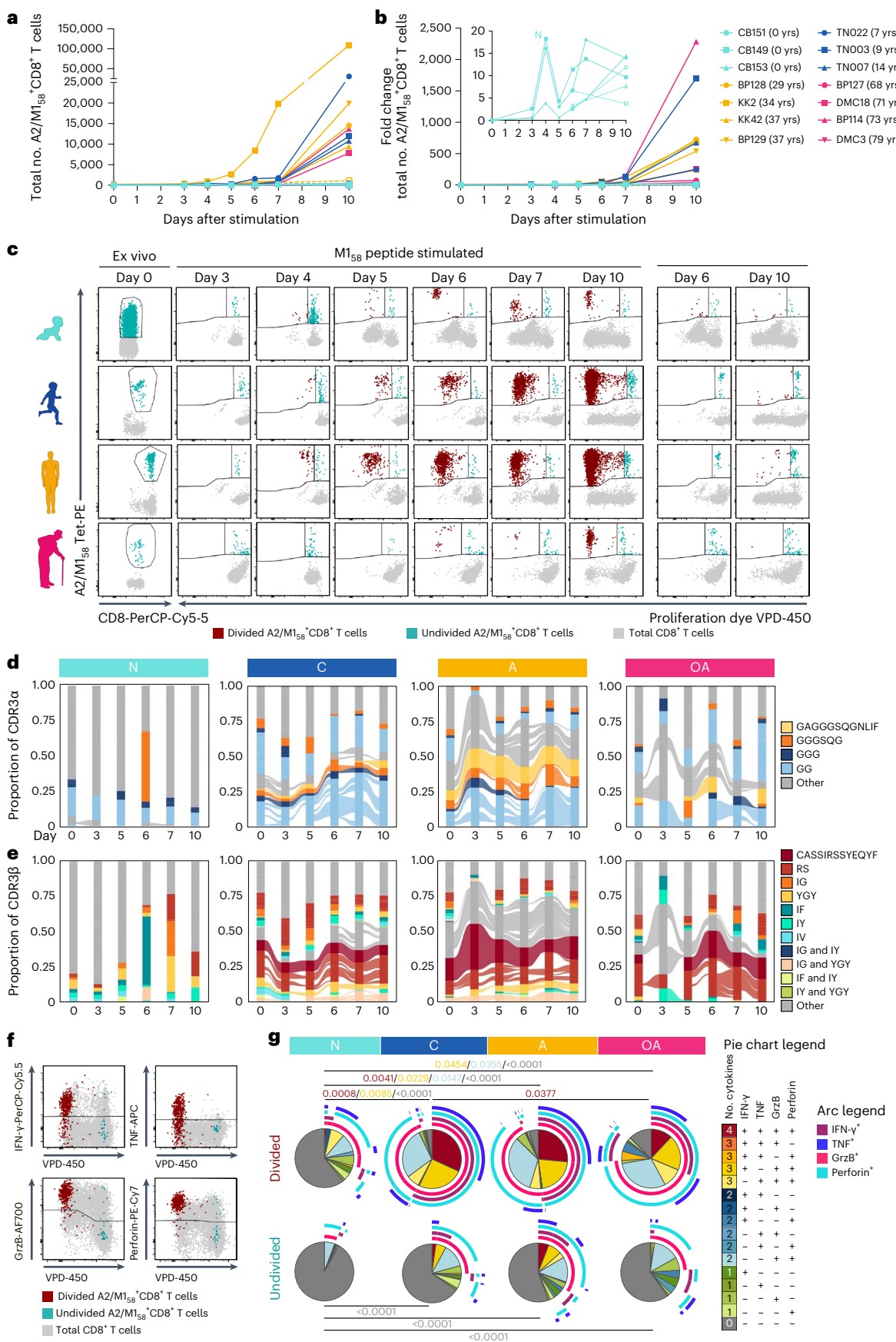

Proliferating A2/M1$_{58}$$^+$CD8$^+$CDR3 motifs reflected age-related changes (Extended Data Fig. 7b). In newborns, no CDR3α motif was identified and CDR3β motifs were scarce (d0-ASSYG; d7-FxGxD). Children and adults displayed high similarity CDR3α-'GGGSQG' and CDR3β-'RS' motifs. Only the CDR3β-'RS' motif was detected in older adults. Ex vivo-identified CDR3α motifs (full public GAGGGSQGNLIF/ GGGSQG/GGG/GG) were observed in proliferating A2/M1$_{58}$$^+$CD8$^+$ clonotypes in children and adults, at lower frequencies in children and near-absent in most older adults (Fig. 7d and Supplementary Table 3), with the exception of older adult BP114 (Supplementary Table 3). Ex vivo-identified public-associated CDR3β motifs (full public CASSIRSSYEQYF/RS/IG/YGY/IF/IY) were detected in dividing A2/M1$_{58}$$^+$CD8$^+$ T cells throughout the assay, with lower prevalence of CDR3β-CASSIRSSYEQYF and -'RS' motifs in children and more inter-donor variation in older adults (Fig. 7e, Supplementary Video and Supplementary Table 3). Irrespective of age, the highly prevalent ex vivo public clonotype (TRAV27/TRAJ42, CDR3α-CAGGGSQGNLIF, TRBV19/TRBJ2-7 and CDR3β-CASSIRSSYEQYF) was not dominant during proliferation (Extended Data Fig. 6; clone ID:p-A in Supplementary Table 3). Less prevalent ex vivo CDR3β-expressing clonotypes (Fig. 5) were among proliferating A2/M1$_{58}$$^+$CD8$^+$ T cells (Supplementary Table 3). High similarity CDR3β-'IG' motifs were identified in child and adult proliferating A2/M1$_{58}$$^+$CD8$^+$ T cells and CDR3β-'IF'-expressing proliferating clonotypes were observed across age groups.

Overall, highly prevalent clonotypes expressing public features in children and adults were associated with robust A2/M1$_{58}$$^+$CD8$^+$ T cell proliferation. These data demonstrated the importance of TCRαβ diversity, as ex vivo low-prevalent TCRs also proliferated upon in vitro stimulation. Further studies are needed to demonstrate whether broad-spectrum TCR proliferation is also observed following influenza infection.

## A2/M1$_{58}$$^+$CD8$^+$ T cell polyfunctionality peaks in children

We investigated polyfunctionality of proliferating A2/M1$_{58}$$^+$CD8$^+$ T cells following re-stimulation with M1$_{58-66}$ peptide on day 9 (Fig. 7f,g). Proliferating polyfunctional A2/M1$_{58}$$^+$CD8$^+$ T cells, expressing interferon (IFN)-γ, tumor necrosis factor (TNF), granzyme B (GrzB) and perforin, decreased with age (children 37.7%; adults 24.8%; older adults 11.3%). IFN-γ/TNF production decreased with age, from children (IFN-γ 59.4%; TNF 46.7%) to adults (IFN-γ 50.9%; TNF 30.6%) and older adults (IFN-γ 29.7%; TNF 22%). Expression of GrzB and perforin remained stable from children (GrzB 82.23%; perforin 81.18%) to adults (GrzB 87.75%; perforin 85.31%) but decreased in older adults (GrzB 68.4%; perforin 69.61%). Newborn proliferating A2/M1$_{58}$$^+$CD8$^+$ T cells were least functional, with 74.2% expressing no functional markers, followed by older adults (12.06%), adults (3.44%) and children (2.26%). Non-dividing cells were minimally polyfunctional (Fig. 7g).

Overall, children's proliferating A2/M1$_{58}$$^+$ CD8$^+$ T cells had the highest polyfunctionality whereas newborns had the lowest polyfunctionality followed by older adults, in line with their lower cytotoxic gene expression profiles ex vivo.

## CD8 co-receptor enhances avidity of young TCRs

To demonstrate that age-related functional changes of A2/M1$_{58}$$^+$CD8$^+$ T cells resulted from differential TCRαβ repertoires across age groups, we generated HEK293T-transient and SKW-3-stable cell lines encompassing full public or prominent age-specific TCRs identified ex vivo (C1, A1, OA1, OA2a, OA2b and OA3) or proliferating in vitro (pN1, pA2 and pOA4) (Fig. 8a and Supplementary Tables 2 and 3). pN1 clonally expanded on day 7 of proliferation (newborn CB151 28 of 39 TCRs) and expressed child/adult-associated CDR3β-'IG' motif. Age-specific private child C1, adult A1 and older adult OA1 TCRs were more prevalent (TN010, 4 of 75; KK2, 5 of 70; DMC18, 6 of 105) compared to the full public TCRαβ within the donor's ex vivo TCR repertoire, without prominent public CDR3α/β features. Low ex vivo frequency adult TCR pA2 (BP128; 1 of 32 TCRs), expressing the public-associated CDR3α-'GG' motif but no public TCRβ features, proliferated early in vitro. Older adult TCRs OA2a and OA2b originated from a prominent ex vivo clonotype (DMC19; 23 of 32 TCRs), encompassing a single 'F'-expressing CDR3 β-chain and double TCR α-chains. Two older adult CDR3β-'IF'-expressing TCRs included the ex vivo low-prevalent OA3 (BP127; 1 of 32 TCRs) and highly proliferating pOA4 (DMC18; 12 of 52 TCRs day 10) (Fig. 8a).

Transiently expressed TCRs together with CD3 complex units in HEK293T cells revealed that all TCRs, except OA2b, expressed CD3 on cell surface, indicative of productive TCRs. A1, pA2 and pOA4 had similar A2/M1$_{58}$ tetramer-binding avidities compared to public TCR, whereas pN1, C1 and OA2a displayed lower A2/M1$_{58}$-binding avidities. Older adult TCRs OA1 and OA3 displayed no A2/M1$_{58}$-specific binding, despite CD3 expression, suggesting potential CD8αβ co-receptor requirement to overcome the binding threshold (Extended Data Fig. 8a,b).

To understand whether CD8 co-receptors enhance A2/M1$_{58}$ avidity, we compared A2/M1$_{58}$ tetramer staining between stable SKW-3 cell lines expressing productive TCRs (except OA2b) and CD3 (SKW-3-CD3$^+$) or CD3 and CD8 (SKW-3-CD3$^+$CD8$^+$) (Fig. 8b–d and Extended Data Fig. 8c). CD8 co-receptors improved A2/M1$_{58}$-binding avidity of the public, pN1, C1, A1, OA2a and pOA4, but not pA2, OA1 and OA3. Overall, older adult OA1, OA2a and OA3 TCRs showed minimal A2/M1$_{58}$ binding avidity irrespectively of CD8 (Fig. 8b–d). To verify, cell lines were stained with wild-type (WT) A2/M1$_{58}$ tetramer (A2/M1$_{58}$-WT), A2/M1$_{58}$ tetramer with a knockout (KO) CD8-binding site (A2/M1$_{58}$-KO) and A2/M1$_{58}$ tetramer with an enhanced CD8-binding site (A2/M1$_{58}$-Enh) (Fig. 8e,f and Extended Data Fig. 8c,d). CD8-binding site KO reduced A2/M1$_{58}$-binding avidity of A1 and pOA4. Binding was lost for pN1 and C1. Conversely, CD8-binding site enhancement improved binding avidity of pN1, C1 and OA2a. No changes were observed for other TCRs (Fig. 8f) or in SKW-3-CD3$^+$ TCR-expressing cell lines (Extended Data Fig. 8c,d).

Overall, CD8 co-receptors enhanced binding avidity of age-specific private TCRs representing newborn, child, one adult and one older TCR, but had less impact on weak-binding older TCRs.

---

**Fig. 8 | SKW-3-CD3$^+$ and SKW-CD3$^+$CD8$^+$ cell lines expressing age-specific TCRs. a**, Selection of A2/M1$_{58}$$^+$CD8$^+$ TCR specifically identified in certain age groups. p, proliferation; aa, amino acid. **b,c**, Representative A2/M1$_{58}$ tetramer-PE staining of SKW-3-CD3$^+$ (**b**) and SKW-3-CD3$^+$CD8$^+$ (**c**) TCR-expressing cell lines. **d**, Median MFI A2/M1$_{58}$ tetramer-PE of SKW-3-CD3$^+$ (open bars) and SKW-3-CD3$^+$CD8$^+$ TCR-expressing cell lines (closed bars) ($n = 5$ independent experiments, median and interquartile range (IQR)); dotted line indicates MFI threshold set by the parental cell line expressing no TCR. MFI, median fluorescence intensity. **e,f**, Representative staining (**e**) and median MFI (**f**) of A2/M1$_{58}$ tetramer-PE staining with a normal CD8-binding site (A2/M1$_{58}$-WT, lightly shaded), knockout CD8-binding site (A2/M1$_{58}$-KO open) and enhanced CD8-binding site (A2/M1$_{58}$-Enh, closed) tetramer of SKW-3-CD3$^+$CD8$^+$ TCR-expressing cell lines ($n = 2$ independent experiments); dotted line indicates MFI threshold set by the parental cell line expressing no TCR. **g**, Representative CD69-PECy7 expression following peptide titration for public and age group specific TCRs expressed in SKW-3-CD3$^+$ cell lines. **h**, Percentage of maximum CD69-PECy7 MFI for age-specific TCRs expressed on SKW-3-CD3$^+$ (open circles) and SKW-3-CD3$^+$CD8$^+$ (closed circles) cell lines following peptide titration; EC$_{50}$ indicated by the dotted line ($n = 3$ independent experiments, median and range). **i**, Representative CD69-PECy7 expression profiles of SKW-3-CD3$^+$ TCR-expressing cell lines following stimulation with the M1$_{58-66}$ peptide with single alanine substitutions, except for binding site (p2). **j**, Percentage of maximum CD69-PECy7 MFI following stimulation with M1$_{58-66}$ alanine scan peptides for age-specific TCRs expressed on SKW-3-CD3$^+$ cells ($n = 3$ independent experiments, median and range). Statistical analysis was performed using a two-sided Kruskal–Wallis with Dunn's correction for multiple tests between the public TCR and the age-specific TCRs within the same cell line (**d,j**, black) or between the same TCRs across two cell lines (**d**, blue). $P$ values are indicated above the graphs.

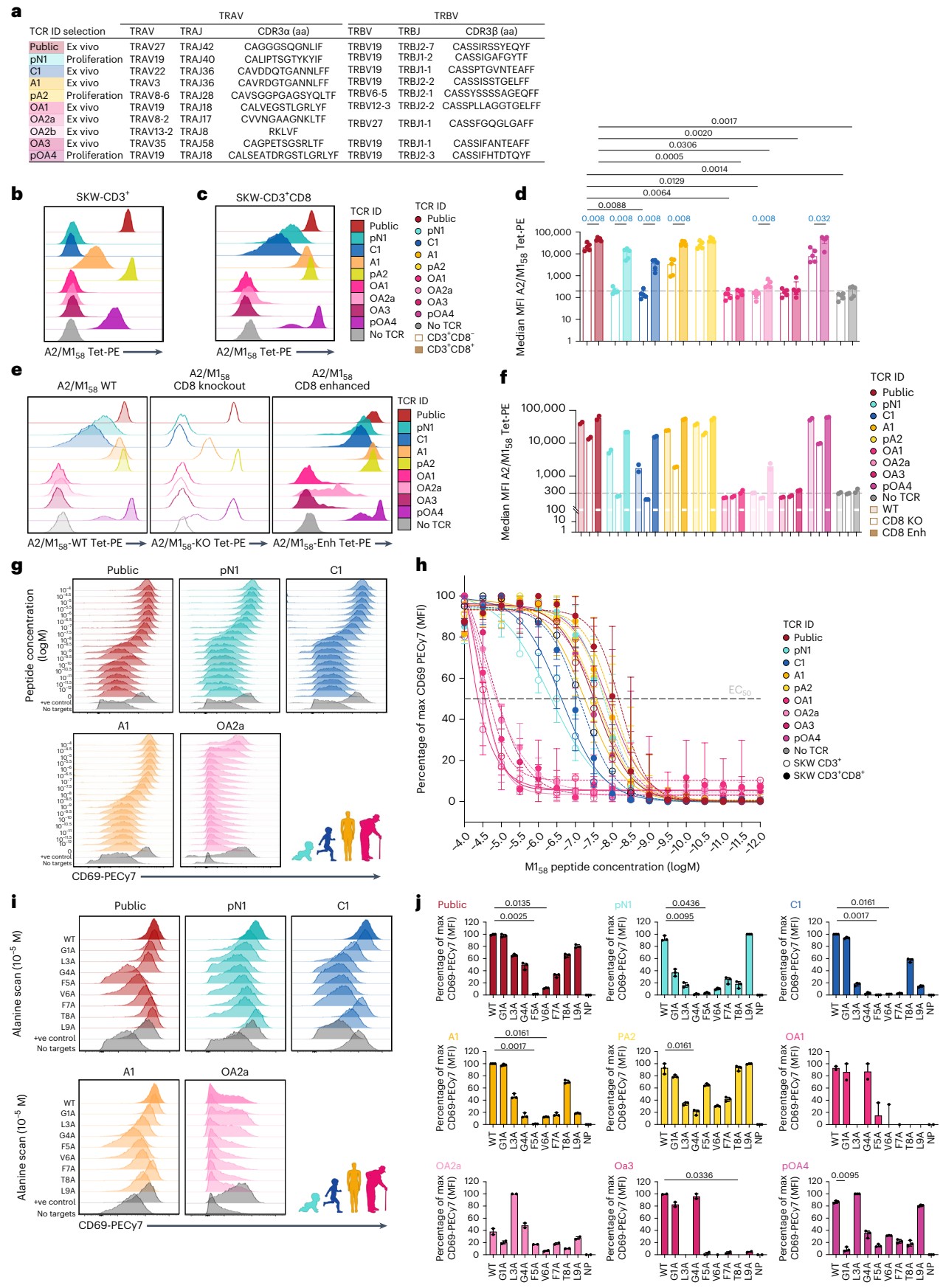

## Older A2/M1$_{58}$$^+$CD8$^+$ TCRs display reduced activation capacity

To determine whether differential A2/M1$_{58}$ avidity of age-specific TCRs affected their activation, we performed peptide M1$_{58-66}$ titrations across nine TCRs in SKW-3-CD3$^+$ and SKW-3-CD3$^+$CD8$^+$ cell lines. TCR activation was measured by CD69 expression and saturation curves provided half maximal effective concentrations (EC$_{50}$), reflecting antigen sensitivity (Fig. 8g,h and Extended Data Fig. 8e,f). Activation thresholds of A1, pA2 and pOA4 TCRs were like public TCRs in SKW-CD3$^+$ cells and SKW-3-CD3$^+$CD8$^+$ cells (<0.6 logM difference in EC$_{50}$). Higher activation thresholds for pN1 and C1 compared to the public TCR in SKW-CD3$^+$ cells were partly overcome in SKW-3-CD3$^+$CD8$^+$ cells for pN1 but not C1. Older TCRs OA1, OA2a and OA3 had the highest activation threshold, with 2.943–3.14 logM higher EC$_{50}$ values compared to the public TCR in SKW-CD3$^+$ cells and was not rescued by CD8 (Fig. 8h and Extended Data Fig. 8f). Activation at high peptide concentrations reassured that older TCRs were A2/M1$_{58}$-specific; however, their binding affinity was too low for detection through tetramer staining (Fig. 8d).

Overall, ex vivo A2/M1$_{58}$-specific TCRs from older adults had reduced activation capacity compared to public and other age-specific TCRs. The CD8 co-receptor decreased the public, newborn (pN1) and adult (A1) TCR activation threshold, potentially resulting from improved binding avidity (Fig. 8b–f).

## Reduced recognition of M1$_{58}$ variants by age-specific TCRs

We stimulated age-specific or public TCR-expressing SKW-3-CD3$^+$ (Fig. 8i,j) and SKW-3-CD3$^+$CD8$^+$ cell lines (Extended Data Fig. 9a,b) with WT and mutated M1$_{58-66}$ peptides containing single alanine substitutions at sequential positions, excluding the anchor residue (position (p) 2).

In accordance with reports[26,38,50], M1$_{58-66}$ p5 was critical for public TCR recognition in SKW-3-CD3$^+$ and SKW-3-CD3$^+$CD8$^+$ cells. Additional loss in TCR recognition was observed for p6 (V6A) and p7 (F7A), and partial loss for L3A, G4A and T8A in SKW-3-CD3$^+$ cells, which were overcome by CD8 coexpression (Fig. 8j and Extended Data Fig. 9b).

Adult TCR A1 displayed a public-like TCR recognition profile, with p5 critical for TCR recognition and reduced recognition at p3, p4, p6, p7, p8 and p9 (Fig. 8i,j); however, CD8 only improved TCR recognition of M1$_{58-66}$ T8A (Extended Data Fig. 9). Newborn pN1 had two critical positions, p4 and p5, whereas CD8 coexpression partly overcame lower TCR recognition at p1, p3, p6, p7 and p8 (Fig. 8i,j and Extended Data Fig. 9). Child TCR C1 had 4 critical positions, p4, p5, p6 and p7, substantial reduction for p3 and p9, and partial loss for p8 (Fig. 8i,j). The CD8 co-receptor only improved recognition of M1$_{58-66}$ T8A (Extended Data Fig. 9). The adult pA2 clonotype strongly proliferated early despite the presence of the full public TCR. No critical TCR recognition positions were identified but decreases were observed for p3, p4, p6 and p7 and partial loss for p5 (Fig. 8j), which was improved by CD8 coexpression (Extended Data Fig. 9). Older TCRs had notably unique recognition patterns, with multiple critical binding sites (OA1: p3, p7, p8 and p9; OA2a: p6 and p8; OA3: p3, p5, p6 and p8) and substantially reduced binding sites (OA1: p5; OA2a: p1, p4, p5, p7 and p9; OA3: p7 and p9) (Fig. 8j). pOA4 had a single critical binding site (p1) and substantial reduction for p4, p5, p6, p7 and p8 (Fig. 8j). Generally, the CD8 co-receptor did not improve TCR recognition (Extended Data Fig. 9). Of note, the alanine substitution at p3 (L3A) in older TCR OA2a increased TCR recognition, suggesting that this TCR might have binding properties for another peptide.

Overall, age-specific TCRs display reduced ability to recognize M1$_{58-66}$ peptide variants. Ex vivo older private TCRs displayed unique A2/M1$_{58}$ binding profiles, underpinning their reduced binding capacity, avidity, functionality and proliferating capacity compared to other prominent age-specific TCRs, especially TCRs with public-associated features found at high frequency in children and adults.

## Discussion

We linked age-specific single-cell molecular gene profiles with phenotypes, functionality and paired single-cell TCRαβ repertoires of influenza-specific HLA-A*02:01/M1$_{58-66}$-specific CD8$^+$ T cells. Unexpectedly, older A2/M1$_{58}$$^+$CD8$^+$ T cells did not reach terminally differentiated or exhausted end points. Instead, reduced functionality was associated with loss of highly functional public TCRαβ clonotypes dominating younger TCRαβ repertoires. Conversely, large clonal expansions of less-functional private TCRαβs dominated older TCRαβ repertoires. Age-specific transcriptomes supported a linear differentiation trajectory from newborns to children, then adults, whereas suboptimal clonal resets in older adults were associated with newborn/child-like molecular signatures.

Older A2/M1$_{58}$$^+$CD8$^+$ T cells lacked immunosenescent hallmarks, including terminally differentiated T$_{EMRA}$ cells and CD57 expression, matching gene expression profiles. High CD8$^+$ T$_{EMRA}$ frequencies are linked to chronic EBV or CMV infections[18,51,52]. Instead, older adults maintained robust A2/M1$_{58}$$^+$CD8$^+$ T$_{naive}$ and T$_{CM}$ phenotypes, suggesting that acute viral infections, such as influenza, do not trigger terminal differentiation.

Age-specific A2/M1$_{58}$$^+$CD8$^+$ T cells transcriptomes corroborated age-specific phenotypic profiles, including mixed naive/memory phenotypes in children and older adults. Newborn naive T cells uniquely expressed *TLB*. The role of *TLB*, encoding TNF-C, remains ill defined. Cytotoxic and *TRAV27*/*TRBV19* genes dominated child and adult A2/M1$_{58}$-specific transcriptomes and their public TCR features resulted in higher proliferative capacity and polyfunctionality, compared to newborns and older adults. Older A2/M1$_{58}$-specific transcriptomes, dominated by *CXCR4* (ref. 39), *KLF2*, *SELL*, *TXNIP*, *PIK3IP1*, *CD37* and *TRBV27*, displayed less-differentiated cell states, lacked exhaustion genes, expressed AP-1 transcription factors *FOS* and *JUN* (progenitor of exhaustion in acute infection[53]), *C-JUN* (resistance to exhaustion[43]) and distinct clonal lineages. Trajectory analysis supported lack of a terminally differentiated end stage and suboptimal clonal reset in older adults. Suboptimal clonotypes expressing older private features were detected at a low frequency in younger age groups.

Older suboptimal A2/M1$_{58}$-specific clonotypes underpinned differences in gene expression and functionality. The public TCRαβ clonotype was first identified in children, peaked in adults and decreased in older adults, coinciding with large private TCRαβ expansions. The relatively featureless A2/M1$_{58}$ structure requires a specific peg-notch mode of interaction by CDR3β-'RS' in the public TRAV27/TRBV19 clonotype without peptide-specific CDR3α binding[26,27,38,44]. Glycine repeats allow the required conformational changes[50]. Fewer CDR3α-glycines reduces flexibility in older 'RS'-expressing TCRs, potentially hampering A2/M1$_{58}$ binding. Alternatively, CDR3α amino acids with bulkier side chains might hamper the peg-notch recognition[26,27,38]. Low-frequency public-associated A2/M1$_{58}$-specific CDR3β motifs, including 'xGxY' and 'F', were described[27]; however, gradual decline of public-associated CDR3β-RS/-IG/-YGY/-IY/-IF/-IV motifs across the human lifespan was not reported. Our study supports that alternative TCR structures recognize the featureless A2/M1$_{58}$-complex[27,46].

During infections, 'best-fit' high-avidity clonotypes are selected from naive TCR repertoires and expand following subsequent encounters[54,55]. We demonstrate that 'best-fit' high-avidity public clonotypes peak in adults and are gradually replaced by low-avidity clonotypes in older adults. TRBV19/CDR3β-RS-expressing clonotypes dominate over TRBV19-expressing clonotypes with other public CDR3β-associated features, possibly because CDR3β-RS only requires two amino acids to bind the A2/M1$_{58}$ complex regardless of the TCR α-chain[27,38]; however, TCR repertoire diversity remains important to protect against escape variants[5,27]. How this delicate balance between expanded best-fit clonotypes and TCR diversity is maintained following repeated infections remains unexplored. Children and adults maintain diverse TCR repertoires during in vitro expansion. Reduced public TCRαβ clonotypes and TCRαβ diversity within older TCR repertoires explains why older adults, in the absence of pre-existing antibodies, are at higher risk of severe disease during

influenza epidemics and pandemics. Conversely, highly functional and diverse public TCRαβ repertoires in children clarifies their relative superiority in fighting influenza infections. Understanding how we can preserve this delicate balance between expansion of 'best-fit' TCRs while maintaining TCR diversity may be the Holy Grail in defining how we can maintain optimal immunity across the human lifespan through vaccination and/or immunotherapies.

## Online content

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

[1]Department of Microbiology and Immunology, University of Melbourne at the Peter Doherty Institute for Infection and Immunity, Melbourne, Victoria, Australia. [2]Department of Hematopoiesis, Sanquin Research and Landsteiner Laboratory, Amsterdam UMC, University of Amsterdam, Amsterdam, The Netherlands. [3]Department of Immunology, St. Jude Children's Research Hospital, Memphis, TN, USA. [4]School of Medical Sciences and The Kirby Institute, UNSW Sydney, Sydney, New South Wales, Australia. [5]Viral and Structural Immunology Laboratory, Department of Biochemistry and Chemistry, La Trobe Institute for Molecular Science, La Trobe University, Bundoora, Victoria, Australia. [6]Immunity Program and Department of Biochemistry and Molecular Biology, Biomedicine Discovery Institute, Monash University, Clayton, Victoria, Australia. [7]School of Health Sciences and School of Medicine, University of Tasmania, Launceston, Tasmania, Australia. [8]Victorian Infectious Diseases Reference Laboratory, The Royal Melbourne Hospital at The Peter Doherty Institute for Infection and Immunity, Melbourne, Victoria, Australia. [9]Obstetrics, Nutrition and Endocrinology Group, Department of Obstetrics and Gynaecology, University of Melbourne, Melbourne, Victoria, Australia. [10]Deepdene Surgery, Deepdene, Victoria, Australia. [11]Institute of Infection and Immunity, Cardiff University School of Medicine, Cardiff, UK. [12]School of Health and Biomedical Science, RMIT University, Melbourne, Victoria, Australia. [13]Tasmanian Vaccine Trial Centre, Clifford Craig Foundation, Launceston General Hospital, Launceston, Tasmania, Australia. [14]These authors contributed equally: Fabio Luciani, Katherine Kedzierska. ✉e-mail: kkedz@unimelb.edu.au

## Methods

### Study participants and ethics

Donors were selected from a large HLA-typed randomly recruited lifespan cohort $n \geq 500$, consisting of 154 newborns, 30 children, 360 adults/older adults, based on their expression of HLA-A*02:01. Overall, 11 HLA-A*02:01+ newborns (0 years), 12 HLA-A*02:01+ children (median 9 years, range 3–16), 20 HLA-A*02:01+ adults (median 37 years, range 18–58) and 18 HLA-A*02:01+ older adults (median 72 years, range 63–88) were included in the study (Supplementary Table 1). Adults and older adults were recruited via the University of Melbourne (UoM), Deepdene Medical Clinic and the Australian Red Cross Lifeblood. Children were recruited via the Launceston General Hospital and St Jude Children's Research Hospital. Umbilical cord blood, reflecting newborn's blood, was obtained via Mercy Hospital for Women. Peripheral blood was collected before the COVID-19 pandemic. All participants or their guardians provided informed written consent. Participants of the study did not receive any compensation. PBMCs were isolated using Ficoll-Paque (GE HealthCare) gradient centrifugation, and then cryopreserved in liquid nitrogen until required. HLA class I and II molecular genotyping was performed from genomic DNA by the Australian Red Cross Lifeblood. CMV status was determined as described previously[56]. Experiments conformed to the Declaration of Helsinki Principles and the Australian National Health and Medical Research Council Code of Practice. The study was approved by the Human Research Ethics Committee of the UoM (ethics IDs 24567, 13344 and 23852), Australian Red Cross Lifeblood (ID 2015 8), St Jude Children's Research Hospital (XPD12-089 IIBANK), Mercy Hospital for Women (R14-25) and Tasmanian Health and Medical Human Research Ethics Committee (ID H0017479).

### Peptides and tetramers

WT $M1_{58-66}$ peptide (GILGFVFTL) and its single amino acid alanine substitution variants (AIGFVFTL, GIAGFVFTL, GILAFVFTL, GILGAVFTL, GILGFAFTL, GILGFVATL, GILGFVFAL and GILGFVFTA) were purchased from GeneScript. HLA-A*02:01-$M1_{58}$ (GILGFVFTL) monomers: WT, KO CD8-binding site (D227K; CD8 KO) and enhanced CD8-binding site (Q115E; CD8 Enh) were generated by refolding each peptide with its restricted HLA α-heavy chain-BirA and β2-microglobulin[57–59] before 8:1 conjugation with PE-streptavidin (BD Biosciences) to generate A2/$M1_{58}$ tetramers.

### Ex vivo A2/$M1_{58}$ tetramer enrichment

PBMCs ($1-5 \times 10^7$) were thawed in complete RPMI (cRPMI) medium (RPMI1640 medium (Invitrogen) supplemented with 2 mM L-glutamine (Gibco), 1 mM MEM sodium pyruvate (Gibco), 100 μM MEM non-essential amino acids (Gibco), 5 mM HEPES buffer solution (Gibco), 55 μM 2-mercaptoethanol (Gibco), 100 U ml⁻¹ penicillin (Gibco), 100 μg ml⁻¹ streptomycin (Gibco) and 10% fetal bovine serum (Gibco)) supplemented with 50 U ml⁻¹ Benzonase (Novagen Merck) before undergoing TAME as previously described[6,13]. Briefly, cells were resuspended in MACS buffer (PBS plus 0.5% BSA and 2 mM EDTA) and magnetically enriched with PE-streptavidin conjugated A2/$M1_{58}$ (GILGFVFTL) tetramer using anti-PE Microbeads (Miltenyi Biotec) and passed through an LS column (Miltenyi Biotec) to enrich A2/$M1_{58}$ tetramer-positive cells. Cells were then surfaced stained in MACS buffer using anti-CD71-BV421 (1:50 dilution, BD Biosciences 562995), anti-CD3-BV510 (1:200 dilution, BioLegend 317332), anti-HLA-DR-BV605 (1:100 dilution, BioLegend 307640), anti-CD4-BV650 (1:100 dilution, BD Biosciences 563875), anti-CD27-BV711 (1:200 dilution, BD Horizon 563167), anti-CD38-BV785 (1:100 dilution, BD Biosciences 563964), anti-CD57-APC (1:400 dilution, BD Biosciences 560845), anti-CCR7-AF700 (1:50 dilution, BD Biosciences 561143), anti-CD14-APC-Cy7 (1:100 dilution, BD Biosciences 560180), anti-CD19 (1:100 dilution, BD Biosciences 560177), anti-CD45RA-FITC (1:200 dilution, BD Biosciences 555488), anti-CD8-PerCP-Cy5.5 (1:200 dilution, BD Biosciences 565310), anti-CD95-PECF594 (1:100 dilution, BD Horizon 562395), anti-PD-1-1-PE-Cy7 (1:50 dilution, BD Biosciences 561272) and Live/Dead fixable aqua dead-cell stain (1:800 dilution, Invitrogen L10119), fixed with 1% paraformaldehyde (PFA) (ProSciTec) for acquiring on an LSR Fortessa II (BD Biosciences) or resuspended in MACS buffer for single cell-(index)-sorting using a BD FACSAria III (BD Biosciences), followed by the analysis using FlowJo software (v.10.8.1) (BD Biosciences). A2/$M1_{58}$+CD8+ T cell frequencies were calculated relative to the total CD8+T cell numbers in an unenriched fraction as described previously[6,13,60]. Samples with <10 A2/$M1_{58}$+CD8+ T cells counted were excluded for phenotypic analyses.

### Single-cell RNA sequencing

Three HLA-A*02:01-expressing donors from each age group (Supplementary Fig. 2a) were selected for scRNA-seq analysis. A2/$M1_{58}$+CD8+ T cells were TAME-enriched and then individually (index-) sorted into chilled 96-well twin.tec PCR plates (Eppendorf) containing lysis buffer (1 μl RNase inhibitor and 19 μl Triton X-100) after TAME and on a BD Aria III sorter. Libraries were generated as described previously[61,62]. A Nextera XT DNA Library Prep kit was used for the generation of sequencing libraries and sequencing performed on a NextSeq500 platform with 150-bp high-output paired-end chemistry for 901 A2/$M1_{58}$-tetramer+CD8+ T cells/donor and 101 controls.

Raw sequencing reads were trimmed using Trimmomatic (v.0.39) (ref. [63]) and aligned to the human reference genome (GRCh38.89) using TopHat (v.2.1.1). Gene expression FPKM (fragments per kilobase of transcript per million mapped reads) values were quantified using Cufflinks (v.2.2.1) (ref. [64]). TCR sequences were reconstructed using VDJPuzzle[65,66].

### scRNA-seq quality control, normalization and batch correction

Downstream analysis was performed in R using packages downloaded from Bioconductor v.3.10. Cells were removed from each batch if they did not meet these criteria: less than 40% reads aligned to mitochondrial genes, number of detected genes more than 400. The bulk samples and genes expressed in zero cells were removed. Normalization was performed using the NormalizeData function from Seurat (v.4.1.0). To minimize potential batch effects, the experiment was designed to distribute donors from each age group to be included across separate experiments. In addition, donors were distributed over across plates within each experiment. Batch effect was tested using the FindIntegrationAnchors and IntegrateData functions from Seurat[67]. For scRNA-seq bioinformatics analysis no correction for batch effect was performed because of poor evidence of bias and also for sufficient mix of cells among clusters based on experimental design (plates and experiment number). The observed segregation based on age group and phenotype was an expected biological bias given the experimental design.

### Analysis of scRNA-seq data

Dimensionality reduction and clustering were also performed in Seurat using the normalized matrix. PCA was performed on the normalized data using the 3,000 most variable genes. Clustering was performed using the shared nearest neighbor modularity optimization-based clustering algorithm (FindClusters(resolution = 0.8, algorithm = 'louvain')) as implemented in Seurat.

Differential gene expression was performed using MAST (v.1.16.0) implemented in the function FindMarkers in Seurat v.4 and using two-sided P values to report the results REF_SEURAT[67]. Notably, the test used was MAST and batch was used as a latent variable with a log(FC) threshold of 0.3. Other parameters were kept as default. Signature scores were computed from the normalized single cell transcriptomic matrix as the average log(FPKM + 1) of all genes in the signature. Differential expression output across all the analyses is reported in Supplementary Table 4 (UMAP clusters) and Supplementary Table 5 (age groups).

## Gene set enrichment analysis

GSEA was performed using the R package fsgea (v.1.16.0). Normalized enrichment scores were assessed using the fgsea(…, maxSize = 500, nperm = 10,000) function across the curated Molecular Signatures Database (MSigDB) Hallmark, C2 curated gene sets consisting of canonical gene sets PID). Customized gene signatures for T cell phenotypes are reported in Supplementary Table 6, which were prepared by manually curating published data.

## Trajectory inference

PAGA analysis was performed through Scanpy (v.1.7.1) (ref. [68]) with parameters as recommended[22]. The FPKM matrix following normalization (with Seurat), along with the top 20 PCAs (previously generated with Seurat) were used and visualization was performed using sc.pp. neighbors(n_neighbors = 8, n_pcs = 20) and a coarse-grained and simplified graph using sc.tl.paga. Clusters were calculated using sc.tl. louvain(resolution = 0.8) and visualization was performed using sc.pl. paga and sc.pl.draw_graph. Pseudotime analysis was performed using the diffusion map algorithm (sc.tl.dpt) by manually assigning an initial iroot value. Scaled pseudotime were used with Loess smoothing and were calculated as uniformly distributed mapping of the diffusion pseudotime values to preserve the cell order and account for heterogeneous distribution of gaps between pseudotime values. The growth rate with which T cell phenotypic/age group subsets change along the inferred pseudotime trajectories were calculated as the ratio between the difference in cell numbers and the scaled pseudotime values over a window of size 0.05. These values were then plotted using the geom_smooth R function with default parameters.

## Single-cell RT–PCR and paired TCRαβ sequencing

A2/M1$_{58}$$^+$CD8$^+$ T cells were TAME-enriched and subsequently individually (index-) sorted into chilled 96-well twin.tec PCR plates (Eppendorf) and immediately stored at −80 °C until required. Single-cell paired CDR3α and CDR3β regions were analyzed by multiplex-nested PCR with reverse transcription and followed by sequencing of the CDRα and CDRβ products, essentially as described previously[6,26,69], except for using double amounts of reaction mix in the complementary DNA step for older adult samples. Sequences were analyzed with FinchTV. V-J regions were identified by IMGT query (www.imgt.org/IMGT_vquest). TCR sequences were parsed using the TCRdist analytical pipeline[44]. Clonotypes were defined as single-cell TCRαβ pairs that exhibit the same V, J and CDR3 regions.

## A2/M1$_{58}$-specific TCR motifs in publicly available datasets

A2/M1$_{58}$-specific TCRβ sequences identified in our study were further verified in two publicly available bulk TCRβ datasets from independent cohorts of healthy individuals[8,47]. TCR was considered matched when it had the same CDR3β and genomic V segment. For all the donors in the cohorts, we calculated the total frequency of matched TCRs within each CDR3β-motif. As conventional HLA typing was not available for all donors in the independent cohorts, we divided them into HLA-A*2$^+$ (n = 329; age group 0–9, n = 6; 10–19, n = 14; 20–39, n = 157; 40–59, n = 111; 60–79, n = 24; and 80–103, n = 17) and HLA-A*2$^-$ (n = 423; age group 0–9, n = 22; 10–19, n = 24; 20–39, n = 189; 40–59, n = 150; 60–79, n = 29; and 80–103, n = 9) using TCR repertoire based HLA typing procedure as previously described[70].

## A2/M1$_{58}$$^+$CD8$^+$ T cell proliferation assay

PBMCs from HLA-A*02:01$^+$ donors (-21 × 10$^6$) were pre-incubated with cell trace violet (Violet Proliferation Dye 450, BD Horizon) according to the manufacturer's instructions before generating A2/M1$_{58}$-specific CD8$^+$ T cell lines[6]. Briefly, one-third of the labeled PBMCs were pulsed with 10 µM M1$_{58–66}$ peptide (GILGFVFTL), or DMSO as unstimulated control, for 60 min at 37 °C, washed twice with RPMI and incubated with the remaining two-thirds of the non-peptide-pulsed autologous

PBMCs in cRPMI at a final concentration of non-pulsed cells at 1 × 10$^6$ per well for each day of the proliferation assay. Cells were cultured in a 48-wells plate for 10 d at 37 °C and 5% CO$_2$. Cultures were supplemented on day 4 with 20 U ml$^{-1}$ rIL-2 (Roche) and were maintained with fresh medium containing 10 U ml$^{-1}$ IL-2 when needed. On day 3, 4, 5, 6, 7, 9 (ICS) and 10, respective wells were collected, counted and washed once in MACS buffer and incubated with anti-human FcR block (20 µl per 1 × 10$^7$ cells) (Miltenyi Biotec) for 15 min on ice before staining with PE-streptavidin-conjugated tetramers (1:100 dilution in MACS buffer) for 1 h at room temperature. After one wash, cells were incubated for 30 min on ice with the same surface stain as described for the TAME, except without anti-CD71-BV421. Cells were subsequently washed once and resuspended in MACS buffer for single cell-(index)-sorting using a BD FACSAria III for subsequent TCRαβ sequencing (days 3, 5, 6, 7 and 10) or fixed with 1% PFA for acquiring on an LSR Fortessa II (day 4), followed by analysis using FlowJo software (v.10.8.1). Proliferation due in combination with A2/M1$_{58}$ tetramer staining of CD8$^+$ T cells was used to track proliferating A2/M1$_{58}$-specific CD8$^+$ T cells over time. Ex vivo numbers of A2/M1$_{58}$$^+$CD8$^+$ T cells on day 0 were based on the frequencies obtained from TAME analysis, as frequencies were relatively low.

On day 9, expanded A2/M1$_{58}$$^+$CD8$^+$ T cells were simulated with 1 µM M1$_{58–66}$ peptide and cultured for 5 h in the presence of 10 U ml$^{-1}$ rIL-2 and Golgi Stop (BD Biosciences). Following activation, cells were surface stained for 30 min with human anti-CD3-BV510 (1:200 dilution, BioLegend 317332), anti-CD8-BV605 (1:200 dilution, BD Horizon 564116), anti-CD4-BV650 (1:200 dilution, BD Horizon 563875), anti-CD27-BV711 (1:200 dilution, BD 563167), anti-CD14-APC-H7 (1:100 dilution, BD Pharmingen 560180), anti-CD19-APC-H7 (1:100 dilution, BD Pharmingen 560177), Live/Dead near-infrared (1:800 dilution, Invitrogen L10119), anti-CD45RA-FITC (1:200 dilution, BD Biosciences 555488), PE-streptavidin-conjugated A2/M1$_{58}$ (GILGFVFTL) tetramer and anti-CD95-PECF594 (1:100 dilution, BD Horizon 562395). Cells were fixed with BD Fix-Perm buffer (BD Biosciences) for 20 min, before intracellular staining for 30 min on ice with anti-TNF-α-APC (1:100 dilution, BD 340534), anti-granzyme B-AF700 (1:50 dilution, BD 560213), anti-IFN-γ-FITC (1:100 dilution, eBioscience 45-7-319-42) and perforin (1:10 dilution, BioLegend 353316) in perm wash buffer (BD Biosciences). Cells were washed in perm wash buffer and resuspended in MACS buffer for acquiring on an LSRFortesa II followed by analysis using FlowJo software (v.10.8.1).

## Transient transfections of A2/M1$_{58}$-specific TCR α/β-chains

Genes encoding the full-length A2/M1$_{58}$ TCR α- and β-chains joined by a P2A linker for the public, newborn (pN1), child (C1), adult (A1 and pA2) and older adults (OA1, OA2a, OA2b, OA3 and pOA4) TCRs (Fig. 7a) were synthesized as double-stranded DNA fragments (Thermo Fisher Scientific) and cloned into the pMSCV-IRES-GFP II (pMIG II) expression vector[71]; a gift from D. Vignali (Addgene plasmid #52107; RRID Addgene_52107). The pMIG.TCR plasmid was transiently co-transfected with the pMIG.huCD3 plasmid in HEK293T cells using FuGENE6 transfection reagent (Promega)[72,73]. Cells were cultured for 72 h, collected by mechanical disruption and stained with anti-CD3-BV421 (1:100 dilution, BD Biosciences 562426), Live/Dead near-infrared (1:500 dilution) and PE-streptavidin-conjugated A2/M1$_{58}$ (GILGFVFTL) tetramer. Cells were fixed with 1% paraformaldehyde for acquiring on an LSRFortesa II (day 4) followed by analysis using FlowJo software (v.10.8.1).

## Generation of A2/M1$_{58}$-specific TCR-expressing cell lines

pMIG.TCR plasmids were used to retrovirally transduce SKW-3 cells expressing CD3 only or CD3 and CD8 (refs. [73–75]) as previously described[71] to generate SKW-3.CD3.public, SKW-3.CD3.pN1, SKW-3.CD3.C1, SKW-3.CD3.A1, SKW-3.CD3.pA2, SKW-3.CD3.OA1, SKW-3.CD3. OA2a, SKW-3.CD3.OA3, SKW-3.CD3.pOA4 and SKW-3.CD3 + CD8.public, SKW-3.CD3 + CD8.pN1, SKW-3.CD3 + CD8.C1, SKW-3.CD3 + CD8. A1, SKW-3.CD3 + CD8.pA2, SKW-3.CD3 + CD8.OA1, SKW-3.CD3 + CD8.

OA2a, SKW-3.CD3 + CD8.OA3, SKW-3.CD3 + CD8.pOA4. Transduced cells were stained with anti-CD3-BV421 (1:100 dilution) and Live/Dead near-infrared (1:500 dilution) and FACS sorted for GFP$^{hi}$CD3$^{hi}$ cells.

## A2/M1$_{58}$ tetramer staining of TCR-expressing cell lines

SKW-3.CD3.TCR and SKW-3.CD3$^+$CD8.TCR cell lines were stained with PE-streptavidin-conjugated A2/M1$_{58}$ (GILGFVFTL) tetramers (WT, CD8 KO or CD8 Enh) (1:100 dilution in MACS buffer), anti-CD3-BV421 (1:100 dilution) and Live/Dead near-infrared (1:500 dilution) in MACS buffer for 30 min at room temperature, washed twice and fixed with 1% para-formaldehyde for acquiring on an LSR Fortessa II followed by the analysis using FlowJo software (v.10.8.1). Median A2/M1$_{58}$ tetramer-PE MFI value across all SKW-3.TCR cell lines was established for cells expressing the same MFI for CD3-BV421 and GFP to ensure similar expression of the TCR on the cell surface.

## Peptide titration and alanine scan

C1R cells expressing HLA-A*02:01 (C1R.A2; a gift from W. Chen, La Trobe University) were maintained in cRPMI supplemented with Hygromycin B (50 mg ml$^{-1}$). C1R.A2 cells were pulsed with WT M1$_{58-66}$ (GILGFVFTL) peptide at different dilutions ranging from $10^{-4}$ to $10^{-12}$ M (half logM steps) for WT M1$_{58-66}$ peptide titration or at a dilution of $10^{-5}$ M for the alanine scan with WT M1$_{58-66}$ (GILGFVFTL), P1A (AILGFVFTL), P3A (GIAGFVFTL), P4A (GILAFVFTL), P5A (GILGAVFTL), P6A (GILGFAFTL), P7A (GILGFVATL), P8A (GILGFVFAL) or P9A (GILGFVFTA) peptides for 1 h at 37 °C. Due to a lower activation threshold found in the older adult OA1, OA2a and OA3 TCRs, a higher peptide concentration of $10^{-4}$ M was used. Peptide pulsed C1R.A2 cells were washed twice, resuspended in cRPMI and incubated at 1:1 with SKW-3.CD3.TCR or SKW-3.CD3$^+$CD8. TCR cell lines for 16–18 h at 37 °C. Dynabeads human T activator CD3/CD28 (Gibco) were used as a positive control. Cells were subsequently collected and stained with CD69-PE-Cy7 (1:100 dilution, BD 557745), Live/Dead near-infrared (1:500 dilution) in MACS buffer for 30 min at room temperature in the dark, washed twice and fixed with 1% PFA for acquiring on an LSRFortesa II, followed by analysis using FlowJo software (v.10.8.1).

## Statistical analysis

No statistical methods were used to predetermine sample sizes but our sample sizes are larger than those reported in previous publications[16]. Normality tests were not performed and nonparametric statistical analyses were performed in the study. Unless otherwise indicated, data were analyzed by GraphPad Prism (v.9.3.0, GraphPad) using a two-sided Kruskal–Wallis test combined with Dunn's correction for multiple tests. A Mann–Whitney U-test with Holm multiple testing correction was used to compare TCR frequencies in HLA-A*2$^+$ and HLA-A*2$^-$ donors as well as CDR3β motif frequencies across different age groups within the independent datasets. Differences were considered significant at $P < 0.05$. For statistical comparisons of TCR $P_{gen}$, N insertions and deletions, analyses were conducted in the R statistical environment (v.4.2.2). Each unique clonotype was included in the analyses only once and mixed-effects models were fitted via Penalized Quasi-Likelihood (glmmPQL function from the MASS package, v.7.3.58) (ref. [76]). To correct for the non-independence of the data owing to shared donors across clonotypes, subject was included as a random effect. As $P_{gen}$ best fitted a log-normal distribution, we utilized the Gaussian family with a log link. For N insertions and deletions, which best fitted a negative bino-mial distribution, we used the negative.binomial family, with theta first estimated by the glm.nb function. To control type II error, $P$ values were corrected for multiple testing using FDR adjustment. Data collection and analysis were not performed blind to the conditions of the experiments.

## Reporting summary

Further information on research design is available in the Nature Port-folio Reporting Summary linked to this article.

## Data availability

TCR sequence data (ex vivo Supplementary Table 2; in vitro Supplementary Table 3, Source Data) have been deposited in Mendeley (https://doi.org/10.17632/8jrbh6rgmx.1) and VDJdb (https://vdjdb.cdr3.net). scRNA-seq data have been deposited in NCBI Gene Expression Omnibus under accession code GSE237817. All other data are present in the article and Supplementary files or from the corresponding author upon reasonable request. Source data are provided with this paper.

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

## Acknowledgements

We thank the participating donors involved in the study as well as B. McCudden and J. Mitchell for their medical support. We thank staff at the Melbourne Cytometry Platform for the technical support and assistance. This work was supported by the ARC-Discovery grant to K.K., F.L. and S.G. (DP190102704), the Clifford Craig Foundation Project Grant to K.F. and K.K. (186), the Research Grants Council of the Hong Kong Special Administrative Region, China (T11-712/19-N) to K.K. and the NHMRC Leadership Investigator Grant to K.K. (1173871). C.E.S. received funding from the European Union's Horizon 2020 research program under the Marie Skłodowska-Curie Grant agreement (792532) and is supported by the ARC-DECRA Fellowship (DE200100185) and the UoM Establishment Grant. F.L. was supported by the National Health and Medical Research Council, Australia (project grant 1121643) and Career Development Fellow (1128416). T.H.O.N. is supported by the NHMRC Emerging Leadership Level 1 Investigator Grant (1194036), N.A.G. is supported by the ARC-DECRA Fellowship (DE210100705), E.B.C. is supported by a NHMRC Peter Doherty Fellowship (1091516), J.R. and N.L.L.G. are supported by NHMRC Leadership Investigator grants, S.G. is supported by NHMRC Senior Research Fellowship (1159272), D.I.G. was supported by an NHMRC Senior Principal Research Fellowship (1117766) and subsequently by an NHMRC investigator grant (2008913). J.C.C., A.A.M., M.V.P. and P.G.T. are supported by NIH NIAID R01 AI136514, U01AI150747 and ALSAC at St. Jude. We also thank L. Wooldridge (University of Bristol) for the provision of vectors encoding CD8 mutants of soluble HLA-A*02:01 molecules.

## Author contributions

K.K. led the study. K.K. and F.L. supervised the study. C.E.S. and K.K. designed the experiments. C.E.S., T.H.O.N., N.A.G., S.R., K.P., T.K., N.C., S.N., E.B.C. and A.E. performed and analyzed the experiments. C.E.S., J.C.C., J.S., A.A.M., M.V.P., S.R., H.A.M., T.M. and S. Sant analyzed data. N.A.G., C.S., Z.C., J.R., S.G. and D.I.G. provided crucial reagents. C.E.S., T.H.O.N., J.K., N.R., S Sonda, A.H., X.J., R.L., J.C., M.L. and K.F. recruited donor cohorts. C.E.S., J.C.C., A.A.M., M.V.P., H.A.M. and P.G.T. analyzed TCR sequences. C.E.S., J.S., S.R. and F.L. analyzed scRNA-seq data. C.E.S., T.H.O.N., N.A.G., J.C.C., A.A.M., M.V.P., N.L.L.G., P.G.T., D.I.G., S.G., F.L. and K.K. provided intellectual input into the study design and data interpretation. C.E.S., F.L. and K.K. wrote the manuscript. All authors reviewed and approved the manuscript.

## Competing interests

P.T. is on the SAB of Immunoscape and Cytoagents and has consulted for JNJ. P.T. has also received travel support and/or honoraria from Illumina and 10x Genomics and has patents related to TCR discovery and expression. All other authors declare no competing interests.

## Additional information

**Extended data** is available for this paper at https://doi.org/10.1038/s41590-023-01633-8.

**Correspondence and requests for materials** should be addressed to Katherine Kedzierska.

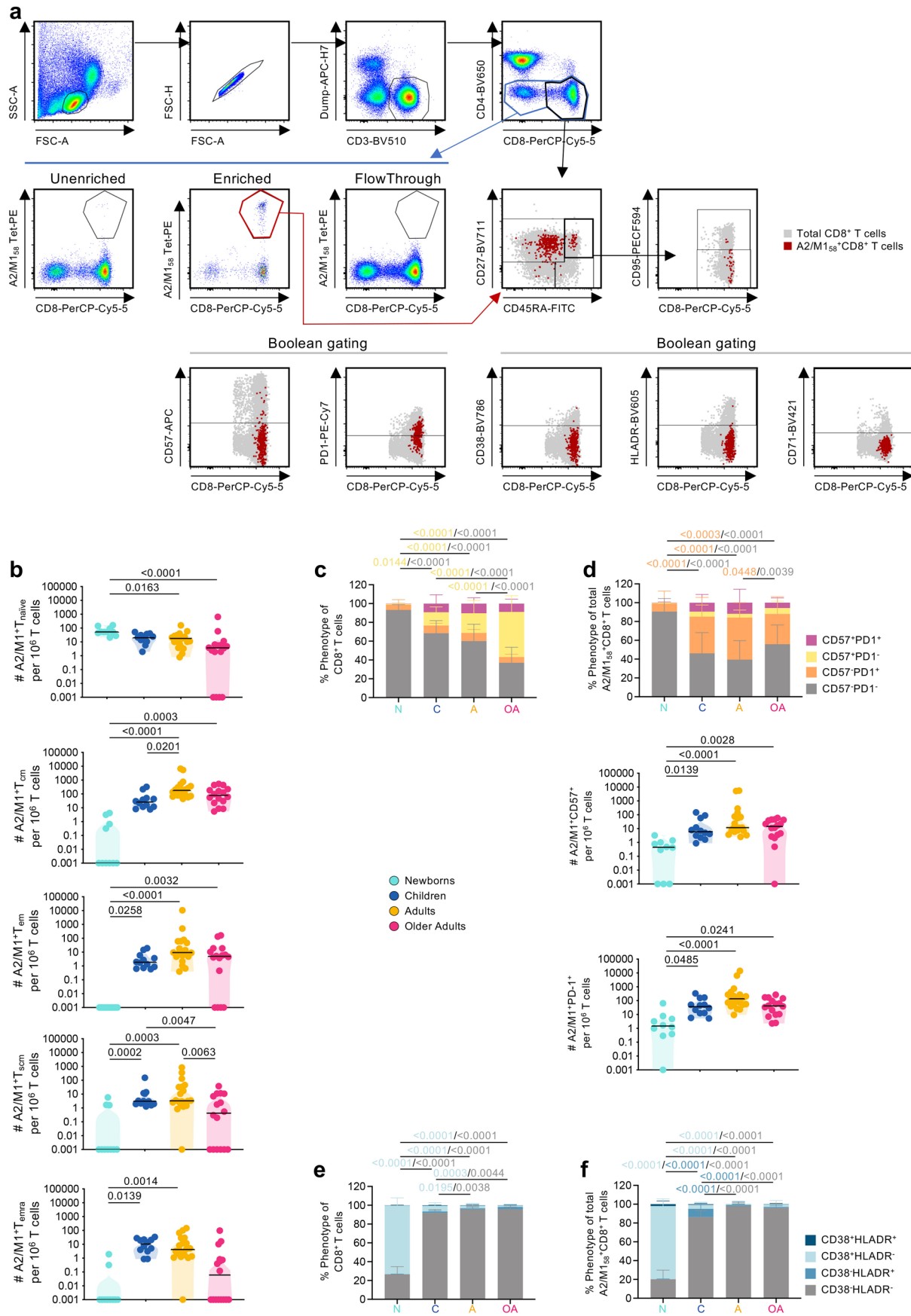

**Extended Data Fig. 1** | See next page for caption.

**Extended Data Fig. 1 | Age-related changes in A2/M1$_{58}$$^+$CD8$^+$ T cell frequencies and phenotypes. a**, Representative FACS panels indicate the gating strategy used to characterize the total CD8$^+$ T cell and the A2/M1$_{58}$$^+$CD8$^+$ T cell populations. The second line indicates unenriched, enriched and flowthrough fractions of the TAME assay. The unenriched fraction was used to define the frequency and phenotype of the total CD8$^+$ T cell population (gray gate and cell populations), whereas the A2/M1$_{58}$ tetramer-positive CD8$^+$ T cells of the enriched fraction were used to define the frequency and phenotype of the total A2/M1$_{58}$$^+$CD8$^+$ T cell population (red gate and cell populations). Naïve/memory T cell subsets were identified as T$_{cm}$ (CD27$^+$CD45RA$^-$) cells, T$_{em}$ (CD27$^-$CD45RA$^-$), T$_{emra}$ (CD27$^-$CD45RA$^+$), T$_{naïve}$ (CD27$^+$CD45RA$^+$CD95$^-$) and T$_{scm}$ (CD27$^+$CD45RA$^+$CD95$^+$) cells. Boolean gating was used to identify frequencies of individual and combined expression of CD57 and PD-1 and combined expression of CD38, HLA-DR and CD71. **b**, number of A2/M1$_{58}$$^+$phenotype$^+$CD8$^+$ T cells per 10$^6$ CD8$^+$ T cells across all age groups. Co-expression of CD57 and PD-1 in total CD8$^+$ T cells (**c**) and A2/M1$_{58}$$^+$CD8$^+$ T cells (**d**-top panel) and number of A2/M1$_{58}$$^+$CD57$^+$CD8$^+$ T cells

(**d**-middle panel) or A2/M1$_{58}$$^+$PD-1$^+$CD8$^+$ T cells (**d**-bottom panel) per 10$^6$ CD8$^+$ T cells across all age groups. Co-expression of CD38 and HLA-DR in total CD8$^+$ T cells (**e**) and A2/M1$_{58}$$^+$CD8$^+$ T cells (**f**). Numbers of A2/M1$_{58}$$^+$phenotype$^+$CD8$^+$ T cells per 10$^6$ CD8$^+$ T cell data were right shifted by 0.001 (that is absence of A2/M1$_{58}$ events in a specific phenotypic population are displayed as 0.001) (**b,d**-middle and bottom panel). Only samples with 10 or more total A2/M1$_{58}$$^+$ events were included for phenotype analysis (**b-f**). Horizontal bars indicate the median, dots represent individual donors, with n = 10 newborns, n = 12 children, n = 20 adults and n = 16 older adults in **b, d** and **f** and n = 11 newborns, n = 12 children, n = 20 adults and n = 18 older adults in **c** and **e**. Technical replicates were not performed due to limited samples. Statistical analysis was performed using a two-sided Kruskal-Wallis with Dunn's correction for multiple tests (**b, d**-middle and bottom panel) or a two-sided Tukey's multiple comparisons test (**c, d**-top panel, **e, f**). Exact significant *p*-values are indicated above the graphs. N, newborns; C, children; A, adults; OA, older Adults.

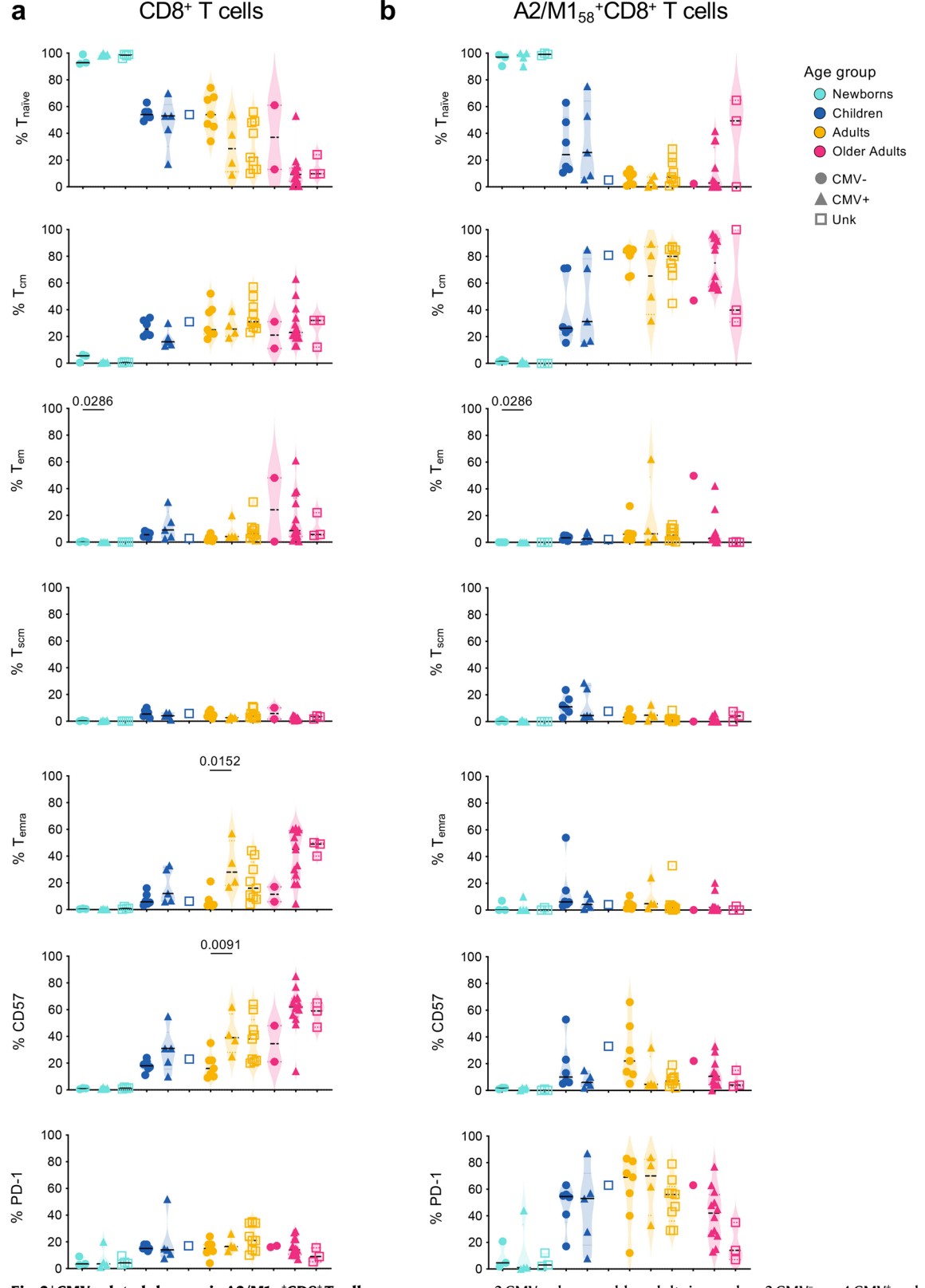

**Extended Data Fig. 2 | CMV-related changes in A2/M1$_{58}$+CD8+ T cell phenotypes.** Frequency of naïve, memory, CD57 or PD-1 expressing subsets within the total CD8+ T cell (**a**) or A2/M1$_{58}$+CD8+ T cell populations (**b**) across all age groups split based on their CMV status. Horizontal bars indicate the median, dots represent individual donors. Horizontal bars indicate the median, dots represent individual donors, with n = 3 CMV⁻, n = 4 CMV⁺ and n = 4 CMV unknown newborns, n = 6 CMV⁻, n = 5 CMV⁺ and n = 1 CMV unknown children, n = 7 CMV⁻, n = 4 CMV⁺ and n = 9 CMV unknown adults and n = 2 CMV⁻, n = 13 CMV⁺ and

n = 3 CMV unknown older adults in **a** and n = 3 CMV⁻, n = 4 CMV⁺ and n = 3 CMV unknown newborns, n = 6 CMV⁻, n = 5 CMV⁺ and n = 1 CMV unknown children, n = 7 CMV⁻, n = 4 CMV⁺ and n = 9 CMV unknown adults and n = 1 CMV⁻, n = 12 CMV⁺ and n = 3 CMV unknown older adults in **b**. Statistical analysis was performed between donors with a known positive or negative CMV status within each age group using a two-sided Mann-Whitney U-test. Exact significant *p*-values are indicated above the graphs.

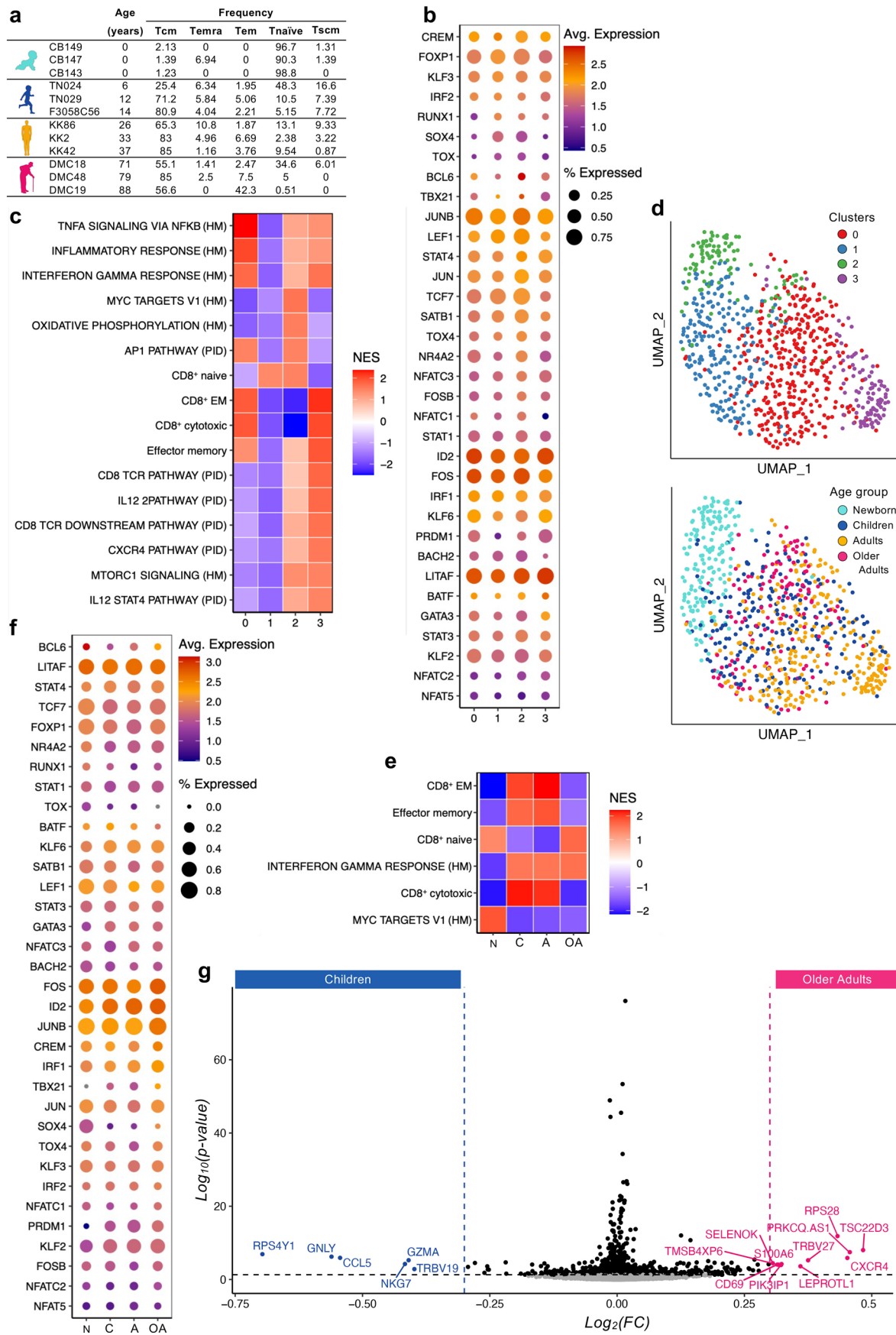

**Extended Data Fig. 3** | See next page for caption.

**Extended Data Fig. 3 | Molecular and phenotypic differentiation within A2/M1$_{58}$+CD8+ T cells across human lifespan. a**, Specification selected donors for single-cell multi-omic analysis. Phenotype frequencies were obtained via flow cytometry protein expression data. **b**, Dot plot of selected transcription factors grouped by UMAP clusters. Dot size represents the proportion with non-zero expression from each age group. The color represents average mean expression. **c**, Heatmap of enriched pathways identified from GSEA using differentially expressed genes between each UMAP cluster. All pathways shown have Benjamini–Hochberg adjusted $p$-values < 0.05 in at least one cluster. NES: Normalized enrichment score. **d**, Dimensionality reduction (UMAP) and clustering of scRNAseq data excluding TCR genes colored by clusters (top), age groups (bottom). **e**, Heatmap of enriched pathways identified from GSEA using differentially expressed genes between each age group. All pathways shown have Benjamini–Hochberg adjusted $p$-values < 0.05 in at least one age group. NES: Normalized enrichment score. **f**, Dot plot of selected transcription factors grouped by age group. Dot size represents the proportion with non-zero expression from each age group (N, newborns; C, children; A, adults; OA, older Adults). The color represents average mean expression. **g**, Volcano plot of a pairwise comparison without correction for multiple testing of differentially expressed genes between children (blue) and older adults (pink) with a fold change |log2(FC)| >0.3 and $p$-value < 0.05.

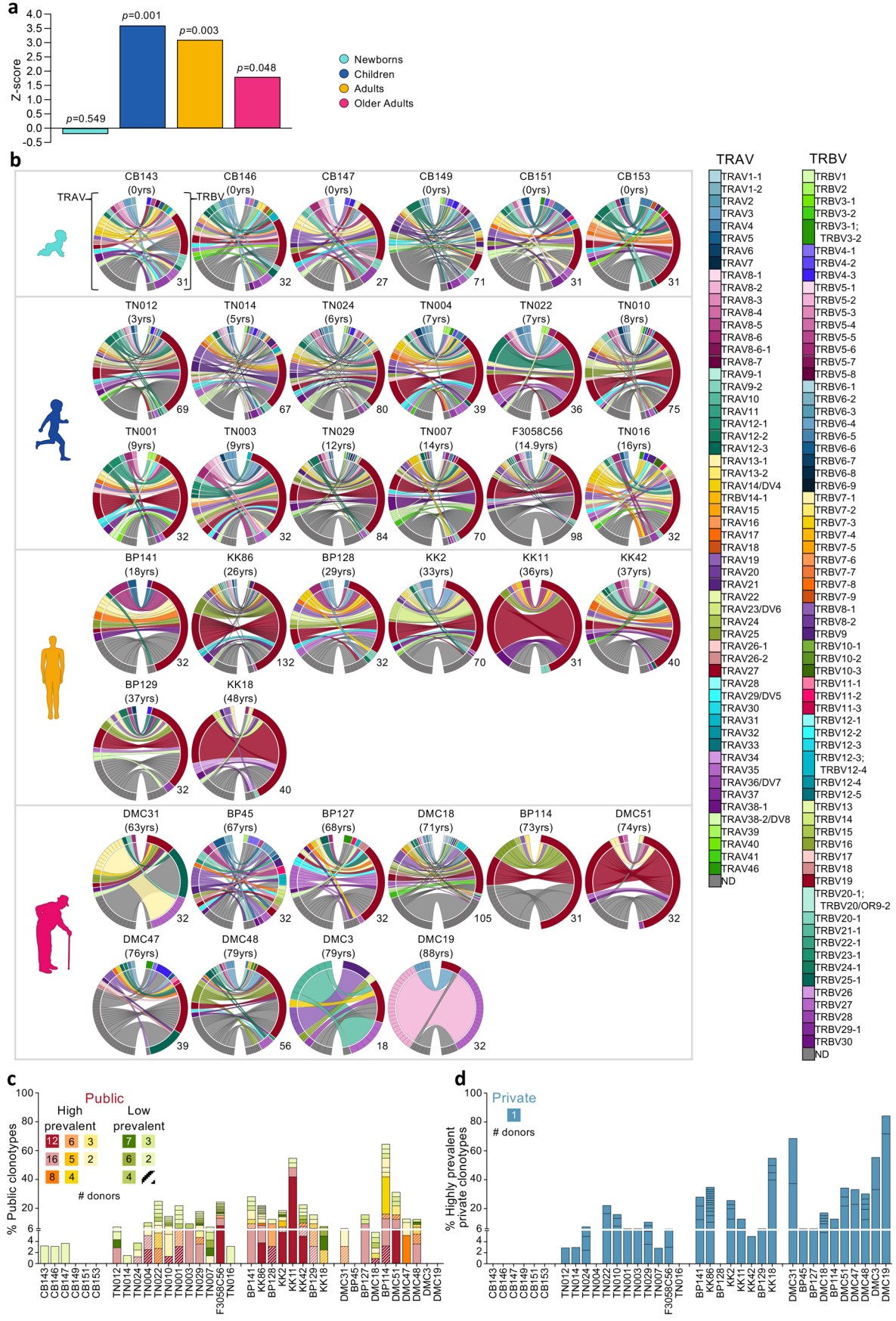

**Extended Data Fig. 4** | See next page for caption.

**Extended Data Fig. 4 | Age-related changes in A2/M1$_{58}$$^+$CD8$^+$ TCRαβ repertoire. a**, Z-score for intra- versus inter-donor distance, larger Z means intra-donor distances are smaller than inter-donor distances, that is greater heterogenticity across donors. **b**, TRAV and TRBV clonotype pairing for individual donors within each age group illustrated by circos plots. Left arch segment colors indicate TRAV usage, right outer arch colors depict TRBV usage. Connecting lines indicated TRAV-TRBV gene pairing and are colored based on their TRAV usage and segmented based on their CRD3α and CDR3β sequence, the thickness is proportional to the number of TCR clones with the respective pair. The number of sequences considered for each circos plot is shown at the right bottom. **c**, Frequency of high (TCRs detected ≥2 within a single individual)

and low (TCRs detected once with a single individual) prevalent public (shared) clonotypes across individuals of age groups. Dark red represents high prevalent public TCR (TRAV27, TRAJ42, CDR3α GAGGGSQGNLIF, TRBV19, TRBV2-7, CDR3β CASSIRSSYEQYF), whereas the light red are clonotypes expressing the full public TCRβ chain (TRBV19, TRBV2-7, CDR3β CASSIRSSYEQYF) but TCRα chain could not be identified. Numbers in the squares represent the number of donors in which the specific high versus low prevalent clonotype was identified. **d**, Frequency of high prevalent (TCRs detected ≥2 within a single individual) private (not shared) clonotypes across individuals of age groups. Statistical analysis was performed using a two-sided Kruskal-Wallis with Dunn's correction for multiple tests. Exact significant $p$-values are indicated above the graphs.

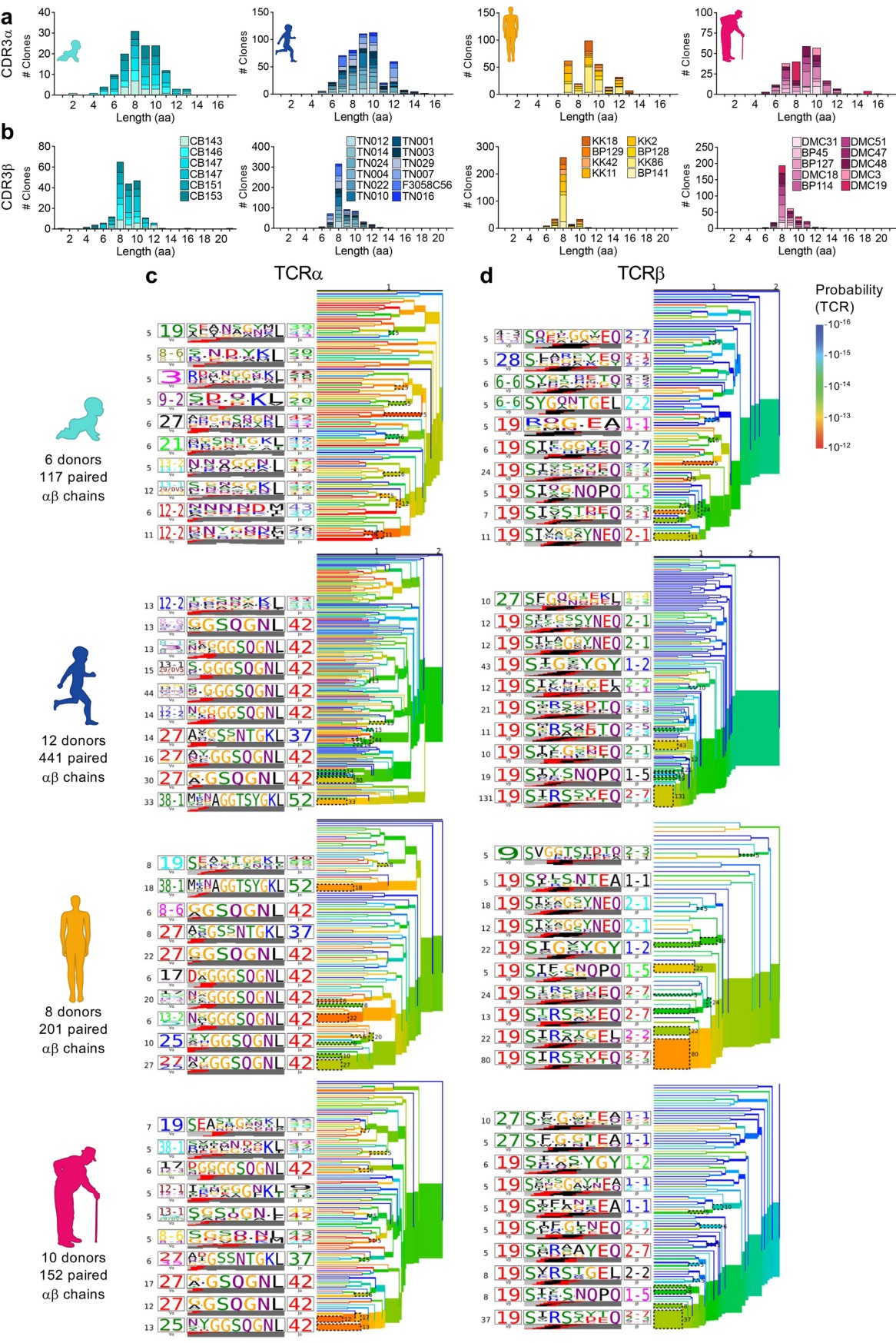

**Extended Data Fig. 5** | See next page for caption.

**Extended Data Fig. 5 | Age-related changes in A2/M1$_{58}$+CD8+ CDR3αβ repertoire and probability of generation.** Distribution of CDR3α (**a**) and CDRβ (**b**) amino acid lengths, calculated using the Chothia nomenclature, across all age groups. Average-linkage dendrograms of TCR clustering for the TCRα (**c**) and TCRβ (**d**) A2/M1$_{58}$+CD8+ repertoires generated by TCRdist. Each clustering was generated using a fixed-distance threshold algorithm and colored by generation probability (red, highest; blue, lowest probability of ease of TCR recombination).

The probability is relative between TCRs across different age groups. TCRlogos for selected subsets (corresponding to the branches of the dendrogram enclosed in dashed boxes) are shown, labeled by cluster size both to the left of each logo and to the right of the corresponding branches. Each TCR logo depicts the V- and J-gene frequencies, the CDR3 amino acid sequence, and the inferred rearrangement structure of the grouped receptors (colored by source region, light gray for the V-region, dark gray for J, black for D, and red for N-insertions).

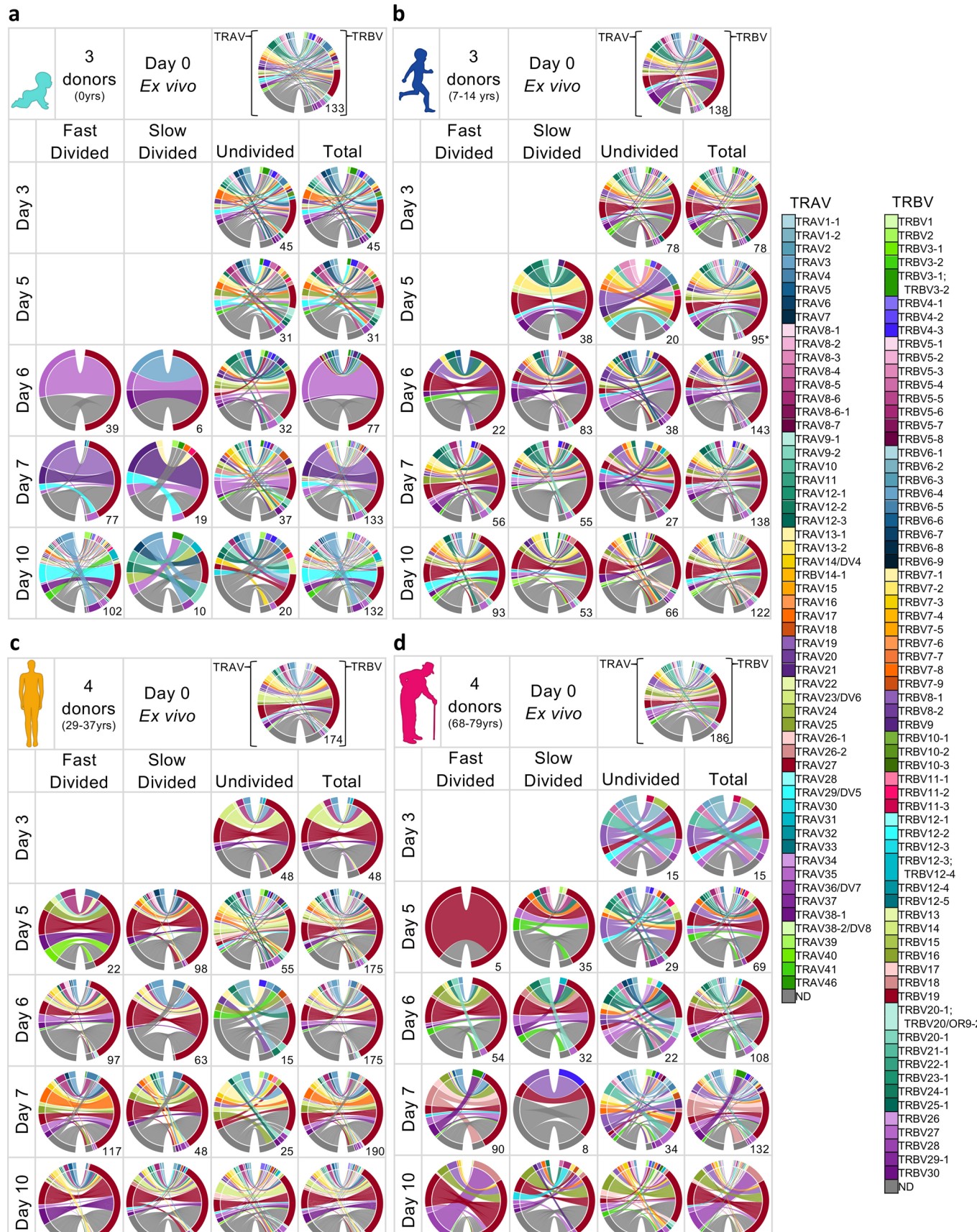

**Extended Data Fig. 6** | See next page for caption.

**Extended Data Fig. 6 | Divided and undivided A2/M1$_{58}$$^+$CD8$^+$ TCRαβ repertoires across age groups.** TRAV and TRBV clonotype pairing for pooled donors within each age group, newborns (**a**), children (**b**), adults (**c**) and older adults (**d**), illustrated by circos plots for fast, slow, undivided and total A2/M1$_{58}$$^+$CD8$^+$ T cells. Left arch segment color indicates TRAV usage, right outer arch color depicts TRBV usage. Connecting lines indicate TRAV-TRBV gene pairing and are colored based on their TRAV usage and segmented based on their CRD3α and CDR3β sequence. The thickness is proportional to the number of TCR clones with the respective pair. The number of sequences considered for each circos plot is shown at the right bottom.

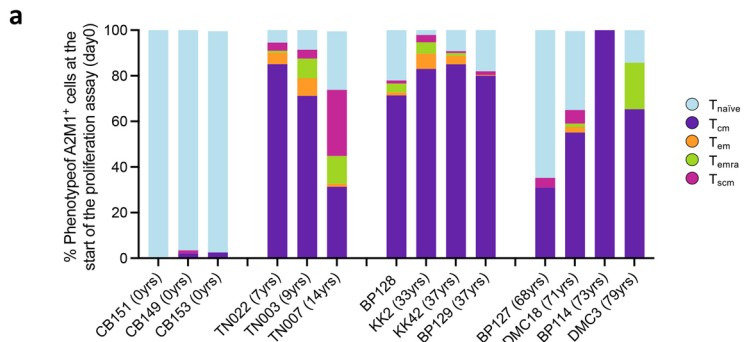

**Extended Data Fig. 7** | See next page for caption.

**Extended Data Fig. 7 | Age-related changes to A2/M1$_{58}$$^+$CD8$^+$ CDR3αβ-motifs.** **a**, Phenotypic distribution of individual donors at the start of the proliferation assay (day 0). **b**, The top-scoring A2/M1$_{58}$$^+$CD8$^+$ CDR3α (top TCR logo) and CDR3β (bottom TCR logo) sequence motifs for each age group on day 0, 3, 5, 6, 7 and 10 following M1$_{58-66}$ peptide stimulation. Each logo depicts the V- (left side) and J- (right side) gene frequencies with the CDR3 amino acid sequence in the middle with the full height (top) and scaled (bottom) by per-residue reparative entropy to background frequencies derived from TCRs with matching gene-segment composition to highlight motif positions under selection. The middle section indicates the inferred rearrangement structure by source region (light gray for V-region, dark gray for J, black for D and red for N-insertions) of the grouped receptors.

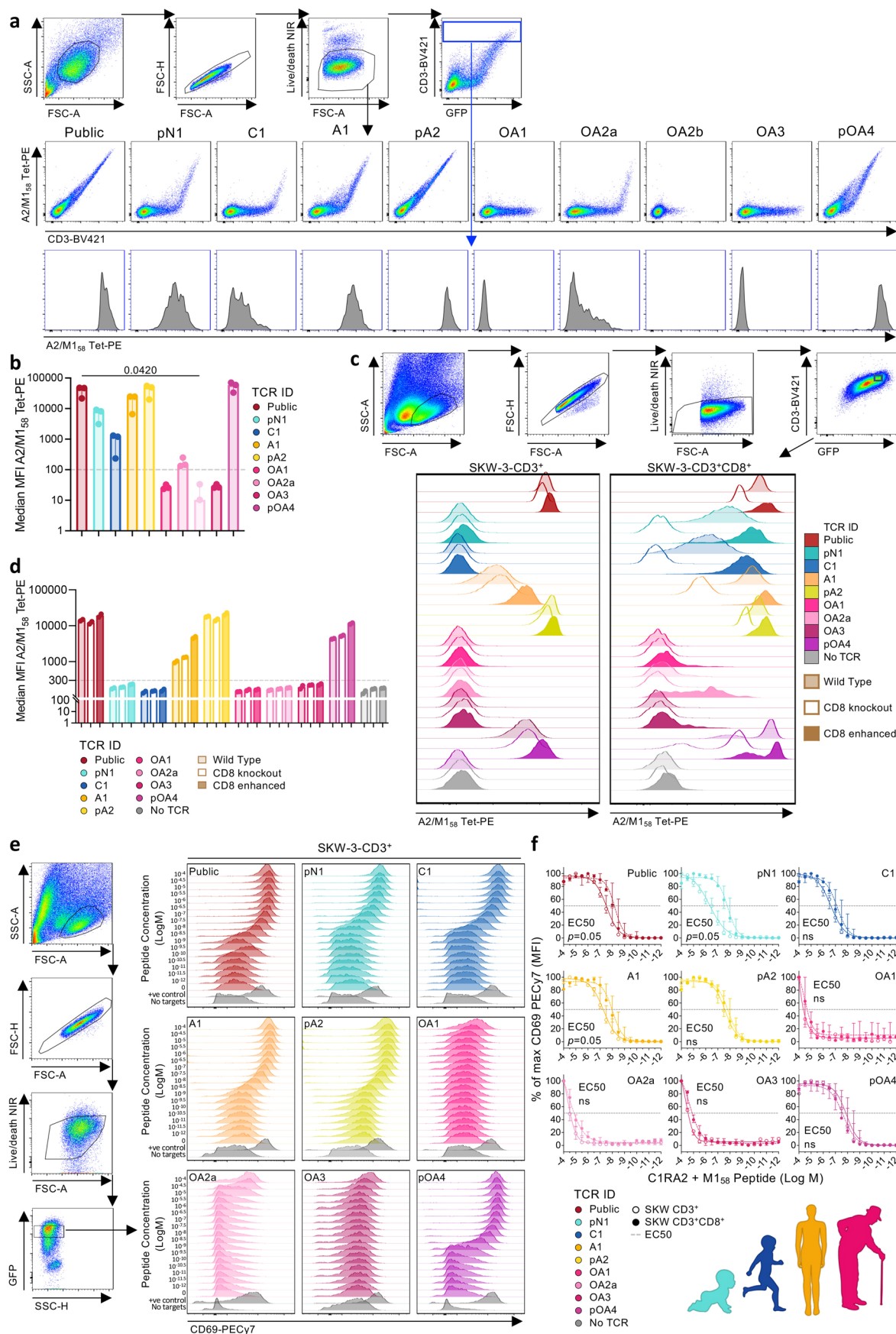

**Extended Data Fig. 8** | See next page for caption.

**Extended Data Fig. 8 | Transient and stable expression of age-specific TCR.**
**a**, Gating strategy transient transfection age-specific TCRs in HEK293T cells.
**b**, Median MFI A2/M1$_{58}$ tetramer-PE staining of transiently expressed TCRs in
HEK293T cells, dotted line indicates MFI threshold set by the parental cell line
expressing no TCR (n = 3 independent experiments, median and IQR). **c**, Gating
strategy and A2/M1$_{58}$ tetramer-PE staining with a normal (A2/M1$_{58}$-WT, lightly
shaded), knockout CD8-binding site (A2/M1$_{58}$-KO open) and enhanced CD8-
binding site (A2/M1$_{58}$-Enh, closed) tetramer of SKW-3-CD3$^+$ and SKW-3-CD3$^+$CD8$^+$
TCR-expressing cell lines. **d**, Median MFI of A2/M1$_{58}$ tetramer-PE staining with a
normal (A2/M1$_{58}$-WT, lightly shaded), knockout CD8-binding site (A2/M1$_{58}$-KO
open) and enhanced CD8-binding site (A2/M1$_{58}$-Enh, closed) tetramer of SKW-
3-CD3$^+$ TCR-expressing cell lines (n = 2 independent experiments), dotted line

indicates MFI threshold set by the parental cell line expressing no TCR. **e**, Gating
strategy and CD69-PECy7 expression following peptide titration for public and
age group specific TCRs (SKW-3-CD3$^+$ cells). **f**, Percentage of maximum CD69-
PECy7 MFI for age-specific TCRs expressed on SKW-3-CD3$^+$ (open circles) and
SKW-3-CD3$^+$CD8$^+$ (closed circles) cell lines following peptide titration, EC50
indicated by the dotted line (n = 3 independent experiments, median and range).
Statistical analysis was performed using a two-sided Kruskal-Wallis with Dunn's
correction for multiple tests between the public and the age-specific TCRs (**b**)
and a one-tailed Mann-Whitney U-test was performed for the EC50 between
SKW-3-CD3$^+$CD8$^-$ SKW-3-CD3$^+$CD8$^+$ TCR-expressing cell lines (**f**). Exact significant
*p*-values are indicated above the graphs.

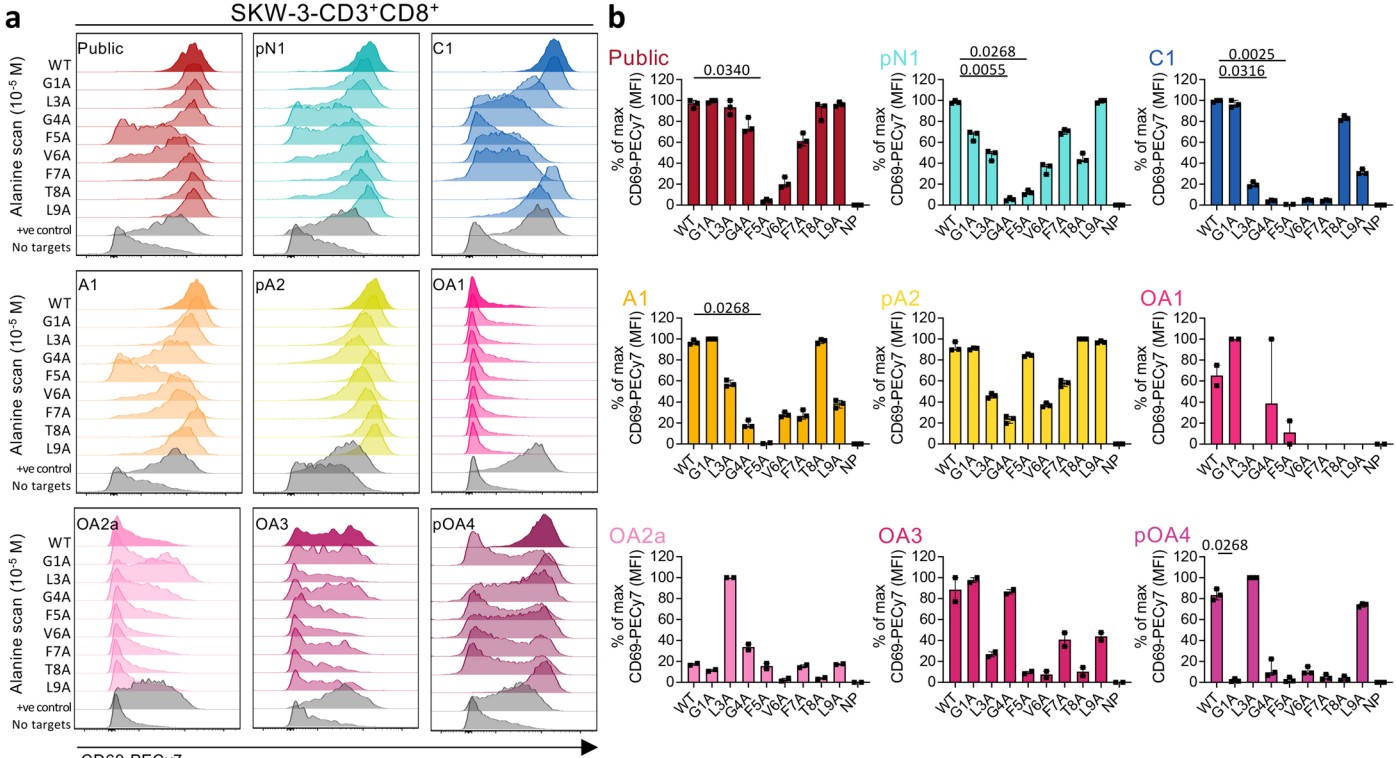

**Extended Data Fig. 9 | TCR recognition patterns of age-specific TCRs. a**, Gating strategy and CD69-PECy7 expression profiles of SKW-3-CD3+CD8+ TCR-expressing cell lines following stimulation with the single alanine mutant M1$_{58-66}$ peptides. (**b**) Frequency of maximum CD69-PECy7 MFI following stimulation with M1$_{58-66}$ alanine scan peptides for age-specific TCRs expressed on SKW-3-CD3+CD8+ cells (n = 3 independent experiments, median and range). Statistical analysis was performed using a two-sided Kruskal-Wallis with Dunn's correction for multiple tests between the public and the age-specific TCRs (**b**). Exact significant p-values are indicated above the graphs.

# Reporting Summary

## Statistics

For all statistical analyses, confirm that the following items are present in the figure legend, table legend, main text, or Methods section.

| n/a | Confirmed | |
|---|---|---|
| ☐ | ☒ | The exact sample size ($n$) for each experimental group/condition, given as a discrete number and unit of measurement |
| ☐ | ☒ | A statement on whether measurements were taken from distinct samples or whether the same sample was measured repeatedly |
| ☐ | ☒ | The statistical test(s) used AND whether they are one- or two-sided *Only common tests should be described solely by name; describe more complex techniques in the Methods section.* |
| ☒ | ☐ | A description of all covariates tested |
| ☐ | ☒ | A description of any assumptions or corrections, such as tests of normality and adjustment for multiple comparisons |
| ☐ | ☒ | A full description of the statistical parameters including central tendency (e.g. means) or other basic estimates (e.g. regression coefficient) AND variation (e.g. standard deviation) or associated estimates of uncertainty (e.g. confidence intervals) |
| ☐ | ☒ | For null hypothesis testing, the test statistic (e.g. $F$, $t$, $r$) with confidence intervals, effect sizes, degrees of freedom and $P$ value noted *Give P values as exact values whenever suitable.* |
| ☒ | ☐ | For Bayesian analysis, information on the choice of priors and Markov chain Monte Carlo settings |
| ☒ | ☐ | For hierarchical and complex designs, identification of the appropriate level for tests and full reporting of outcomes |
| ☒ | ☐ | Estimates of effect sizes (e.g. Cohen's $d$, Pearson's $r$), indicating how they were calculated |

*Our web collection on statistics for biologists contains articles on many of the points above.*

## Software and code

Policy information about availability of computer code

| Data collection | LSR Fortessa II (BD Biosciences), BD FACSAria III (BD Biosciences) ; NextSeq500 platform |
|---|---|
| Data analysis | FlowJo v10.8.1; Prism v9.3.0; R v.4.2.2, TCRdist3, Trimmomatic v0.39, TopHat v2.1.1, Cufflinks v2.2.1, VDJPuzzle v3, Seurat v4.1.0, MAST v1.16.0, Scanpy v1.7.1, FinchTV v1.5.0, MASS package, v7.3.58 |

For manuscripts utilizing custom algorithms or software that are central to the research but not yet described in published literature, software must be made available to editors and reviewers. We strongly encourage code deposition in a community repository (e.g. GitHub). See the Nature Portfolio guidelines for submitting code & software for further information.

## Data

Policy information about availability of data

All manuscripts must include a data availability statement. This statement should provide the following information, where applicable:

- Accession codes, unique identifiers, or web links for publicly available datasets
- A description of any restrictions on data availability
- For clinical datasets or third party data, please ensure that the statement adheres to our policy

Data will be made available according to our Data availability statement provided in the manuscript
Data availability
TCR sequence data (ex vivo Supplementary Table 2; in vitro Supplementary Table 3, Source Data) will be deposited into VDJdb [https://vdjdb.cdr3.net] following manuscript acceptance. The published article includes all datasets generated or analyzed during the study in the source data file. scRNA-seq data that support the study is deposited in NCBI-GEO with accession GSE237817 and will be available following publication of the manuscript. This paper does not report original code. Any additional information required to reanalyze the data reported in this paper is available from the lead contact upon request.

Specific details:
TCR sequence data will be deposited into VDJdb [https://vdjdb.cdr3.net] following manuscript acceptance and will be linked to the PMID
scRNA-seq data that support the study will be deposited in NCBI-GEO with accession GSE237817 following manuscript acceptance.
A source data file will be generated to show all data points.
Raw FACS data are shown in the manuscript. FACS-source files are available upon request.
Msigdb was used to perform GSEA and was accessed automatically by the GSEA R package fgsea function.

# Field-specific reporting

Please select the one below that is the best fit for your research. If you are not sure, read the appropriate sections before making your selection.

☒ Life sciences ☐ Behavioural & social sciences ☐ Ecological, evolutionary & environmental sciences

For a reference copy of the document with all sections, see nature.com/documents/nr-reporting-summary-flat.pdf

# Life sciences study design

All studies must disclose on these points even when the disclosure is negative.

| | |
|---|---|
| Sample size | The sample size was determined by the availability of samples. |
| Data exclusions | No data were excluded with the following exception: Donors who had a total number of less then 10 counted A2M1+CD8+ T cells within the whole enriched fraction were excluded for further phenotypic analysis as cell numbers were to low. This was indicated in the manuscript. |
| Replication | Experiments in figure 1-7 could not be replicated due to limited PBMC numbers. SKW-3 cell line experiments (figure 8) were repeated in 2 or 3 independent experiments as indicated in the figure legends. |
| Randomization | N/A. Donors were selected from a large randomly recruited and HLA-typed lifespan cohort n = >500, consisting of 154 newborns, 30 children, ~300 adults and 57 older adults, based on their expression of HLA-A*02:01. Samples were used based on availability of material. |
| Blinding | Experiments were not blinded, instead all analysis have been independently checked by multiple researches |

# Reporting for specific materials, systems and methods

We require information from authors about some types of materials, experimental systems and methods used in many studies. Here, indicate whether each material, system or method listed is relevant to your study. If you are not sure if a list item applies to your research, read the appropriate section before selecting a response.

### Materials & experimental systems

| n/a | Involved in the study |
|---|---|
| ☐ | ☒ Antibodies |
| ☐ | ☒ Eukaryotic cell lines |
| ☒ | ☐ Palaeontology and archaeology |
| ☒ | ☐ Animals and other organisms |
| ☐ | ☒ Human research participants |
| ☒ | ☐ Clinical data |
| ☒ | ☐ Dual use research of concern |

### Methods

| n/a | Involved in the study |
|---|---|
| ☒ | ☐ ChIP-seq |
| ☐ | ☒ Flow cytometry |
| ☒ | ☐ MRI-based neuroimaging |

## Antibodies

| | |
|---|---|
| Antibodies used | We used commercially-available antibodies as per Material and Methods.<br><br>TAME and proliferation<br>Surface staining: anti-CD71-BV421 (Clone M-A712, 1:50, BD Biosciences #562995, TAME only), anti-CD3-BV510 (Clone OKT3, 1:200, BioLegend #317332), anti-HLA-DR-BV605 (Clone L243, 1:100, BioLegend 307640), anti-CD4-BV650 (Clone SK3, 1:100, BD Biosciences #563875), anti-CD27-BV711 (Clone L128, 1:200, BD Horizon #563167), anti-CD38-BV785 (Clone HIT2, 1:100, BD Biosciences #563964), anti-CD57-APC (Clone NK-1, 1:400, BD Biosciences #560845), anti-CCR7-AF700 (Clone 150503, 1:50, BD Biosciences #561143), anti-CD14-APC-Cy7 (Clone MϕP91:100, BD Biosciences #560180), anti-CD19 (Clone SJ25C1, 1:100, BD Biosciences #560177), anti-CD45RA-FITC (Clone HI100, 1:200, BD Biosciences #555488), anti-CD8-PerCP-Cy5.5 (1:200, BD Biosciences #565310), anti-CD95-PECF594 (Clone SK1, 1:100, BD Horizon #562395), anti-PD1-1-PE-Cy7 (Clone EH12.1, 1:50, BD Biosciences #561272), Live/Dead fixable aqua dead-cell stain (1:800, Invitrogen #L10119), cell trace violet (Violet Proliferation Dye 450, BD Horizon, proliferation only)<br><br>ICS<br>Surface staining: anti-CD3-BV510 (Clone OKT3, 1:200, Biolegend #317332), anti-CD8-BV605 (Clone SK1, 1:200, BD Horizon #564116), |

anti-CD4-BV650 (Clone SK3, 1:200, BD Horizon #563875), anti-CD27-BV711 (Clone L128, 1:200, BD #563167), anti-CD14-APC-H7 (Clone MφP91:100, 1:100, BD Pharmingen #560180), anti-CD19-APC-H7 (Clone SJ25C1, 1:100, BD Pharmingen #560177), Live/Dead near-infrared (1:800, Invitrogen #L10119), anti-CD45RA-FITC (Clone HI100, 1:200, BD Biosciences #555488), anti-CD95-PECF594 (Clone SK1, 1:100, BD Horizon #562395).
Intracellular staining: anti-TNFa-APC (Clone 6401.1111, 1:100 BD, #340534), anti-Granzyme B-AF700 (Clone GB11, 1:50, BD #560213), anti-IFNy-FITC (Clone 4S.B3, 1:100, eBioscience #45-7-319-42), Perforin (Clone B-D48, 1:10, Biolegend #353316)

TCR cell lines
anti-CD3-BV421 (1:100, BD Biosciences #562426), Live/Dead fixable aqua dead-cell stain (1:500, Invitrogen #L10119), CD69-PE-Cy7 (Clone FN50, 1:100, BD #557745)

| Validation | All antibodies were validated for use on human cells by the manufacturers, and were titrated in our laboratory before use. |
| --- | --- |

# Eukaryotic cell lines

Policy information about cell lines

| Cell line source(s) | HEK293T cells were obtained from ATCC (www.atcc.org), C1RA2 cells were obtained from the Chen laboratory (La Trobe University), C1R parental cell line was obtained from the Department of Biochemistry and Molecular Biology & Infection and Immunity Program, Biomedicine Discovery Institute Monash University SKW-3-CD3 and SKW-3-CD3+CD8 were obtained from the McCluskey Laborartory (University of Melbourne) |
| --- | --- |
| Authentication | Cell lines were not formally authenticated. C1R-A2 cells were routinely tested for HLA expression level prior to each experiment. Presence of CD8 and/or CD3 in SKW-3 lines were confirmed by GFP expression |
| Mycoplasma contamination | Cell lines tested negative for mycoplasma contamination |
| Commonly misidentified lines (See ICLAC register) | No commonly misidentified cell lines were used in this study |

# Human research participants

Policy information about studies involving human research participants

| Population characteristics | Please refer to Supplementary Tables 1 for details |
| --- | --- |
| Recruitment | Samples were recruited through the University of Melbourne (UoM). Australian Red Cross Blood Service (ARCBS), Deepdene Medical Clinic (DMC) (through co-author J. Crowe), Launceston General Hospital (through co-author K. Flanagan), St Jude Children's Research Hospital (through co-author P. Thomas), Mercy Hospital for Women (through co-author M. Lappas). All donors were recruited randomly and on voluntary basis. Signed informed consents were obtained from all blood donors or their guardians prior to the study. Participants of the study did not receive any compensation. |
| Ethics oversight | Experiments conformed to the Declaration of Helsinki Principles and the Australian National Health and Medical Research Council Code of Practice. Written informed consent was obtained from all blood donors or their guardians prior to the study. The study was approved by the Human Research Ethics Committee (HREC) of the University of Melbourne (Ethics ID #24567; #13344; #23852), Australian Red Cross Lifeblood (ID 2015#8), St Jude Children's Research Hospital (XPD12-089 IIBANK), Mercy Hospital for Women (#R14-25) and Tasmanian Health and Medical Human Research Ethics Committee (ID H0017479). |

Note that full information on the approval of the study protocol must also be provided in the manuscript.

# Flow Cytometry

## Plots

Confirm that:

☒ The axis labels state the marker and fluorochrome used (e.g. CD4-FITC).

☒ The axis scales are clearly visible. Include numbers along axes only for bottom left plot of group (a 'group' is an analysis of identical markers).

☒ All plots are contour plots with outliers or pseudocolor plots.

☒ A numerical value for number of cells or percentage (with statistics) is provided.

## Methodology

| Sample preparation | Samples were prepared as described in Methods |
| --- | --- |
| Instrument | BD LSRII Fortessa and BD FACSAriaIII were used for acquisition of data, BD FACS AriaIII was used for single cell index sorting |
| Software | BD FACS Diva, FlowJo |

| | |
|---|---|
| Cell population abundance | Only single cell sorting was performed, which was confirmed by the presence of single TCR strains |
| Gating strategy | Gating strategy has been described in Figure 1c, Figure 7c and f, Figure 8b,c,e, g,and i, extended data figure 1a, extended data figure 8a,c and e and extended data figure 9a |

TAME: FSC-A vs SSC-A gating --> doublet exclusion (FSC-A vs FSC-H) --> dead cell exclusion (LD staining). Immune cells were gated by classical markers (e.g. CD3, CD4 and CD8). T cell subsets were defined by CD27, CD45RA and CD95. A2/M1 specific cells were identified by teramer staining. Boolean gating was performed for CD57, PD1, CD38, HLADR and CD71 among viable CD8+ or tetramer+ populations

TAME: FSC-A vs SSC-A gating --> doublet exclusion (FSC-A vs FSC-H) --> dead cell exclusion (LD staining). Immune cells were gated by classical markers (e.g. CD3, CD4 and CD8). T cell subsets were defined by CD27, CD45RA and CD95. A2/M1 specific cells were identified by teramer staining. Boolean gating was performed for CD57, PD1, CD38, HLADR and CD71 among viable CD8+ or tetramer+ populations

Proliferation: similar to TAME, except for CD71 staining which was not included. Instead proliferating tetramer populations were identified by the loss of VPD450

ICS: FSC-A vs SSC-A gating --> doublet exclusion (FSC-A vs FSC-H) --> dead cell exclusion (LD staining). Immune cells were gated by classical markers (e.g. CD3, CD4 and CD8). T cell subsets were defined by CD27, CD45RA and CD95. A2/M1 specific cells were identified by teramer staining. Boolean gating was performed for IFNy, TNFa, GrzB and Perforin among viable CD8+ or tetramer+ populations. Proliferating tetramer populations were identified by the loss of VPD450

TCR cell lines: FSC-A vs SSC-A gating --> doublet exclusion (FSC-A vs FSC-H) --> dead cell exclusion (LD staining). TCR expressing cell lines were identified based on CD3 and GFP coexpression. A2/M1 specificity was established by teramer staining on CD3+GFP+ cells. T cell activation was measured by CD69 expression on GFP positive cells.

☒ Tick this box to confirm that a figure exemplifying the gating strategy is provided in the Supplementary Information.

