## [Peer Review File · Nature Immunology]

Peer Review Information

Journal: Nature Immunology

Manuscript Title: Newborn/child-like molecular signatures in older adults stem from TCR repertoire shifts across the human lifespan

Corresponding author name(s): Professor Katherine Kedzierska

Reviewer Comments & Decisions:

Decision Letter, initial version:
--

10th May 2023

Dear Professor Kedzierska,

Your Article, "Bifurcated trajectory of epitope-specific CD8+ T cells across the human lifespan stems from newborn/child-like molecular signatures in the elderly" has now been seen by 2 referees. You will see from their comments below that while they find your work of interest, some important points are raised. We are very interested in the possibility of publishing your study in Nature Immunology, but would like to consider your response to these concerns in the form of a revised manuscript before we make a final decision on publication.

Please note that reviewer 1 is requesting some further mechanistic insight where they comment that what would be ideal is "...a biological explanation for the observed repertoire diversification and replacement of public with private clonotypes with aging". If you are unable to provide new data to address this point we would not reject the paper just on those grounds, but in that case the manuscript might be better reformatted and submitted as a Resource paper. Please let me know if anything is unclear.

We therefore invite you to revise your manuscript taking into account all reviewer and editor comments. Please highlight all changes in the manuscript text file in Microsoft Word format.

* If you have not done so already please begin to revise your manuscript so that it conforms to our Article format instructions at <http://www.nature.com/ni/authors/index.html>. Refer also to any guidelines provided in this letter.

* Please include a revised version of any required reporting checklist. It will be available to referees to aid in their evaluation of the manuscript goes back for peer review. They are available here:

Reporting summary:

When submitting the revised version of your manuscript, please pay close attention to our [href="https://www.nature.com/nature-portfolio/editorial-policies/image-integrity">Digital Image Integrity Guidelines.](https://www.nature.com/nature-portfolio/editorial-policies/image-integrity) and to the following points below:

[REDACTED]

We hope to receive your revised manuscript within two weeks. If you cannot send it within this time, please let us know. We will be happy to consider your revision so long as nothing similar has been accepted for publication at Nature Immunology or published elsewhere.

Nature Immunology is committed to improving transparency in authorship. As part of our efforts in this direction, we are now requesting that all authors identified as 'corresponding author' on published papers create and link their Open Researcher and Contributor Identifier (ORCID) with their account on the Manuscript Tracking System (MTS), prior to acceptance. ORCID helps the scientific community achieve unambiguous attribution of all scholarly contributions. You can create and link your ORCID from the home page of the MTS by clicking on 'Modify my Springer Nature account'. For more information please visit <http://www.springernature.com/orcid>.

We look forward to seeing the revised manuscript and thank you for the opportunity to review your

work.

Sincerely,

Nick Bernard, PhD
Senior Editor
Nature Immunology

Reviewers' Comments:

Reviewer #1:

Remarks to the Author:

The manuscript by van de Sandt et al. represents an interesting and potentially important study of Ag-specific clonal T cell population changes across ages. This cross-sectional study begins with global phenotype analysis of different T cell subsets and compare them to phenotypes of Influenza virus-specific CD8 T cells directed against the Matrix peptide M158 bound to HLA-A2 in different age groups, followed by transcriptional and TCR diversity, component utilization and motif utilization analysis across aging. The analysis collectively suggests that the very diverse repertoires in the neonates progressively narrow down to generate a repertoire dominated by highly functional, public TCR clonotypes associated with strong, broad and functionally optimal recognition of the pMHC.

Surprisingly, such cells do not appear to persist in most older adults, being partially replaced by private TCR clonotypes whose transcriptional signatures resembled those of neonatal and childhood T cell profiles and gravitated more towards central memory phenotypes. The authors then performed extensive TCR structure-function analysis following transfection of TCRs from participants of different ages into HEK293 cells, and demonstrate that private clonotypes from older adults generally exhibit lower CD8 dependency, higher activation and lower recognition of peptide epitope variants. The study is well designed and well performed, and brings new information to the field. Scope and depth of work is strong. However, several issues need to be addressed before it can be recommended for publication. Major criticisms (all must be addressed by revision or experimentally):

1. The title of the study and the discussion related to it uses the term "bifurcation", which in my opinion is misleading. Bifurcation implies moving in different directions from the same starting point. However, all the older adults in this type of study had to be adults at some point in time, and the authors are really studying (in a cross-sectional manner, and therefore not in the same individuals) how the Ag-response changes with aging. So the phenomena the authors discuss are much better described as clonotype or repertoire shifts. This is particularly true because they document partial, and not complete, replacement of public clonotypes with private ones with aging.
2. No cell numbers are provided for the analysis. This is a critical data point to interpret what is going on – are there relative or absolute changes in different cell subsets.
3. Directly related to #2, authors need to report the status of their participants for CMV infection, to exclude it as a potential confounder. Several graphs show bimodal distributions or outliers of data (e.g. Fig. 1i, with Ag-specific T_{cm}, and somewhat with T_n and T_{em}), and one would want to know whether there were any impact of CMV carriage.
4. Staining experiments to ascertain protein expression are needed to document whether there are cells with virtual memory/antigen-independent memory phenotypes in any of the tested populations. Transcriptomic data could be misleading in this regard.
5. Figure 7, for activation studies, cells were enriched using magnetic beads. What were the surface phenotypes of enriched cells and were they comparable between the age groups? Different

differentiation states could have very well caused differences in proliferation and activation and this needs to be controlled for. This data is also necessary to interpret the results.

6. Critically, the authors leave us without a biological explanation for the observed repertoire diversification and replacement of public with private clonotypes with aging. Where are the private clonotypes coming from and why? Do T cells carrying the public clonotypes move into tissues perhaps, or do they die due to lack of maintenance? Do the private clonotypes that arise in older participants have high affinity/avidity for self pMHC to allow survival? Can the authors do any experiments to address this issue? I understand that a definitive proof may be very difficult to obtain, but at least some insight would help,

Other comments:

- Ln. 259 – were there 793 cells total, or per group, for this analysis? Please elaborate
- There is a typo on ln. 381 – should be PAGA, not PAGE.
- Ln. 653, likely should be “children and elderly” and not children and adult.
- I urge the authors to carefully examine references and make sure that they credit original work. For example, the reference for thymic involution timing in humans cannot be the THome reference (#13); at a minimum, the elegant studies of Douek and colleagues [Douek, DC et al., Nature 1998], should be quoted instead, and I urge the authors to look to earlier work by groups that discovered thymus (Miller, JFAP; Good RA; Waksman, B) or Barton F. Haynes to assure proper attribution. Second, the statement in ln.95-98 is supported by references 11, 15 and 18-20. However, the statement that Tn cells are lost with aging and terminally differentiated cells increase, needs additional references Wertheimer, A.M. et al., J. Immunol 2014; and Thyagarajan B., et al., J.Geront.Biol.Sci, 2022, because these studies, unlike the quoted studies, provided absolute as well as relative cell numbers, and also carefully controlled for CMV infection.
- “Elderly” is considered an inappropriate term, as it has an ageist connotation in the eyes of many adults over a certain age. The authors should use the term “older”.

Reviewer #2:

Remarks to the Author:

This manuscript examines the T cell response to the influenza virus matrix 58-66 peptide, restricted by HLA-A*0201 in humans of different age. There are four groups, newborn (cord blood), children, adults and elderly. Focusing on peptide/A2 tetramer sorted T cells, the main findings are a diversity of repertoire in TCRs in the naïve T cells of newborns, rapid appearance and then immunodominance of the BV19 TCRs with restricted pairing with alpha chains, and then a loss of these, described as a bifurcation, giving a more diverse repertoire in the elderly. These changes are matched by changes in T cell phenotype from naïve to memory to exhausted memory, which are nicely documented.

.This is an extremely thorough, well executed and highly analysed study, but I have a number of criticisms of how this is presented. The first is the sheer size of the manuscript – around 10,000 words with 8+8 figures many with ten or more panels. Every small feature is discussed in detail, making it very hard to see the wood for the trees. Fortunately they provide a short summary paragraph at the end of each section and I suggest that they shorten the manuscript drastically by focussing on these main points and just show the key evidence that leads to the conclusions in each section. The rest can be moved to supplementary data or, if there is limit on the size there, be put a separate accessible site. The total data set will be valuable to a few specialists but as presented now it presents too big a

challenge to the average journal reader.

I also have some concerns about novelty of some of the key findings. They should acknowledge the original discovery that the T cell response to this peptide was dominated by BV19, then called V β 17, by Moss et al in 1991 (PNAS 88, 8987-8990, doi:10.1073/pnas.88.20.8987) and also discuss the structural features that select the RS/IRSS motif in the VBCDR3, that was first described in reference 52. Also they should mention that Lawson et al reported in 2001 the lack of TCRBV19 (V β 17) usage in cord blood and its appearance in memory T cells with this specificity in early childhood. Similarly the changes in phenotype of CD8 T cells in early life are well known. On the other hand the analysis of this T cell response in the elderly is more novel, although the broader phenotype changes are already known. In contrast, the detailed single cell sequencing analysis stratified by age, specificity and TCR use is novel.

My third problem is with some of the bioinformatics. While some of the analyses are widely used and well understood, others are not (at least to me, and I accept that I may not be typical here). Figure 3 is particularly difficult showing PAGA analysis and clustering to reveal the bifurcation; this is important as it appears in the title but it is unclear exactly what it is. Similarly, the Loess curves and pseudotime values are not fully explained. Although readers could head to Google and spend much time trying to work it out, a simple explanation of what these analyses are and what they show is needed, given that this is for Nature Immunology rather than a specialist Bioinformatics journal.

The TCR analyses are mostly understandable except for figure 4e. Don't 4a and 4b alone make the point eloquently? Maybe they could put the rest into supplementary data. Also could Figures 6, 7 and 8 be compressed into one figure with just a few panels to show the main message?

I recognise that this is a huge repository of unique data that explores an important T cell immune response in unprecedented detail, but as a paper in its present form it is very hard to tease out the novel take home messages.

Author Rebuttal to Initial comments

We immensely thank the Reviewers for their comments and insightful suggestions, which allowed us to greatly improve the manuscript. We also thank the Editors for giving us the opportunity of re-submitting the revised version of our manuscript to *Nature Immunology*.

After further discussions with the Editor, we have decided to re-submit our manuscript as a *Nature Immunology Resource* manuscript.

Following the comments from the Editor and the Reviewers, we have substantially revised our manuscript, as outlined below.

RESPONSES TO REVIEWERS' COMMENTS:

We thank the Reviewers for their appreciation of our study: "The study is well designed and well performed, and brings new information to the field. Scope and depth of work is strong." (Reviewer 1); "This is an extremely thorough, well executed and highly analysed study.... huge

repository of unique data that explores an important T cell immune response in unprecedented detail” (Reviewer 2).

We respond to Reviewer’s questions and comments in a point-by-point form.

Reviewer #1:

The manuscript by van de Sandt et al. represents an interesting and potentially important study of Ag-specific clonal T cell population changes across ages. This cross-sectional study begins with global phenotype analysis of different T cell subsets and compare them to phenotypes of Influenza virus-specific CD8 T cells directed against the Matrix peptide M158 bound to HLA-A2 in different age groups, followed by transcriptional and TCR diversity, component utilization and motif utilization analysis across aging. The analysis collectively suggests that the very diverse repertoires in the neonates progressively narrow down to generate a repertoire dominated by highly functional, public TCR clonotypes associated with strong, broad and functionally optimal recognition of the pMHC. Surprisingly, such cells do not appear to persist in most older adults, being partially replaced by private TCR clonotypes whose transcriptional signatures resembled those of neonatal and childhood T cell profiles and gravitated more towards central memory phenotypes. The authors then performed extensive TCR structure-function analysis following transfection of TCRs from participants of different ages into HEK293 cells, and demonstrate that private clonotypes from older adults generally exhibit lower CD8 dependency, higher activation and lower recognition of peptide epitope variants. The study is well designed and well performed, and brings new information to the field. Scope and depth of work is strong. However, several issues need to be addressed before it can be recommended for publication

Major criticisms (all must be addressed by revision or experimentally):

1. The title of the study and the discussion related to it uses the term “bifurcation”, which in my opinion is misleading. Bifurcation implies moving in different directions from the same starting point. However, all the older adults in this type of study had to be adults at some point in time, and the authors are really studying (in a cross-sectional manner, and therefore not in the same individuals) how the Ag-response changes with aging. So the phenomena the authors discuss are much better described as clonotype or repertoire shifts. This is particularly true because they document partial, and not complete, replacement of public clonotypes with private ones with aging.

We thank the Reviewer for their suggestion. We identified older adult private TCR features in the newborn TCR repertoire but not the exact older adult clonotypes. As antigen-specific CD8⁺ T cells are relatively low in the naïve population and mainly consist of singletons (instead of clonally-expanded TCRs), it would require us to sequence a much larger set of antigen-specific T cells to provide a higher level of certainty whether private older adult clonotypes would be present at birth. The volume of blood from newborns is however limited, thus we cannot sequence enough

tetramer-specific CD8⁺ T cells to provide a solid answer whether private clonotypes are present at birth. We can only speculate that they are as we have detected the private older adult TCR features within younger TCR repertoires at relatively low frequencies (explained in lines 720-721). We have therefore followed the Reviewers suggestion and replaced “bifurcation” with “repertoire shifts” in both the title and the manuscript.

The title now reads:

Newborn and child-like molecular signatures in older adults stem from age-related TCR repertoire shifts across the human lifespan

2. No cell numbers are provided for the analysis. This is a critical data point to interpret what is going on – are there relative or absolute changes in different cell subsets.

We agree with the Reviewer that cell frequencies do not always match numerical changes. Hence, following the Reviewer’s suggestion, we have now included data that show the number of A2/M1⁺phenotype⁺ cells / 10⁶ CD8⁺ T cells in Supplemental Figure 1. We hope the Reviewer is pleased to see that the numerical changes closely reflect the A2/M1-specific CD8⁺ T cell phenotypic frequencies plotted in Figure 1. A side-by-side comparison of the frequencies in Figure 1 (left) and numbers in Sup Fig 1 (right) is plotted below.

3. Directly related to #2, authors need to report the status of their participants for CMV infection, to exclude it as a potential confounder. Several graphs show bimodal distributions or outliers of data (e.g. Fig. 1i, with Ag-specific Tcm, and somewhat with Tn and Tem), and one would want to know whether there were any impact of CMV carriage.

We thank the Reviewer for this excellent suggestion. Following the Reviewer’s comment, we have performed CMV testing for the donors with available plasma. We have included the CMV status (based on CMV IgG antibodies) in the revised version of our manuscript. The results for individual donors can be found in Supplementary Table 1. Unfortunately, splitting the donors based on their CMV status did not have enough power to perform statistical analyses. However, we have included the split data in this rebuttal letter for the Reviewer (please see below). Consistent with the literature, we found a trend for higher Temra and CD57 frequencies in CMV-positive donors, however no clear trends were observed for A2/M1-specific CD8⁺ T cells.

We have also re-analysed CDR3 motifs based on the CMV status. Again, splitting the data did not have enough power for statistical analyses. Furthermore, the large spread between CMV+ and CMV- donors makes it particularly difficult to observe any clear trends. However, adult CMV+ donors seem to have a higher frequency of GG- and RS-expressing clonotypes. It is difficult to establish what the trend is in CMV+ vs CMV- older adult donors as we have only a single older adult donor with a confirmed CMV-negative status.

4. Staining experiments to ascertain protein expression are needed to document whether there are cells with virtual memory/antigen-independent memory phenotypes in any of the tested populations. Transcriptomic data could be misleading in this regard.

We agree with the Reviewer that ideally, we would confirm our statement on virtual memory or memory T cells with a naïve phenotype with additional flow cytometry staining to confirm the expression of CD49d and CXCR3. Unfortunately, we used all the PBMCs from these donors in the scRNAseq experiments, and therefore were unable to confirm their expression with flow cytometry. Following the Reviewer's comment, we have removed the statement about virtual memory from Discussion.

5. Figure 7, for activation studies, cells were enriched using magnetic beads. What were the surface phenotypes of enriched cells and were they comparable between the age groups? Different differentiation states could have very well caused differences in proliferation and activation and this needs to be controlled for. This data is also necessary to interpret the results.

We thank the Reviewer for their feedback on Figure 7. We would like to clarify that for the proliferation assay, we stimulated PBMCs directly *ex vivo* without prior enrichment for antigen-specific CD8⁺ T cells, as described in the Methods section. The frequencies of antigen-specific CD8⁺

cells at day 0 is, however, relatively low and difficult to detect without prior enrichment. We therefore used the *ex vivo* frequencies established in Figure 1 for these particular donors (individual phenotypes of these donors at day 0 are shown below). No enrichments were performed on subsequent days.

We have now clarified this in the figure legend, which now reads (page 25):

“Total numbers (a) and fold-increase (b) of A2/M1₅₈-tetramer⁺CD8⁺ T cells from day 0 (ex vivo tetramer enrichment, due to low frequency), day 3, 4, 5, 6, 7 and 10 following in vitro M1₅₈₋₆₆ peptide stimulation of newborn, child, adult and older adult PBMCs (no prior enrichment).”

We have also included data on the individual donor *ex vivo* phenotype frequencies on day 0 of the proliferation assay (new Supplementary Figure 6a), demonstrating predominantly naïve phenotype in newborns and robust memory populations in children, adults and older adults (except BP127).

The relevant lines in the manuscript now read (line 524-527):

“Except for one older adult, expressing a predominant T_{cm} at d0 (Supplementary Fig.6a), who had the largest fold-change on d10, despite low cell numbers (Fig.7a,b). Newborn A2M1⁺CD8⁺ T-cells had low proliferation capacity, reflecting their naïve phenotype (Fig.7b; Supplementary Fig.6a).”

6. Critically, the authors have not provided a biological explanation for the observed repertoire diversification and replacement of public with private clonotypes with aging. Where are the private clonotypes coming from and why? Do T cells carrying the public clonotypes move into tissues perhaps, or do they die due to lack of maintenance? Do the private clonotypes that arise in older participants have high affinity/avidity for self pMHC to allow survival? Can the authors do any experiments to address this issue? I understand that a definitive proof may be very difficult to obtain, but at least some insight would help,

We thank the Reviewer for these excellent suggestions. To fully unravel the underlying mechanisms, we would need to follow multiple HLA-A2⁺ individuals over their lifespan. We currently do not have access to such samples.

The questions raised by the Reviewer are relevant and could provide great insights. We are currently working on a follow-up study which addresses some of these questions, but such studies require more time to execute. Furthermore, considering the overall length and depth-of the current manuscript, which provides novel insights into the TCR repertoire shifts across the human lifespan, there are limitation on the additional data which can be included.

Furthermore, the data presented in the manuscript, as appreciated by both Reviewers, represents a valuable resource which can be used by others. Thus, in consultation with the Editor, we have decided to resubmit the manuscript in a *Nature Immunology* Resource format, which allows others to access our data and has less emphasis on the underlying mechanisms. We hope the Reviewer agrees.

Other comments:

- Ln. 259 – were there 793 cells total, or per group, for this analysis? Please elaborate

Apologies for the confusion, the sentence now reads (line 190-191):

“Ex vivo-isolated tetramer-specific A2/M158⁺CD8⁺ T-cells were sorted for full-length single-cell transcriptome analysis (scRNAseq). 793 cells across all age groups passed quality control and were used for downstream analyses”

- There is a typo on ln. 381 – should be PAGA, not PAGE.

We have corrected the typographical error.

- Ln. 653, likely should be “children and elderly” and not children and adult.

This has been corrected accordingly.

- I urge the authors to carefully examine references and make sure that they credit original work. For example, the reference for thymic involution timing in humans cannot be the THome reference (#13); at a minimum, the elegant studies of Douek and colleagues [Douek, DC et al., *Nature* 1998], should be quoted instead, and I urge the authors to look to earlier work by groups that discovered thymus (Miller, *JFAP*; Good RA; Waksman, B) or Barton F. Haynes to assure proper attribution. Second, the statement in ln.95-98 is supported by references 11, 15 and 18-20. However, the statement that Tn cells are lost with aging and terminally differentiated cells increase, needs additional references Wertheimer, A.M. et al., *J. Immunol* 2014; and Thyagarajan B., et al., *J.Geront.Biol.Sci*, 2022, because these studies, unlike the quoted studies, provided absolute as well as relative cell numbers, and also carefully controlled for CMV infection.

We thank the Reviewer for the suggested references. We have changed them accordingly.

- “Elderly” is considered an inappropriate term, as it has an ageist connotation in the eyes of many adults over a certain age. The authors should use the term “older”.

We thank the Reviewer for bringing this to our attention. According to the Reviewer’s suggestion, we have changed ‘elderly’ to ‘older adults’ throughout the manuscript and the figures.

Reviewer #2:

This manuscript examines the T cell response to the influenza virus matrix 58-66 peptide, restricted by HLA-A*0201 in humans of different age. There are four groups, newborn (cord blood), children, adults and elderly. Focusing on peptide/A2 tetramer sorted T cells, the main findings are a diversity of repertoire in TCRs in the naïve T cells of newborns, rapid appearance and then immunodominance of the BV19 TCRs with restricted pairing with alpha chains, and then a loss of these, described as a bifurcation, giving a more diverse repertoire in the elderly. These changes are matched by changes in T cell phenotype from naïve to memory to exhausted memory, which are nicely documented.

This is an extremely thorough, well executed and highly analysed study, but I have a number of criticisms of how this is presented. The first is the sheer size of the manuscript – around 10,000 words with 8+8 figures many with ten or more panels. Every small feature is discussed in detail, making it very hard to see the wood for the trees. Fortunately they provide a short summary paragraph at the end of each section and I suggest that they shorten the manuscript drastically by focussing on these main points and just show the key evidence that leads to the conclusions in each section. The rest can be moved to supplementary data or, if there is limit on the size there, be put a separate accessible site. The total data set will be valuable to a few specialists but as presented now it presents too big a challenge to the average journal reader.

We thank the Reviewer for their suggestion. Following the Reviewer’s comment, we have shortened the manuscript by 39%, and in consultation with the Editor, we have decided to resubmit the manuscript in a *Nature Immunology* Resource format, as this allows us to keep the data together in a single manuscript. Our original manuscript had 11,881 words (Introduction, Results, Discussion) of which we removed 4,627 words in the revised version of our manuscript.

I also have some concerns about novelty of some of the key findings. They should acknowledge the original discovery that the T cell response to this peptide was dominated by BV19, then called V β 17, by Moss et al in 1991 (PNAS 88, 8987-8990, doi:10.1073/pnas.88.20.8987) and also discuss the structural features that select the RS/IRSS motif in the VBCDR3, that was first described in reference 52. Also they should mention that Lawson et al reported in 2001 the lack of TCRBV19 (V β 17) usage in cord blood and its appearance in memory T cells with this specificity

in early childhood. Similarly the changes in phenotype of CD8 T cells in early life are well known. On the other hand the analysis of this T cell response in the elderly is more novel, although the broader phenotype changes are already known. In contrast, the detailed single cell sequencing analysis stratified by age, specificity and TCR use is novel.

We thank the Reviewer for recognizing the novelty of our work, specifically the single-cell sequencing analysis combined with in-depth single-cell paired TCR $\alpha\beta$ repertoire and the functional/avidity analyses. We appreciate the Reviewer's suggestion of emphasizing more how previous studies fit with our data. We had already referenced previous studies by Stewart-Jones (Ref 47) and Lawson (Ref 29), and have now also included a reference to Stewart-Jones (Ref 47; Results lines: 213, 275, 435, 658; Discussion Lines 729, 734, 756) and Lawson (Ref 29, Introduction Line 99, 120; Results line 379; Discussion lines 745 and 747). We thank the Reviewer for their suggestion to include the discovery of A2/M1₅₈₋₆₆-specific CD8⁺ T cell repertoire being dominated by TRBV19, we have now included references to Moss in Introduction (Ref 43, Introduction Line 121; Results line 390).

My third problem is with some of the bioinformatics. While some of the analyses are widely used and well understood, others are not (at least to me, and I accept that I may not be typical here). Figure 3 is particularly difficult showing PAGA analysis and clustering to reveal the bifurcation; this is important as it appears in the title but it is unclear exactly what it is. Similarly, the Loess curves and pseudotime values are not fully explained. Although readers could head to Google and spend much time trying to work it out, a simple explanation of what these analyses are and what they show is needed, given that this is for Nature Immunology rather than a specialist Bioinformatics journal.

We appreciate the Reviewer's suggestion to clarify why we performed the PAGA, Loess and pseudotime analyses and how to read the individual panels. We have now rewritten the PAGA section and simplified the Figure by taking out the growth rate and including a bubble plot to show the mean gene expression of key genes over the pseudotime (now Figure 3d and below), which we hope will be more intuitive to most readers, as per Reviewer's suggestion.

As for the bifurcation, to avoid any confusion, we have now used the phrase 'clonotype or repertoire shift' throughout the manuscript, including the title, as suggested by Reviewer 1.

The TCR analyses are mostly understandable except for figure 4e. Don't 4a and 4b alone make the point eloquently? Maybe they could put the rest into supplementary data. Also could Figures 6, 7 and 8 be compressed into one figure with just a few panels to show the main message?

We thank the Reviewer for this comment. The additional value of the tangle plots in Figure 4e is that it also shows an enrichment score above background. However, we agree with the Reviewer that this additional information is not needed. Hence, we have removed the tangle plots from Figure 4.

As for combining Figures 6, 7 and 8, we felt that this may be confusing to the reader, as Figure 6 is related to antigen-specific analysis directly *ex vivo*, Figure 7 represents *in vitro* proliferation assay, while Figure 8 is based on TCR-expressing cell lines. However, we agree with the Reviewer that the data can be overwhelming, especially in Figure 8. Hence, we have now reduced the number of panels in Figure 8 from 14 to 10 panels to support our main message.

I recognise that this is a huge repository of unique data that explores an important T cell immune response in unprecedented detail, but as a paper in its present form it is very hard to tease out the novel take home messages.

We greatly appreciate the Reviewer's feedback to focus more on the main take home message and less on the details. Following the Reviewer's suggestions, we have substantially shortened the manuscript by 39% (Introduction, Results and Discussion) and removed details focussing on single donors. We are confident that this has improved the readability of our manuscript and has offered a more prominent placement of the main take home messages.

Decision Letter, first revision:

10th Jul 2023

Dear Dr. Kedzierska,

Thank you for submitting your revised manuscript "Newborn and child-like molecular signatures in older adults stem from age-related TCR repertoire shifts across the human lifespan" (NI-A35606A). It has now been seen by the original referees and their comments are below. The reviewers find that the paper has improved in revision, and therefore we'll be happy in principle to publish it in Nature Immunology, pending minor textual revisions to satisfy the referees' final requests and to comply

with our editorial and formatting guidelines.

We will now perform detailed checks on your paper and will send you a checklist detailing our editorial and formatting requirements in about a week. Please do not upload the final materials and make any revisions until you receive this additional information from us.

If you had not uploaded a Word file for the current version of the manuscript, we will need one before beginning the editing process; please email that to immunology@us.nature.com at your earliest convenience.

Thank you again for your interest in Nature Immunology. Please do not hesitate to contact me if you have any questions.

Sincerely,

Nick Bernard, PhD
Senior Editor
Nature Immunology

Reviewer #1 (Remarks to the Author):

The revised manuscript by van de Sant et al, now submitted as a resource paper, has been considerably improved. The Resource Paper format is also well suited for this important set of varied and well-curated data. Overall, it is my assessment that the manuscript now meets the essential criteria of Nature Immunology, and I recommend acceptance once the remaining issues, listed below, are addressed.

I am satisfied with the authors' answers to my criticisms #1, 2 and 4-6, including the response to other criticisms. The following changes need to be made to address issue #3 and in part #2:

1. Original criticism #2; Absolute numbers should be discussed in the text in addition to percentages.
2. Original criticism #3; CMV data should be discussed in the text and showed in the supplementary data. (As the authors do that, I am asking that they increase the size of symbols on the figures to improve legibility.) It is critical to assign age-related changes to aging, and persistent infection changes to persistent infection. I understand that the dataset is underpowered to an extent, but even this level of analysis shows that influenza-specific responses do not appear to be majorly modulated by the concurrent CMV infection, which was one of the key points of asking for this analysis.
3. Accordingly, the authors should change the text on lines 162-165. It is not clear that the finding of low Temra fraction in response to an acute, if recurrently exposed, viral challenge should be surprising (line 162). Also, high Temra cell levels are found in persistent infections, and may be further potentiated over time during aging, but only in people with CMV infection. Aging in the absence of CMV does not lead to an absolute accumulation of Temra cells, consistent with the idea that repeated and frequent stimulation is needed to drive T cells to this stage of differentiation. The analysis of CD57 further confirms the above point.
4. A separate issue is treatment of PD-1 as an exhaustion marker – it cannot be considered as such in the absence of other exhaustion markers, because alone, it is only diagnostic of T cell activation. At a minimum the authors need to make that note. Changes in PD-1 expression with aging are still

interesting, but their meaning is unclear at the moment.

Minor comments:

- Ln. 101-102 – increase in TCR diversity in octogenarians could be a consequence of a survivor effect, whereby many people with extreme TCR constriction may have been purged from the population. The authors may want to make a comment to that effect.
- Ln 94. – last word should be in plural (lack, not lacks).

Reviewer #2 (Remarks to the Author):

The authors have revised the manuscript extensively and have shortened it as requested. They have addressed alley concerns. It is now much easier to read and the points are clear and well made. I am happy with this new version.

Author Rebuttal, first revision:

We thank the Reviewers and the Editor for their constructive suggestions, which allowed us to further improve our manuscript. We also thank the Editor for giving us the opportunity to publish our manuscript as a resource article in *Nature Immunology*.

We respond to the remaining of the Reviewers' questions and suggestions in a point-by-point form below and we have addressed the Editors comments in the Author Guidance form.

RESPONSES TO REVIEWERS' COMMENTS

We are excited to hear that both Reviewers found that the previously proposed and implemented changes “considerable improved” our manuscript (Reviewer 1) and “have addressed alley concerns “(Reviewer 2).

Reviewer #1:

The revised manuscript by van de Sant et al, now submitted as a resource paper, has been considerably improved. The Resource Paper format is also well suited for this important set of varied and well-curated data. Overall, it is my assessment that the manuscript now meets the essential criteria of Nature Immunology, and I recommend acceptance once the remaining issues, listed below, are addressed.

I am satisfied with the authors' answers to my criticisms #1, 2 and 4-6, including the response to other criticisms. The following changes need to be made to address issue #3 and in part #2:

1. Original criticism #2; Absolute numbers should be discussed in the text in addition to percentages.

We thank the Reviewer for bringing this to our attention. We have now included the following sentence referring to T_{naive} , T_{cm} , T_{em} , T_{scm} and T_{emra} populations (page 4):

“Similar trends were observed for absolute numbers of A2/M158⁺ phenotype⁺ cells/10⁶ CD8⁺ T-cells (Extended Data Fig.1b).”

Absolute cell numbers for CD57 and PD-1 expressing cells were already provided in the text with a direct reference to Extended Data Fig 1b.

2. Original criticism #3; CMV data should be discussed in the text and showed in the supplementary data. (As the authors do that, I am asking that they increase the size of symbols on the figures to improve legibility.) It is critical to assign age-related changes to aging, and persistent infection changes to persistent infection. I understand that the dataset is underpowered to an extent, but even this level of analysis shows that influenza-specific responses do not appear to be majorly modulated by the concurrent CMV infection, which was one of the key points of asking for this analysis.

We thank the Reviewer for their added suggestion. We have now included the CMV figures with larger symbols in the manuscript (Extended Data Fig 2). The text now reads (page 4):

“CMV testing would reveal whether CMV status affects total CD8⁺ and/or A2/M158-specific CD8⁺ T-cells phenotypes (Supplementary Table 1). Although our dataset was underpowered, A2/M158-specific CD8⁺ T-cell responses were not modulated by concurrent CMV infection, conversely to total CD8⁺ T-cells (Extended Data Fig.2).”

3. Accordingly, the authors should change the text on lines 162-165. It is not clear that the finding of low Temra fraction in response to an acute, if recurrently exposed, viral challenge should be surprising (line 162). Also, high Temra cell levels are found in persistent infections, and may be further potentiated over time during aging, but only in people with CMV infection. Aging in the absence of CMV does not lead to an absolute accumulation of Temra cells, consistent with the idea that repeated and frequent stimulation is needed to drive T cells to this stage of differentiation. The analysis of CD57 further confirms the above point.

We thank the Reviewer for bringing this to our attention. We have now further clarified this in the manuscript. The sentence now reads (page 4):

“Ample T_{emra} populations are characteristic for immunosenescence in older adults experiencing chronic CMV infections^{12,17}”

4. A separate issue is treatment of PD-1 as an exhaustion marker – it cannot be considered as such in the absence of other exhaustion markers, because alone, it is only diagnostic of T cell activation. At a minimum the authors need to make that note. Changes in PD-1 expression with aging are still interesting, but their meaning is unclear at the moment.

We thank the Reviewer for this suggestion and have now changed the sentence accordingly (page 4):

“PD-1 expression, an immune checkpoint marker which can be associated with TCR activation, immunosuppression and/or exhaustion,.....”

Minor comments:

- Ln. 101-102 – increase in TCR diversity in octogenarians could be a consequence of a survivor effect, whereby many people with extreme TCR constriction may have been purged from the population. The authors may want to make a comment to that effect.

We thank the Reviewer for this insight. However, as we had to shorten substantially the manuscript, we have now omitted the sentence.

- Ln 94. – last word should be in plural (lack, not lacks).
The typographical error has been corrected.

Reviewer #2:

The authors have revised the manuscript extensively and have shortened it as requested. They have addressed all concerns. It is now much easier to read and the points are clear and well made. I am happy with this new version.

We thank the Reviewer for their kind words and we are pleased to hear that the previous changes have improved the readability of the manuscript.

Final Decision Letter:

Dear Dr. Kedzierska,

I am delighted to accept your manuscript entitled "Newborn and child-like molecular signatures in

older adults stem from TCR shifts across human lifespan" for publication in an upcoming issue of Nature Immunology.

Over the next few weeks, your paper will be copyedited to ensure that it conforms to Nature Immunology style. Once your paper is typeset, you will receive an email with a link to choose the appropriate publishing options for your paper and our Author Services team will be in touch regarding any additional information that may be required.

Please note that *Nature Immunology* is a Transformative Journal (TJ). Authors may publish their research with us through the traditional subscription access route or make their paper immediately open access through payment of an article-processing charge (APC). Authors will not be required to make a final decision about access to their article until it has been accepted. [Find out more about Transformative Journals](https://www.springernature.com/gp/open-research/transformative-journals).

Authors may need to take specific actions to achieve [compliance with funder and institutional open access mandates](https://www.springernature.com/gp/open-research/funding/policy-compliance-faqs). If your research is supported by a funder that requires immediate open access (e.g. according to [Plan S principles](https://www.springernature.com/gp/open-research/plan-s-compliance)) then you should select the gold OA route, and we will direct you to the compliant route where possible. For authors selecting the subscription publication route, the journal's standard licensing terms will need to be accepted, including [self-archiving policies](https://www.springernature.com/gp/open-research/policies/journal-policies). Those licensing terms will supersede any other terms that the author or any third party may assert apply to any version of the manuscript.

Your paper will be published online soon after we receive your corrections and will appear in print in the next available issue. Content is published online weekly on Mondays and Thursdays, and the embargo is set at 16:00 London time (GMT)/11:00 am US Eastern time (EST) on the day of

publication. Now is the time to inform your Public Relations or Press Office about your paper, as they might be interested in promoting its publication. This will allow them time to prepare an accurate and satisfactory press release. Include your manuscript tracking number (NI-A35606B) and the name of the journal, which they will need when they contact our office.

About one week before your paper is published online, we shall be distributing a press release to news organizations worldwide, which may very well include details of your work. We are happy for your institution or funding agency to prepare its own press release, but it must mention the embargo date and Nature Immunology. Our Press Office will contact you closer to the time of publication, but if you or your Press Office have any enquiries in the meantime, please contact press@nature.com.

Also, if you have any spectacular or outstanding figures or graphics associated with your manuscript - though not necessarily included with your submission - we'd be delighted to consider them as candidates for our cover. Simply send an electronic version (accompanied by a hard copy) to us with a possible cover caption enclosed.

If you have not already done so, we strongly recommend that you upload the step-by-step protocols used in this manuscript to the Protocol Exchange. Protocol Exchange is an open online resource that allows researchers to share their detailed experimental know-how. All uploaded protocols are made freely available, assigned DOIs for ease of citation and fully searchable through nature.com. Protocols can be linked to any publications in which they are used and will be linked to from your article. You can also establish a dedicated page to collect all your lab Protocols. By uploading your Protocols to Protocol Exchange, you are enabling researchers to more readily reproduce or adapt the methodology you use, as well as increasing the visibility of your protocols and papers. Upload your Protocols at www.nature.com/protocolexchange/. Further information can be found at www.nature.com/protocolexchange/about .

Please note that we encourage the authors to self-archive their manuscript (the accepted version before copy editing) in their institutional repository, and in their funders' archives, six months after publication. Nature Portfolio recognizes the efforts of funding bodies to increase access of the research they fund, and strongly encourages authors to participate in such efforts. For information about our editorial policy, including license agreement and author copyright, please visit www.nature.com/ni/about/ed_policies/index.html

An online order form for reprints of your paper is available at <https://www.nature.com/reprints/author->

reprints.html"><https://www.nature.com/reprints/author-reprints.html>. Please let your coauthors and your institutions' public affairs office know that they are also welcome to order reprints by this method.

Sincerely,

Nick Bernard, PhD
Senior Editor
Nature Immunology